# Regulation of cellular cholesterol distribution via non-vesicular lipid transport at ER-Golgi contact sites

Tomoki Naito [1], Haoning Yang [1], Dylan Hong Zheng Koh [1], Divyanshu Mahajan[2], Lei Lu [2] & Yasunori Saheki [1,3] ✉

Abnormal distribution of cellular cholesterol is associated with numerous diseases, including cardiovascular and neurodegenerative diseases. Regulated transport of cholesterol is critical for maintaining its proper distribution in the cell, yet the underlying mechanisms remain unclear. Here, we show that lipid transfer proteins, namely ORP9, OSBP, and GRAMD1s/Asters (GRAMD1a/GRAMD1b/GRAMD1c), control non-vesicular cholesterol transport at points of contact between the ER and the trans-Golgi network (TGN), thereby maintaining cellular cholesterol distribution. ORP9 localizes to the TGN via interaction between its tandem α-helices and ORP10/ORP11. ORP9 extracts PI4P from the TGN to prevent its overaccumulation and suppresses OSBP-mediated PI4P-driven cholesterol transport to the Golgi. By contrast, GRAMD1s transport excess cholesterol from the Golgi to the ER, thereby preventing its build-up. Cells lacking ORP9 exhibit accumulation of cholesterol at the Golgi, which is further enhanced by additional depletion of GRAMD1s with major accumulation in the plasma membrane. This is accompanied by chronic activation of the SREBP-2 signalling pathway. Our findings reveal the importance of regulated lipid transport at ER-Golgi contacts for maintaining cellular cholesterol distribution and homeostasis.

Sterol is a major lipid component of eukaryotic cellular membranes and is essential for membrane integrity and cell signalling[1]. In mammals, cholesterol is the dominant sterol, representing approximately 20% of total cellular lipids[2]. Cholesterol and its metabolites, such as oxysterol and steroid hormones, play diverse roles in physiology[3,4]. Hence, dysregulation of cholesterol metabolism is linked to numerous health disorders, including neurodegenerative and cardiovascular diseases[5,6].

Cholesterol is unevenly distributed across the cell, with highest levels of enrichment in the plasma membrane (PM)[1,2,7–11]. Cholesterol is either synthesized de novo in the endoplasmic reticulum (ER) or taken up from external sources, primarily through the endocytosis of low-density lipoproteins (LDLs)[6]. Regardless of the source, cellular cholesterol is then delivered to the PM, where cholesterol contributes to almost half of the total lipids in this bilayer[7–9]. Importantly, sterol regulatory element binding proteins (SREBPs), which sense levels of cellular cholesterol and control the expression of genes involved in cholesterol biosynthesis and uptake, are located in the ER[12–14]. Thus, cells must deliver cholesterol to the ER so that SREBPs can monitor cellular cholesterol levels and adjust its synthesis and uptake. Non-vesicular transport of cholesterol facilitated by lipid transfer proteins (LTPs) plays a critical role in this process[15–19]. While most lipids, including cholesterol, are also transported by vesicular transport, this mode of lipid transport is often non-selective, causing the intermixing of membrane lipids between different cellular organelles and membranes. By contrast, non-vesicular lipid transport is generally more

[1]Lee Kong Chian School of Medicine, Nanyang Technological University, Singapore 308232, Singapore. [2]School of Biological Sciences, Nanyang Technological University, Singapore 637551, Singapore. [3]Institute of Resource Development and Analysis, Kumamoto University, Kumamoto 860-0811, Japan. ✉e-mail: yasunori.saheki@ntu.edu.sg

selective, allowing cells to counteract the lipid intermixing caused by vesicular transport[20–24]. Hence, LTPs not only help the ER sense cellular cholesterol levels, but they also play a major role in maintaining cellular cholesterol distribution.

LTP-mediated non-vesicular transport of cholesterol relies on the accessibility (i.e., the chemical activity) of cholesterol in cellular membranes. At steady state, the majority of PM cholesterol is "inaccessible" for LTPs due to the formation of complexes between cholesterol and other membrane lipids, including phospholipids and sphingomyelin[25–33]. However, when levels of PM cholesterol increase beyond the sequestration capacity of other membrane lipids, the excess pool of accessible cholesterol is transported to the ER via LTPs to restore the balance of chemical activity between these two cellular compartments[17]. This results in suppression of the SREBP pathway to maintain homeostasis[31,34–38]. Numerous LTPs have been suggested to participate in this process[17,39–41]. Some of these LTPs function primarily at membrane contact sites, where the ER forms close contact with other organelles and membranes[22,39,42–44]. Growing evidence suggest that evolutionarily conserved and ER-anchored LTPs, namely the GRAMD1s/Asters (GRAMD1a/Aster-A, GRAMD1b/Aster-B, and GRAMD1c/Aster-C), form homo- and hetero-meric complexes and play major roles in accessible cholesterol transport to the ER at membrane contact sites[39,43]. GRAMD1s possess an N-terminal GRAM domain, which acts as a co-incidence detector for accessible cholesterol and anionic lipids, including phosphatidylserine (PS)[45]. The GRAM domain is followed by a StART-like domain, which extracts and transports sterol[45–54]. When levels of PM cholesterol are elevated, GRAMD1s move to sites of contact between the ER and PM (i.e., ER-PM contact sites) via interactions between the GRAM domain and accessible PM cholesterol. Once at an ER-PM contact site, the GRAMD1 StART-like domain transports cholesterol to the ER to suppress SREBP-2 signalling[45,50,53–55]. Given the property of the GRAMD1 GRAM domain to sense elevations of accessible cholesterol in membranes, GRAMD1s may also transport accessible cholesterol to the ER at other membrane contact sites and prevent build-up of cholesterol in various cellular membranes and organelles to maintain homeostasis.

Compared to the regulatory mechanisms by which cholesterol is transported from the PM to the ER, we know very little about how cholesterol is transported from the ER to the PM. Another major family of evolutionarily conserved LTPs involved in intracellular cholesterol transport are the oxysterol binding protein (OSBP)-related proteins (ORPs), which include OSBP and 11 other ORPs (ORP1 to ORP11)[56]. ORPs share a common lipid-harboring domain, OSBP-related ligand-binding domain (ORD), and transport selective lipids, such as phosphoinositides and sterols, between various cellular compartments. Many ORPs, including OSBP, also have an N-terminal pleckstrin homology (PH) domain, which binds specific lipids within cellular membranes, as well as two phenylalanine in an acidic tract (FFAT) motifs, which bind ER-anchored VAMP associated proteins (VAPs), thereby targeting ORPs to the ER. In this way, ORPs facilitate lipid transport at membrane contact sites formed between the ER and other organelles[44,56–59]. In particular, OSBP senses phosphatidylinositol 4-phosphate (PI4P) and localizes to membrane contact sites between the ER and other organelles, including the trans-Golgi network (TGN), endosomes, and lysosomes, transporting cholesterol from the ER to these organelles in exchange for PI4P[60–64]. Inhibition of OSBP results in aberrant buildup of PI4P in the TGN[62] and endosomes[61], as well as reduced levels of cholesterol in the Golgi and post-Golgi membranes, including the PM[65]. This suggests that OSBP plays a major role in maintaining levels of cholesterol in the PM. However, the molecular mechanisms that control the activity of OSBP are not well understood. Such mechanisms likely play a major role in determining the distribution of cellular cholesterol and in maintaining lipid homeostasis.

While the Golgi complex serves as a central hub for membrane trafficking, the TGN forms membrane contact sites with the ER and receives newly synthesized lipids, including cholesterol, from the ER via non-vesicular transport[42,66–68]. Thus, the TGN serves as an important relay station, linking non-vesicular transport and vesicular transport to control cellular distribution of lipids. Here, we show that three LTPs, namely ORP9, OSBP, and the GRAMD1s, act together at ER-TGN contact sites to control the abundance of cholesterol in the Golgi, thereby influencing levels of cholesterol in the PM. We performed a screen in GRAMD1 triple knock-out (TKO) cells using a novel accessible cholesterol biosensor, GRAM-H[45]. We identified ORP9 as a critical regulator of cellular cholesterol distribution. ORP9 localized to the TGN via interaction between its α-helices and ORP10/ORP11. HeLa cells lacking ORP9 exhibited accumulation of accessible cholesterol in the Golgi. Additional depletion of GRAMD1s in these cells resulted in further accumulation of accessible cholesterol in the Golgi, which was accompanied by major accumulation of accessible cholesterol in the PM. Strikingly, inhibition of OSBP in these cells was sufficient to restore normal levels of cholesterol in these cellular compartments. Mechanistically, ORP9 extracts PI4P from the TGN and inhibits OSBP-mediated PI4P-driven cholesterol transport to the Golgi, whereas GRAMD1s extract cholesterol from the Golgi to prevent its build-up. Our findings reveal that intricate cross-talk between ORP9, OSBP, and the GRAMD1s at ER-TGN contact sites is critical for maintaining cellular cholesterol distribution and homeostasis.

## Results
### ORP9 regulates the abundance of accessible cholesterol in the PM
GRAMD1s mediate the transport of accessible cholesterol from the PM to the ER[45,50,53–55]. Accordingly, HeLa cells lacking all three GRAMD1s (GRAMD1 TKO cells) showed chronic accumulation of accessible cholesterol in the PM compared to wild-type (WT) HeLa cells, as revealed by an accessible cholesterol biosensor based on the GRAM domain (Fig. 1a, b). This biosensor, called EGFP-GRAM-H, is an EGFP-tagged GRAMD1b GRAM domain with a G187L mutation[45]. Binding of EGFP-GRAM-H to the PM depended on the presence of cholesterol in the PM, as depleting cellular cholesterol by treating cells with a combination of lipoprotein-deficient serum (LPDS) and mevastatin, an HMG-CoA reductase inhibitor, resulted in complete dissociation of EGFP-GRAM-H from the PM and redistribution to the cytosol (Fig. 1a, b).

OSBP influences the levels of accessible cholesterol in post-Golgi membranes, including the PM, via its ability to transport cholesterol from the ER to the TGN at ER-TGN contact sites[60,65]. Strikingly, treatment of GRAMD1 TKO cells with OSW-1, a specific inhibitor of OSBP[69], induced dramatic dissociation of EGFP-GRAM-H from the PM (Fig. 1a, b). This suggests that OSBP plays a major role in maintaining accessible PM cholesterol in GRAMD1 TKO cells and that the TGN may play a central role in this process. One key feature of OSBP-mediated cholesterol transport to the TGN is that it is driven by counter exchange of ER-derived cholesterol for TGN-derived PI4P. TGN-derived PI4P is delivered to the ER for consumption by the ER-anchored PI4P phosphatase, Sac1[60]. Thus, PI4P plays a central role in controlling OSBP-mediated cholesterol transport to the TGN[60,62]. As the ORDs of OSBP and ORPs share a common ligand, namely PI4P[58,63], other ORPs may regulate OSBP-mediated cholesterol transport by controlling the abundance of PI4P in TGN membranes.

To test this possibility, we performed a screen looking for changes in PM cholesterol levels caused by overexpression of individual ORPs in GRAMD1 TKO cells. Localization of EGFP-GRAM-H to the PM was monitored using spinning disc confocal (SDC) microscopy (Supplementary Fig. 1a, b). Compared to GRAMD1 TKO HeLa cells expressing an mCherry control, cells expressing mRuby-tagged GRAMD1b (mRuby-GRAMD1b) showed reduced levels of GRAM-H at the PM, thereby restoring levels of accessible PM cholesterol, as we have previously reported (Supplementary Fig. 1a, b)[45,53]. Cells overexpressing

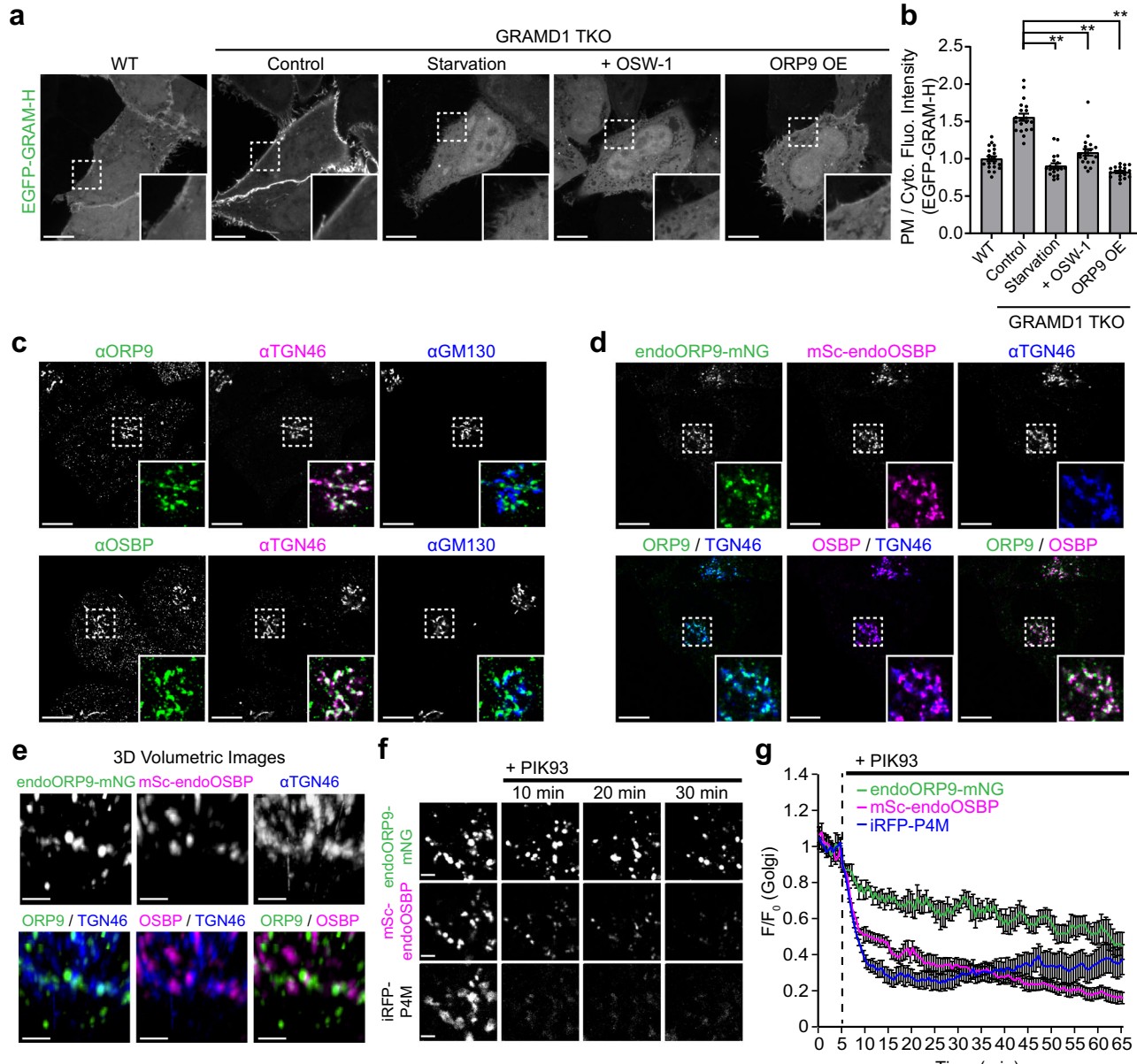

**Fig. 1 | ORP9 regulates the abundance of accessible cholesterol in the PM.**
**a** Confocal images of live wild-type (WT) HeLa cells and HeLa cells lacking GRAMD1s (GRAMD1 TKO), expressing EGFP-tagged GRAM-H (EGFP-GRAM-H) (accessible cholesterol biosensor), under indicated conditions [Starvation: treatment with DMEM supplemented with 10% LPDS and 50 μM mevastatin for 16 h; OSW-1: treatment with 20 nM OSW-1 (OSBP inhibitor) for 1 h; ORP9 OE: co-expression with mCherry-tagged ORP9 for 16 h]. Insets show at higher magnification the regions indicated by white dashed boxes. Scale bars, 10 μm. **b** Quantification of the ratio of PM signals to the cytosolic signals of EGFP-GRAM-H, as assessed by confocal microscopy and line scan analysis (mean ± SEM, $n = 20$ cells for each condition; data are pooled from two independent experiments; Dunnett's multiple comparisons test, **$P < 0.0001$). **c** Confocal images of fixed HeLa cells immunolabeled with antibodies against ORP9, OSBP, TGN46, and GM130. Scale bars, 10 μm. **d** Confocal images of fixed HeLa cells, in which mNeonGreen was tagged to the C-terminus of

ORP9 (endoORP9-mNG) and mScarlet-I was tagged to the N-terminus of OSBP (mSc-endoOSBP). To visualize TGN, cells were fixed and immunolabeled with antibodies against TGN46. Scale bars, 10 μm. **e** 3D reconstruction of SDC-SIM images of fixed HeLa cells expressing endoORP9-mNG and mSc-endoOSBP that were immunolabeled with antibodies against TGN46, mNeonGreen, and mCherry. Scale bars 1 μm. **f** Confocal images from the regions around the Golgi of a live HeLa cell expressing endoORP9-mNG and mSc-endoOSBP together with iRFP-tagged P4M (iRFP-P4M) (PI4P biosensor) that were treated with PIK93 (250 nM) for the indicated minutes. Scale bars, 2 μm. **g** Time course of normalized signals of endoORP9-mNG, mSc-endoOSBP, and iRFP-P4M at the regions around the TGN in response to PIK93 (250 nM), as assessed by confocal microscopy as shown in (**f**) (mean ± SEM, $n = 21$ cells for each condition; data are pooled from two independent experiments). Source data are provided as a Source Data file.

mRuby-tagged OSBP, ORP2, or ORP4 (mRuby-OSBP, mRuby-ORP2, or mRuby-ORP4) showed enhanced PM association of GRAM-H, indicating the role of OSBP and its close homolog, ORP4, in delivering cholesterol to the Golgi and post-Golgi membranes[60,62,65,70]. This is also consistent with the role of ORP2 in delivering cholesterol to the PM[71,72]. Interestingly, cells overexpressing mCherry-tagged ORP9 (mCherry-ORP9) exhibited lower levels of EGFP-GRAM-H at the PM (ORP9 OE in

Fig. 1a, b and Supplementary Fig. 1a, b). To assess the interplay between ORP9 and OSBP or ORP2, mCherry-tagged OSBP (mCherry-OSBP) or mRuby-ORP2 was expressed with or without tagBFP-tagged ORP9 (tagBFP-ORP9) in GRAMD1 TKO cells, and the association of GRAM-H to the PM was monitored. Remarkably, enhanced PM association of GRAM-H by OSBP overexpression was largely suppressed by co-expression of ORP9 (Supplementary Fig. 1c, d). By contrast, enhanced

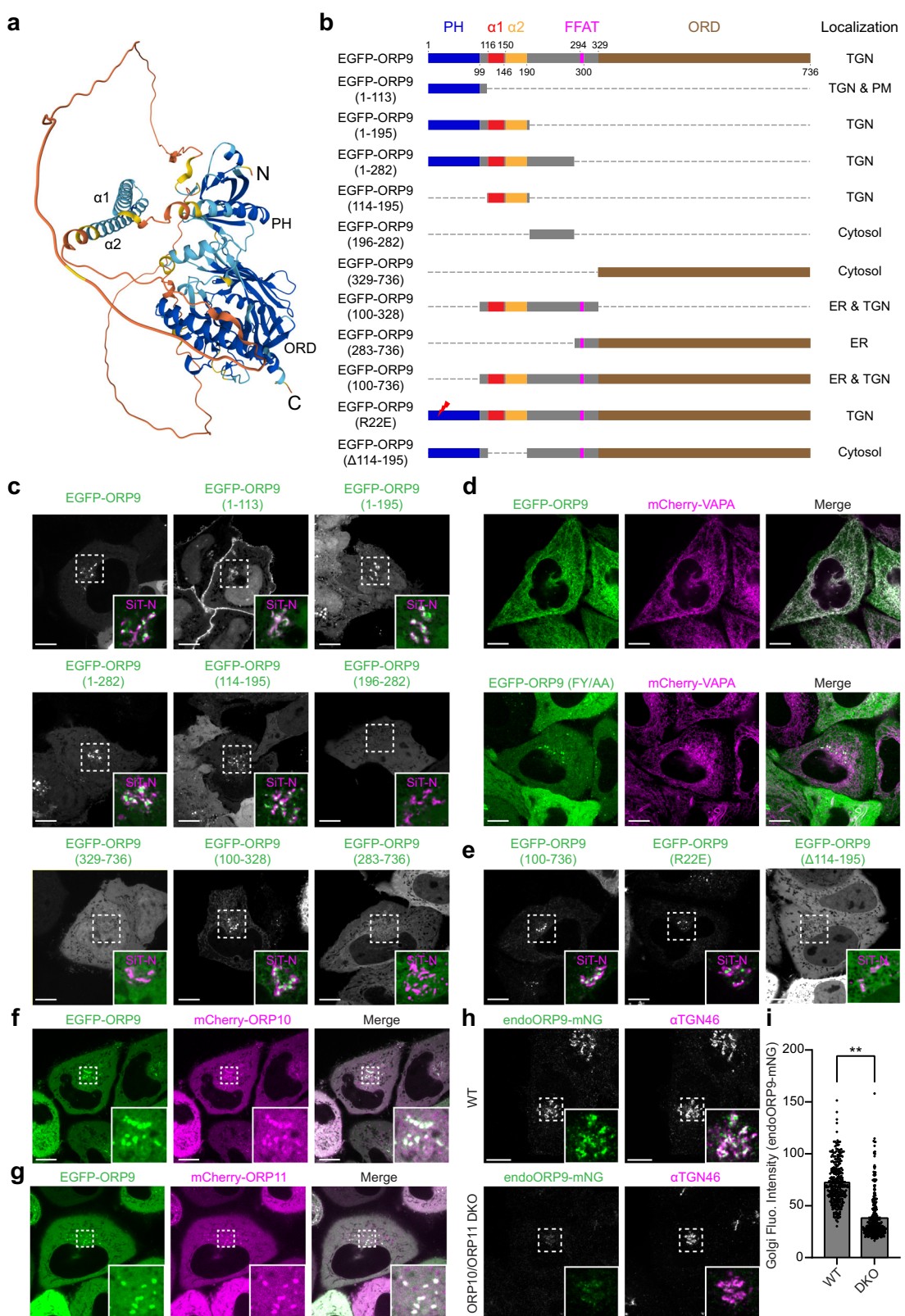

PM association of GRAM-H by ORP2 overexpression was not affected by co-expression of ORP9 (Supplementary Fig. 1c, d). These results suggest that ORP9 regulates the abundance of accessible cholesterol in the PM, possibly by selectively modulating the activity of OSBP.

To investigate the endogenous localization of ORP9 and OSBP, HeLa cells were fixed and labelled with antibodies against ORP9 or OSBP, together with antibodies against TGN46 (TGN marker) and GM130 (cis-Golgi marker) (Fig. 1c). Both ORP9 and OSBP localized to the TGN rather than the cis-Golgi (Fig. 1c). These results are consistent with the previously reported localization of these proteins to the TGN[73] or ER-TGN contacts[60]. In agreement with the reported localization of this protein to other organelles, such as endosomes and lysosomes[74,75], some ORP9 localized to regions outside the TGN (Fig. 1c).

**Fig. 2 | ORP9 localizes to the TGN via its N-terminal PH domain and tandem α-helices. a** The ribbon diagram of the modeled ORP9 (see Materials and Methods). PH, pleckstrin homology; ORD, OSBP-related domain; N, N-terminus; C, C-terminus. **b** Domain structure of ORP9 and of various versions of ORP9 analyzed in this study. **c** Confocal images of live HeLa cells expressing either EGFP-tagged ORP9 full-length (EGFP-ORP9) or one of the indicated versions of EGFP-ORP9 as shown in (**b**), together with a TGN marker, mCherry-tagged N-terminus of sialyltransferase (SiT-N-mCherry). Insets show at higher magnification the regions around the TGN as indicated by white dashed boxes (green: the indicated version of EGFP-ORP9; magenta: SiT-N-mCherry). Scale bars, 10 μm. **d** Confocal images of live HeLa cells expressing either EGFP-ORP9 or EGFP-ORP9 carrying the FY/AA mutation in its FFAT motif [EGFP-ORP9 (FY/AA)] together with mCherry-tagged VAPA (mCherry-VAPA). Note the recruitment of EGFP-ORP9, but not EGFP-ORP9 (FY/AA), to the ER labelled by mCherry-VAPA. Scale bars, 10 μm. **e** Confocal images of live HeLa cells expressing one of the indicated versions of EGFP-ORP9 as shown in (**b**), together with SiT-N-mCherry. Insets show at higher magnification the regions around the

TGN as indicated by white dashed boxes (green: the indicated versions of EGFP-ORP9; magenta: SiT-N-mCherry). Scale bars, 10 μm. **f, g** Confocal images of live HeLa cells expressing EGFP-ORP9 together with either mCherry-tagged ORP10 (mCherry-ORP10) or mCherry-tagged ORP11 (mCherry-ORP11). Note the extensive colocalization of EGFP-ORP9 with mCherry-ORP10 and mCherry-ORP11. Insets show at higher magnification the regions as indicated by white dashed boxes. Scale bars, 10 μm. **h** Confocal images of fixed WT HeLa cells and HeLa cells lacking ORP10 and ORP11 [ORP10/ORP11 double knockout (DKO)], in which mNeonGreen was tagged to the C-terminus of ORP9 (endoORP9-mNG). Cells were immunolabeled with antibodies against TGN46. Insets show at higher magnification the regions around the Golgi as indicated by white dashed boxes (green: endoORP9-mNG; magenta: TGN46). Scale bars, 10 μm. **i** Quantification of endoORP9-mNG fluorescence signals at the Golgi as shown in (**h**) [mean ± SEM, $n = 296$ cells (WT), $n = 325$ cells (DKO); data are pooled from two independent experiments; two-tailed unpaired Student's $t$-test, **$P < 0.0001$]. Source data are provided as a Source Data file.

---

Antibodies against ORP9 and OSBP were both raised in rabbits, making it difficult to assess the localization of ORP9 and OSBP within the same cell by immunolabelling. To assess their localization within the same cell, HeLa cells carrying endogenously tagged ORP9 and OSBP were generated. We engineered a mNeonGreen tag for the C-terminus of ORP9 (endoORP9-mNG) and a mScarlet-I tag for the N-terminus of OSBP (mSc-endoOSBP) via the ORANGE genomic editing method[76] (Supplementary Fig. 1e). Expression of endoORP9-mNG and mSc-endoOSBP was confirmed by western blot analysis of cell lysates with antibodies against ORP9, mNeonGreen, OSBP, and mCherry (which also recognizes mScarlet-I) (Supplementary Fig. 1f). Consistent with results obtained with antibodies against ORP9 and OSBP, endoORP9-mNG and mSc-endoOSBP localized to the TGN (labelled by anti-TGN46 antibodies). Interestingly, their localization was seemingly non-overlapping (Fig. 1d). To obtain more insights into their TGN localization, cells expressing endoORP9-mNG and mSc-endoOSBP were fixed and observed under SDC-structured illumination (SIM) microscopy, and the localization of endoORP9-mNG and mSc-endoOSBP at the TGN was analyzed in reconstituted 3D volumetric images (Fig. 1e). EndoORP9-mNG and mSc-endoOSBP were both found on the TGN as distinct puncta, but they localized to distinct regions of the TGN (Fig. 1e), suggesting they are recruited to the TGN via different mechanisms.

These endogenously tagged HeLa cells also allowed us to assess the localization of ORP9 and OSBP in live. Localization of OSBP to the TGN depends on the presence of PI4P in TGN membranes[60]. To examine whether the TGN localization of ORP9 also depended on PI4P, cells expressing endoORP9-mNG and mSc-endoOSBP, together with iRFP-tagged P4M (iRFP-P4M), a PI4P marker, were treated with PIK93 to inhibit PI4KIIIβ, a Golgi-localized PI4 kinase that produces the bulk of PI4P in this organelle[62,77]. While mSc-endoOSBP dissociated from the TGN upon Golgi PI4P depletion, endoORP9-mNG remained tightly associated with the TGN (Fig. 1f, g and Supplementary Movie 1). Taken together, these results suggest that ORP9 and OSBP localize to the TGN via distinct mechanisms and that ORP9 may modulate OSBP-mediated cholesterol transport at the TGN.

## ORP9 localizes to the TGN via its N-terminal PH domain and tandem α-helices

To determine the structural elements of ORP9 responsible for its localization to the TGN, we obtained a structural model of ORP9 using Alphafold[78]. ORP9 consists of an N-terminal PH domain, a FFAT motif, and a C-terminal ORD (Fig. 2a, b). In addition, this model predicted the presence of tandem α-helices (α1 and α2) and a disordered region following the PH domain (Fig. 2a). Based on the predicted structure, we generated versions of ORP9 in which different domains/motifs were deleted (Fig. 2b). These N-terminal EGFP-tagged proteins were then co-expressed with mCherry-tagged TGN marker (N-terminal 45 amino

acids of human sialyltransferase), SiT-N-mCherry, in WT HeLa cells and their association with the TGN was determined using SDC microscopy. Full length ORP9 (EGFP-ORP9) localized throughout the cytoplasm with some distinct accumulation in the TGN, labelled by SiT-N-mCherry, mimicking the localization of endogenous ORP9 to the TGN (Fig. 2c and Supplementary Fig. 2a; compare with Fig. 1c, d).

The PH domains of ORPs often bind anionic lipids, including phosphoinositides, and play a role in targeting ORPs to specific cellular compartments[58,59]. The PH domain of ORP9 binds PI4P, which is particularly abundant in the TGN membrane and the PM[73]. Accordingly, the PH domain of ORP9 [EGFP-ORP9 (1–113)] localized uniformly both to the TGN and to the PM (Fig. 2c and Supplementary Fig. 2a). Remarkably, addition of the predicted tandem α-helices to the PH domain [EGFP-ORP9 (1–195)] resulted in recruitment of the PH domain exclusively to the TGN as distinct puncta, similar to EGFP-ORP9 (Fig. 2c and Supplementary Fig. 2a). Further addition of the disordered region [EGFP-ORP9 (1–282)] did not affect localization of the fragment to the TGN (Fig. 2c and Supplementary Fig. 2a). Strikingly, the tandem α-helices alone [EGFP-ORP9 (114–195)] localized to the TGN [albeit less prominently compared to EGFP-ORP9 or EGFP-ORP9 (1–195)] (Fig. 2c and Supplementary Fig. 2a), whereas the disordered region alone [EGFP-ORP9 (196–282)] and the ORD domain alone [EGFP-ORP9 (329–736)] showed diffuse cytosolic localization with no accumulation in the TGN (Fig. 2c and Supplementary Fig. 2a). These results indicate that the tandem α-helices alone are sufficient to localize ORP9 to the TGN.

The region containing the PH domain, tandem α-helices, and the FFAT motif [EGFP-ORP9 (100–328)] localized both to the TGN and to the ER, as indicated by co-expressed BFP-tagged Sec61β (BFP-Sec61β) (Fig. 2c and Supplementary Fig. 2b, c). In addition, the region containing the FFAT motif and the ORD [EGFP-ORP9 (283–736)] localized to the ER with no accumulation in the TGN (Fig. 2c). These results indicate the role of the FFAT motif of ORP9 in its recruitment to the ER. Indeed, ER localization of EGFP-ORP9 (283–736) was much enhanced when it was co-expressed with mCherry-tagged VAPA (mCherry-VAPA) or mCherry-tagged VAPB (mCherry-VAPB), ER proteins that anchor FFAT motif-containing proteins to the ER (Supplementary Fig. 2b, c). Accordingly, co-expression of EGFP-ORP9 and mCherry-VAPA resulted in strong recruitment of EGFP-ORP9 to the ER (Fig. 2d and Supplementary Fig. 2d). Importantly, mutation of two amino acids within the ORP9 FFAT motif (EGFP-ORP9 FY/AA) abolished the recruitment of EGFP-ORP9 to the ER by mCherry-VAPA (Fig. 2d and Supplementary Fig. 2d). These results confirm that the ORP9 FFAT motif is essential for the recruitment of ORP9 to the ER via its interaction with ER-localized VAP proteins.

Finally, we examined whether the PH domain of ORP9 is necessary for its localization to the TGN. Remarkably, both ORP9 lacking the PH domain [EGFP-ORP9 (100–736)] and ORP9 carrying a

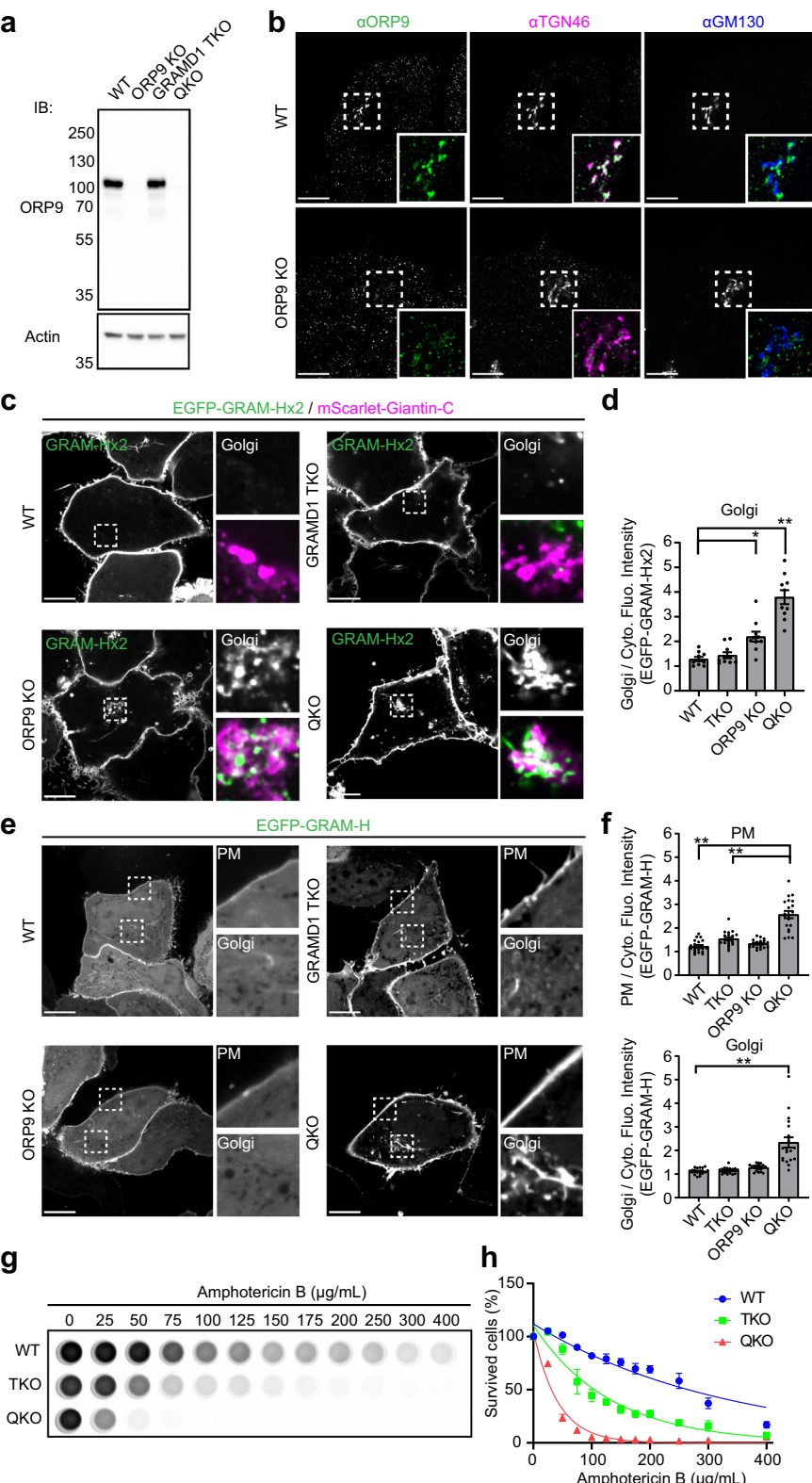

PI4P-binding defective PH domain [EGFP-ORP9 (R22E)] (change of arginine 22 to glutamic acid)[73] localized to the TGN (Fig. 2e and Supplementary Fig. 2e). However, we note that such localization was less prominent compared to the full-length ORP9 (EGFP-ORP9) (Fig. 2e). Furthermore, the recruitment of the tandem α-helices of ORP9 to the TGN was much more enhanced with the addition of the PH domain [compare EGFP-ORP9 (114–195) and EGFP-ORP9 (1–195)] (Fig. 2c and Supplementary Fig. 2a). Thus, although the PH domain

itself is not necessary for the localization of ORP9 to the TGN per se, it acts together with the tandem α-helices to promote the localization of ORP9 to the TGN. Interestingly, EGFP-ORP9 (100–736) showed additional localization to the ER, whereas EGFP-ORP9 (R22E) did not (Fig. 2e), suggesting that the PH domain of ORP9 may possess a property (that is independent from its ability to bind to PI4P) to suppress the interaction of its FFAT motif to ER-anchored VAP proteins.

**Fig. 3 | Depletion of ORP9 causes accumulation of accessible cholesterol in the Golgi, which is further enhanced by the additional depletion of GRAMD1s with major accumulation in the PM. a** Lysates of WT HeLa cells, HeLa cells lacking ORP9 (ORP9 KO), GRAMD1 TKO HeLa cells, and HeLa cells lacking ORP9 and GRAMD1s (QKO) were processed by SDS–PAGE and immunoblotting (IB) with anti-ORP9 and anti-actin antibodies. **b** Confocal images of fixed WT and ORP9 KO HeLa cells that were immunolabeled with antibodies against ORP9, TGN46, and GM130. Scale bars, 10 μm. Note that anti-ORP9 signals in ORP9 KO cells are non-specific. **c** Confocal images of live WT, GRAMD1 TKO, ORP9 KO, and QKO HeLa cells expressing EGFP-GRAM-Hx2 (accessible cholesterol biosensor) and mScarlet-tagged C-terminus of Giantin (mScarlet-Giantin-C). Insets show at higher magnification the regions around the Golgi as indicated by white dashed boxes (in the bottom insets, green: EGFP-GRAM-Hx2; magenta: mScarlet-Giantin-C). Scale bars, 10 μm. **d** Quantification of the ratio of Golgi signals to the cytosolic signals of EGFP-GRAM-Hx2 as shown in (**c**) [mean ± SEM, *n* = 10 cells for each condition; data are

pooled from one experiment; Dunnett's multiple comparisons test, *\*P* = 0.0047 (WT vs ORP9 KO), \*\**P* < 0.0001 (WT vs QKO)]. **e** Confocal images of live WT, GRAMD1 TKO, ORP9 KO, and QKO HeLa cells expressing EGFP-GRAM-H (accessible cholesterol biosensor). Insets show at higher magnification the regions around the plasma membrane (PM) and the Golgi as indicated by white dashed boxes. Scale bars, 10 μm. **f** Quantification of the ratio of PM signals (top) and Golgi signals (bottom) to the cytosolic signals of EGFP-GRAM-H as shown in (**e**) (mean ± SEM, *n* = 20 cells for each condition; data are pooled from two independent experiments; Dunnett's multiple comparisons test, \*\**P* < 0.0001). **g** Amphotericin B resistance of WT, GRAMD1 TKO, and QKO HeLa cells. Cells were treated with indicated concentration of Amphotericin B for 20 min at 37 °C. After overnight recovery in culture media, cell viability was measured by detecting ATP present in each well via luminescence (see Methods). **h** Quantification of cell viability with increasing amount of Amphotericin B, as shown in (**g**) (mean ± SEM, *n* = 4 independent experiments for each condition). Source data are provided as a Source Data file.

## Interaction of ORP9 with ORP10 and ORP11 via its tandem α-helices is important for its localization to the TGN

EGFP-ORP9 lacking the tandem α-helices (EGFP-ORP9 Δ114–195) showed diffuse cytosolic localization with no accumulation in the TGN, revealing the critical importance of the tandem α-helices for ORP9 to localize to the TGN (Fig. 2e and Supplementary Fig. 2e). To identify factors that interact with the α-helices, we expressed EGFP, EGFP-ORP9 (1–113) (the PH domain alone), or EGFP-ORP9 (1–195) (the PH domain with the tandem α-helices) in HEK293T cells and performed immunoprecipitation using cell lysates and nanobodies against EGFP. SDS-PAGE analysis of the anti-EGFP immunoprecipitates revealed bands that appeared only in the EGFP-ORP9 (1–195) immunoprecipitates (Supplementary Fig. 2f). Analysis of these bands by mass spectrometry revealed that ORP11, ORP10, and ORP9 were the top interactors with the α-helices of ORP9 (Supplementary Fig. 2f, g). To confirm their interactions, EGFP-ORP9 was co-expressed together with mRuby-tagged ORP9, ORP10 or ORP11 in HeLa cells. Interactions were assessed via anti-EGFP immunoprecipitation followed by western blotting. EGFP-ORP9 interacted strongly with mRuby-ORP10 as well as mRuby-ORP11, and weakly with mRuby-ORP9 (Supplementary Fig. 2h). These interactions were further validated by imaging live cells via SDC microscopy. When EGFP-ORP9 was co-expressed in HeLa cells with mCherry-tagged ORP10 (mCherry-ORP10) or mCherry-tagged ORP11 (mCherry-ORP11), EGFP-ORP9 co-localized extensively with these proteins at the TGN-like structures around the nucleus (Fig. 2f, g and Supplementary Fig. 2i). These results are consistent with previous findings that ORP9 interacts with ORP10 and ORP11[74,75,79]. Our results additionally reveal the critical importance of ORP9's tandem α-helices in these interactions.

Finally, we examined whether depletion of ORP10 and ORP11 affects the localization of ORP9 to the TGN. To this end, ORP10 and ORP11 were simultaneously depleted either via the CRISPR/Cas9-mediated knockout (KO) approach or via the RNAi-mediated knockdown (KD) approach in HeLa cells carrying endoORP9-mNG. Double depletion of ORP10 and ORP11 was confirmed by western blotting (Supplementary Fig. 2j–m). There was no major impact on the expression of endoORP9-mNG (Supplementary Fig. 2j, l). Importantly, the localization of endoORP9-mNG to the TGN was significantly reduced, although such reduction was more prominent in the double KO (ORP10/ORP11 DKO) cells compared to double KD (ORP10/ORP11 DKD) cells (Fig. 2h, i and Supplementary Fig. 2n, o). Thus, the complex formation of ORP9, ORP10, and ORP11 is important for the recruitment of ORP9 to the TGN.

Taken together, ORP9 is selectively recruited to ER-TGN contacts via binding of its FFAT motif with ER-anchored VAP proteins and binding of its N-terminal PH domain and tandem α-helices, which interact with ORP10 and ORP11, to the TGN.

## Depletion of ORP9 causes accumulation of accessible cholesterol in the Golgi, which is further enhanced by the additional depletion of GRAMD1s with major accumulation in the PM

Based on our localization analysis, we hypothesized that ORP9 helps to maintain cellular cholesterol distribution by facilitating lipid transport/exchange at ER-TGN contact sites. To investigate the potential roles of ORP9 in this process, we used CRISPR/Cas9 system to disrupt ORP9 function in WT and GRAMD1 TKO HeLa cells. Absence of ORP9 was confirmed via western blot analysis (Fig. 3a) and immunostaining of fixed cells (Fig. 3b). Subsequent experiments were performed using a cell clone lacking ORP9 (ORP9 KO) and a cell clone lacking the GRAMD1s and ORP9 (QKO).

To assess the distribution of cholesterol at the Golgi in ORP9 KO and QKO cells, EGFP-tagged tandem GRAM-H (EGFP-GRAM-Hx2), was expressed in these cells and its distribution was determined using SDC microscopy in live cells. EGFP-GRAM-Hx2 was strongly bound to the PM in WT HeLa cells at steady state, consistent with the enhanced sensitivity of this biosensor compared to EGFP-GRAM-H (Fig. 3c). There was no detectable accumulation of accessible cholesterol around the Golgi labelled by mScarlet-tagged Giantin-C (mScarlet-Giantin-C) in WT and GRAMD1 TKO cells (Fig. 3c, d). Remarkably, significant accumulation of accessible cholesterol around the Golgi was observed in ORP9 KO cells, which was further enhanced in QKO cells (Fig. 3c, d).

EGFP-GRAM-Hx2 was strongly bound to the PM in WT HeLa cells (Fig. 3c), making it difficult to assess potential changes in the levels of accessible PM cholesterol. To examine the impact of the depletion of ORP9 on PM cholesterol, EGFP-GRAM-H was expressed in live cells. The levels of accessible PM cholesterol, as assessed by EGFP-GRAM-H, was not altered in ORP9 KO cells compared to WT HeLa cells (Fig. 3e, f). By contrast, levels of accessible PM cholesterol were elevated in GRAMD1 TKO cells and further enhanced in QKO cells (Fig. 3e, f). We previously showed that GRAMD1 TKO cells are highly susceptible to treatment with the polyene antibiotic, Amphotericin B, compared to WT cells[45]. Amphotericin B causes cell death by binding to and sequestering PM sterols, leading to several abnormalities, including the formation of non-selective ion pores at the PM[71,80]. QKO cells were even more efficiently killed by Amphotericin B compared to TKO cells (Fig. 3g, h). This is consistent with the enhanced accumulation of accessible PM cholesterol in QKO cells compared to TKO cells.

A large amount of accessible cholesterol, as assessed by EGFP-GRAM-H, accumulated around the Golgi of QKO cells (Fig. 3e, f). This is consistent with the accumulation of EGFP-GRAM-Hx2 around the Golgi in these cells (Fig. 3c, d). In QKO cells, co-expression of the GRAM-H biosensor with fluorescently tagged Golgi protein, namely Giantin-C (cis/medial-Golgi), GOLPH3 (trans-Golgi), VAMP4 (TGN), or TPST2 (trans-Golgi)[81,82] revealed close association of GRAM-H with these Golgi proteins (Supplementary Fig. 3a, b). While ORP9 KO cells did not show

major changes in the distribution of EGFP-GRAM-H compared to QKO cells, some increases in the association of EGFP-GRAM-H around the Golgi were observed (although such increase was not statistically significant compared to WT HeLa cells) (Fig. 3e, f). This is likely due to the lower sensitivity of EGFP-GRAM-H compared to EGFP-GRAM-Hx2.

We then used another accessible cholesterol biosensor to further assess the distribution of accessible cholesterol in these cells. This sensor is a fusion of the mutated domain 4 (D4) of Perfringolysin O with mCherry (mCherry-D4H)[83–85]. To examine the levels of accessible PM cholesterol, cells were incubated with purified recombinant mCherry-D4H protein and its interaction with the PM was assessed using SDC microscopy. Binding of purified mCherry-D4H proteins to the PM was enhanced in GRAMD1 TKO cells compared to WT HeLa cells, as we previously reported[45,53]. This binding was further enhanced in QKO cells (Supplementary Fig. 3c, d), consistent with the increase in accessible PM cholesterol in QKO cells. We also expressed EGFP-tagged D4H (EGFP-D4H) in cytosol and examined accessible cholesterol distribution in live cells. Binding of EGFP-D4H to the PM was significantly enhanced in both GRAMD1 TKO cells and QKO cells compared to WT HeLa cells (Supplementary Fig. 3e, f). In QKO cells, EGFP-D4H was additionally accumulated around the Golgi labelled by mScarlet-Giantin-C (Supplementary Fig. 3e, f), confirming the accumulation of accessible cholesterol around the Golgi in these cells as observed by GRAM-H.

Collectively, these results show that ORP9 is important for proper distribution of cholesterol in the Golgi. Importantly, accumulation of accessible cholesterol in ORP9 KO cells was further enhanced by the additional depletion of GRAMD1s with major accumulation in the PM (i.e., QKO cells), suggesting that ORP9 and GRAMD1s functionally cooperate. Unless otherwise noted, QKO cells were used in the rest of the study to dissect the role of ORP9 and its interplay with GRAMD1s in the regulation of cellular cholesterol distribution.

## The ORD of ORP9 must extract PI4P to maintain proper distribution of cellular cholesterol

While previous reports have suggested that ORP9 can extract both PI4P and cholesterol from artificial membranes in vitro[73,86], it remains unclear whether the ORD of ORP9 can extract these lipids from membranes of live cells. To determine whether ORP9 can extract PI4P and/or cholesterol from cellular membranes, rapamycin-induced dimerization of the FK506-binding protein (FKBP) and the FKBP-rapamycin-binding domain (FRB) were used to artificially recruit the ER-anchored ORD of ORP9 to the PM in QKO cells, where accessible cholesterol accumulates (Fig. 4a). QKO cells stably expressing EGFP-GRAM-H and the ER-anchored ORD of ORP9 fused with mCherry-tagged FKBP (ER-mCherry-FKBP-ORP9) were generated by lentivirus transduction. We then transiently expressed the PM-targeted FRB module fused with tagBFP (PM-FRB-tagBFP) and iRFP-P4M (PI4P biosensor) in these cells. This allowed us to simultaneously monitor the pools of accessible cholesterol and PI4P in the PM during rapamycin-induced recruitment of the ER-anchored ORD of ORP9 to the PM via total internal reflection fluorescence (TIRF) microscopy.

Rapamycin-dependent recruitment of ER-mCherry-FKBP-ORP9 to the PM induced a rapid reduction of PM PI4P, as assessed by iRFP-P4M. This occurred within 10 min of rapamycin treatment. By contrast, accessible cholesterol, as assessed by EGFP-GRAM-H, was not affected (Fig. 4b). The ORDs of all ORPs, including that of ORP9, contain the amino acid residues "EQVSHHPP", which function as a common PI4P-binding motif[58,59]. Mutations in this motif disrupt the PI4P-harboring property of the ORD. Accordingly, a mutant version of ER-mCherry-FKBP-ORP9 that includes mutations in this motif [ER-mCherry-FKBP-ORP9 (HH/AA)] (change of histidine 501 and 502 to alanines), failed to reduce PM PI4P (Fig. 4c). This demonstrates a specific role of the ORD of ORP9 in extracting PI4P, but not cholesterol, from cellular membranes. As an additional control, we used the same system except that

the ORD of ER-mCherry-FKBP-ORP9 was replaced with the ORD of OSBP (ER-mCherry-FKBP-OSBP) (Supplementary Fig. 4a). Remarkably, acute recruitment of ER-mCherry-FKBP-OSBP to the PM resulted in simultaneous reduction of PI4P and an increase in accessible cholesterol in the PM, consistent with the previously reported function of the ORD of OSBP in mediating PI4P/cholesterol counter exchange at various membrane contact sites[60,61,64], validating our TIRF-based assays (Supplementary Fig. 4b).

When the ORD of ORP9 was artificially recruited to the PM it extracted PI4P, but not cholesterol, from the PM. We examined the importance of this functionality in rescue experiments using QKO cells. Re-expression of mCherry-ORP9 in QKO cells reduced levels of accessible cholesterol, as assessed by EGFP-GRAM-H and EGFP-D4H, in both the PM and Golgi compared to QKO cells expressing an mCherry control. This supports a role for ORP9 in controlling cholesterol distribution in cells by regulating levels of PI4P (Fig. 4d, e and Supplementary Fig. 4c, d). Expression of mCherry-ORP9 (HH/AA), whose ORD cannot extract PI4P, in QKO cells failed to reduce levels of accessible cholesterol in the Golgi and PM (Fig. 4d, e and Supplementary Fig. 4c, d). Compared to mCherry-ORP9, more mCherry-ORP9 (HH/AA) accumulated at the perinuclear region around the Golgi, suggesting that PI4P extraction via ORD might be a critical step for dissociation of ORP9 from the TGN (Fig. 4d and Supplementary Fig. 4c, d). Recently, ORP9 was reported to transport PS in cells[75]. To address whether such property of ORP9 might play a role in regulating cellular cholesterol distribution, a mutant version of mCherry-ORP9 that cannot transport PS [mCherry-ORP9 (AAA)] was expressed in QKO cells. Expression of mCherry-ORP9 (AAA) restored normal cholesterol distribution in QKO cells (Fig. 4d, e), suggesting that PS transport property of ORP9 is not essential for regulating this process. These results show that the ability of ORP9 to extract PI4P is critical for maintaining cellular cholesterol distribution.

## ORP9 extraction of PI4P from the TGN helps maintain levels of accessible cholesterol in the Golgi

Our results suggest that ORP9 controls the abundance of PI4P at the TGN via its ORD, which can extract PI4P. As ORP9 localizes to ER-TGN contacts (Fig. 2), ORP9 may extract PI4P from the TGN and transport it to the ER via its ORD. To assess the impact of ORP9 depletion on the distribution of PI4P in cellular membranes, the PI4P marker, iRFP-P4M, was expressed together with a TGN marker, SiT-N-mCherry in WT, ORP9 KO, and QKO HeLa cells. More PI4P accumulated at the TGN in ORP9 KO cells and in QKO cells than in WT cells (Fig. 5a, b). Co-expression of iRFP-P4M with an EGFP-tagged Golgi protein, either GOLPH3 (trans-Golgi), VAMP4 (TGN), or TPST2 (trans-Golgi), in QKO cells showed co-localization of iRFP-P4M with these Golgi proteins, confirming an accumulation of PI4P at the TGN (Supplementary Fig. 5a, b). Re-expression of mCherry-ORP9, but not mCherry-ORP9 (HH/AA) (whose ORD cannot extract PI4P), in ORP9 KO cells rescued the Golgi PI4P to normal levels (Supplementary Fig. 5c, d). Again, more mCherry-ORP9 (HH/AA) accumulated at the perinuclear region around the Golgi (Supplementary Fig. 5c). These results demonstrate an essential role for ORP9 in sensing and extracting PI4P and preventing this lipid from building up at the TGN. In the absence of ORP9, PI4P aberrantly accumulates in the TGN.

We next asked whether ORP9 acts directly at ER-TGN contact sites using rapamycin-induced recruitment of the ER-anchored ORD of ORP9 to the TGN. For this assay, QKO cells stably expressing EGFP-GRAM-H (accessible cholesterol biosensor) and ER-mCherry-FKBP-ORP9 were transiently transfected with a TGN-targeted FRB module fused with tagBFP (tagBFP-TGN46-FRB) and iRFP-P4M (PI4P biosensor). Levels of PI4P and cholesterol at the Golgi were simultaneously monitored via SDC microscopy during acute rapamycin treatment, which recruits ER-mCherry-FKBP-ORP9 to the TGN (Fig. 5c, d and Supplementary Movie 2). ER-mCherry-FKBP-ORP9 was

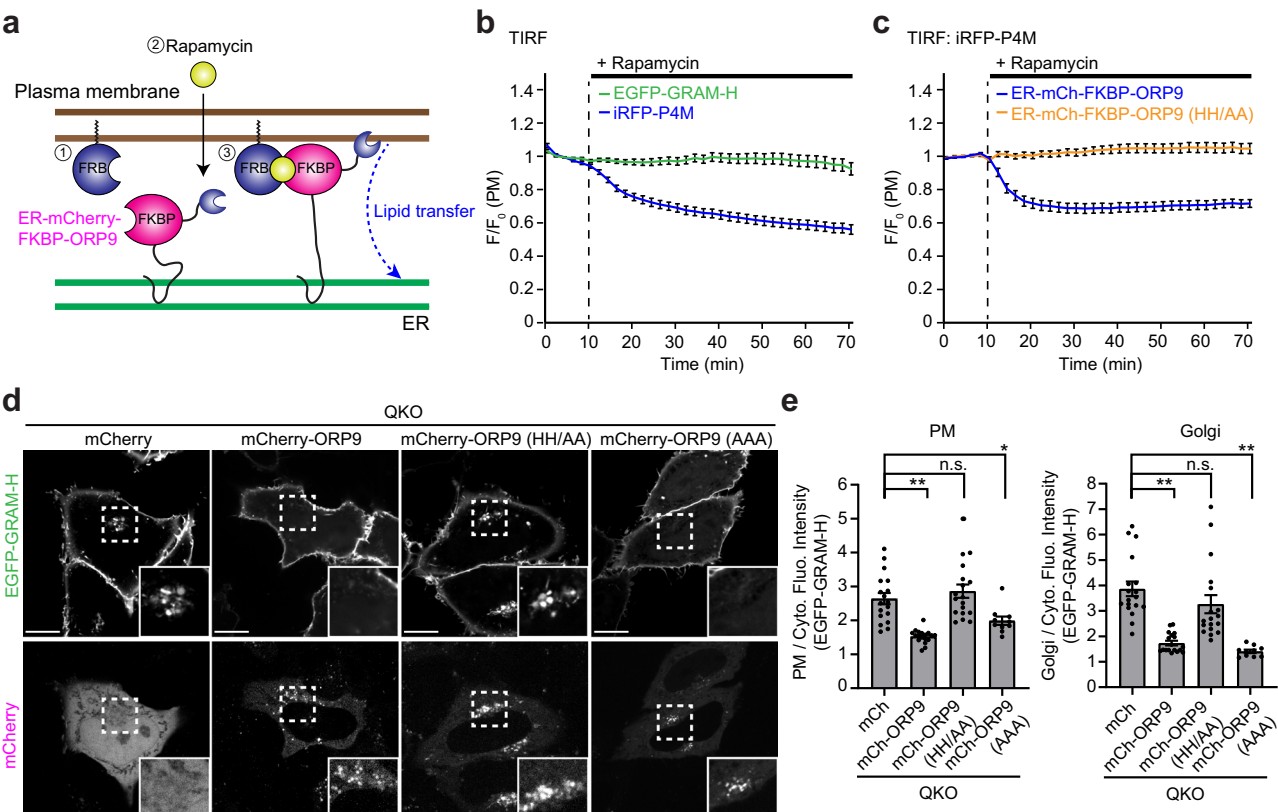

**Fig. 4 | The ORD of ORP9 must extract PI4P to maintain proper distribution of cellular cholesterol. a** Schematic representation of the rapamycin-induced recruitment strategy used for the recruitment of ER-anchored ORD of ORP9 (ER-mCherry-FKBP-ORP9) to the PM. ER-mCherry-FKBP-ORP9 was rapidly recruited to the PM by rapamycin-induced dimerization of FRB and FKBP. ER-mCherry-FKBP-ORP9 was expressed in QKO HeLa cells together with a tagBFP-tagged FRB module that is targeted to the PM (PM-FRB-tagBFP). **b** Time course of normalized signals of EGFP-GRAM-H (accessible cholesterol biosensor) and iRFP-P4M (PI4P biosensor) in response to rapamycin, as assessed by TIRF microscopy of QKO HeLa cells expressing ER-mCherry-FKBP-ORP9 and PM-FRB-tagBFP together with EGFP-GRAM-H and iRFP-P4M. Rapamycin addition (200 nM) is indicated (mean ± SEM, $n = 29$ cells for each condition; data are pooled from three independent experiments). **c** Time course of normalized iRFP-P4M signals in response to rapamycin, as assessed by TIRF microscopy of QKO HeLa cells expressing either ER-mCherry-FKBP-ORP9 or ER-mCherry-FKBP-ORP9 carrying PI4P binding-deficient ORD [ER-mCherry-FKBP-ORP9 (HH/AA)] together with PM-FRB-tagBFP and iRFP-P4M.

Rapamycin addition (200 nM) is indicated [mean ± SEM, $n = 24$ cells (WT), $n = 30$ (HH/AA); data are pooled from two independent experiments]. **d** Confocal images of live QKO HeLa cells expressing EGFP-GRAM-H together with either mCherry control, mCherry-ORP9, mCherry-ORP9 carrying PI4P binding-deficient ORD [mCherry-ORP9 (HH/AA)], or mCherry-ORP9 carrying phosphatidylserine binding-deficient ORD [mCherry-ORP9 (AAA)]. Insets show at higher magnification the regions around the Golgi as indicated by white dashed boxes. Scale bars, 10 μm. **e** Quantification of the ratio of PM signals (left) and Golgi signals (right) to the cytosolic signals of EGFP-GRAM-H, as shown in (**d**) {mean ± SEM, $n = 20$ cells for mCherry, mCherry-ORP9, and mCherry-ORP9 (HH/AA); $n = 10$ cells for mCherry-ORP9 (AAA) (PM), $n = 18$ cells for mCherry, mCherry-ORP9 and mCherry-ORP9 (HH/AA); $n = 10$ cells for mCherry-ORP9 (AAA) (Golgi); data are pooled from two independent experiments for mCherry, mCherry-ORP9, and mCherry-ORP9 (HH/AA); one experiment for mCherry-ORP9 (AAA); Dunnett's multiple comparisons test, **$P < 0.0001$, *$P = 0.0251$ [PM: mCh vs mCh-ORP9 (AAA)]. n.s. denotes not significant}. Source data are provided as a Source Data file.

recruited to the TGN within ~10 min of rapamycin treatment (Fig. 5c, d and Supplementary Movie 2). Strikingly, levels of both PI4P and accessible cholesterol at the Golgi, as assessed by EGFP-GRAM-H and iRFP-P4M respectively, were significantly reduced following rapamycin treatment (Fig. 5c, d and Supplementary Movie 2).

To further examine the role of PI4P in regulating levels of accessible cholesterol at the Golgi, we used the same system except ER-mCherry-FKBP-ORD9 was replaced with the PI4P phosphatase domain of Sac1 fused with mCherry-tagged FKBP (Sac1ΔTM-FKBP-mCherry)[87]. QKO cells transiently expressing Sac1ΔTM-FKBP-mCherry and tagBFP-TGN46-FRB, together with EGFP-GRAM-H and iRFP-P4M, were treated with rapamycin. Sac1ΔTM-FKBP-mCherry was recruited to the TGN within ~4 min of rapamycin treatment, resulting in rapid removal of PI4P from the TGN. Remarkably, this was accompanied by a significant reduction in the levels of accessible cholesterol at the Golgi (Fig. 5e, f and Supplementary Movie 3). Notably, reduction of PI4P preceded the reduction of accessible cholesterol (Fig. 5f and Supplementary Movie 3). Furthermore, inhibition of PI4KIIIβ by the treatment of PIK93 for 1 hr significantly reduced the levels of accessible cholesterol at the

Golgi and PM in QKO cells (Supplementary Fig. 5e, f). These results show that keeping levels of PI4P at the TGN below a certain threshold prevents overaccumulation of accessible cholesterol in the Golgi and PM.

As the ORD of ORP9 extracts PI4P but not cholesterol (Fig. 4b, c), our data suggest that ORP9 extracts PI4P from the TGN membrane at ER-TGN contact sites and contributes to the maintenance of the Golgi PI4P pool. As Golgi PI4P is essential for OSBP-mediated PI4P/cholesterol exchange at ER-TGN contacts[63], this ORP9 function may contribute to the suppression of OSBP-mediated cholesterol transport to the Golgi, thereby keeping levels of accessible cholesterol in the Golgi below a certain threshold.

### Accumulation of TGN PI4P in ORP9 KO and QKO cells is associated with hyperactivation of OSBP-mediated cholesterol transport to the Golgi

As ORP9 plays a critical role in maintaining the Golgi PI4P pool (Fig. 5a, b), the association of OSBP with the TGN, which itself is dependent on PI4P, may be regulated by ORP9. To investigate the

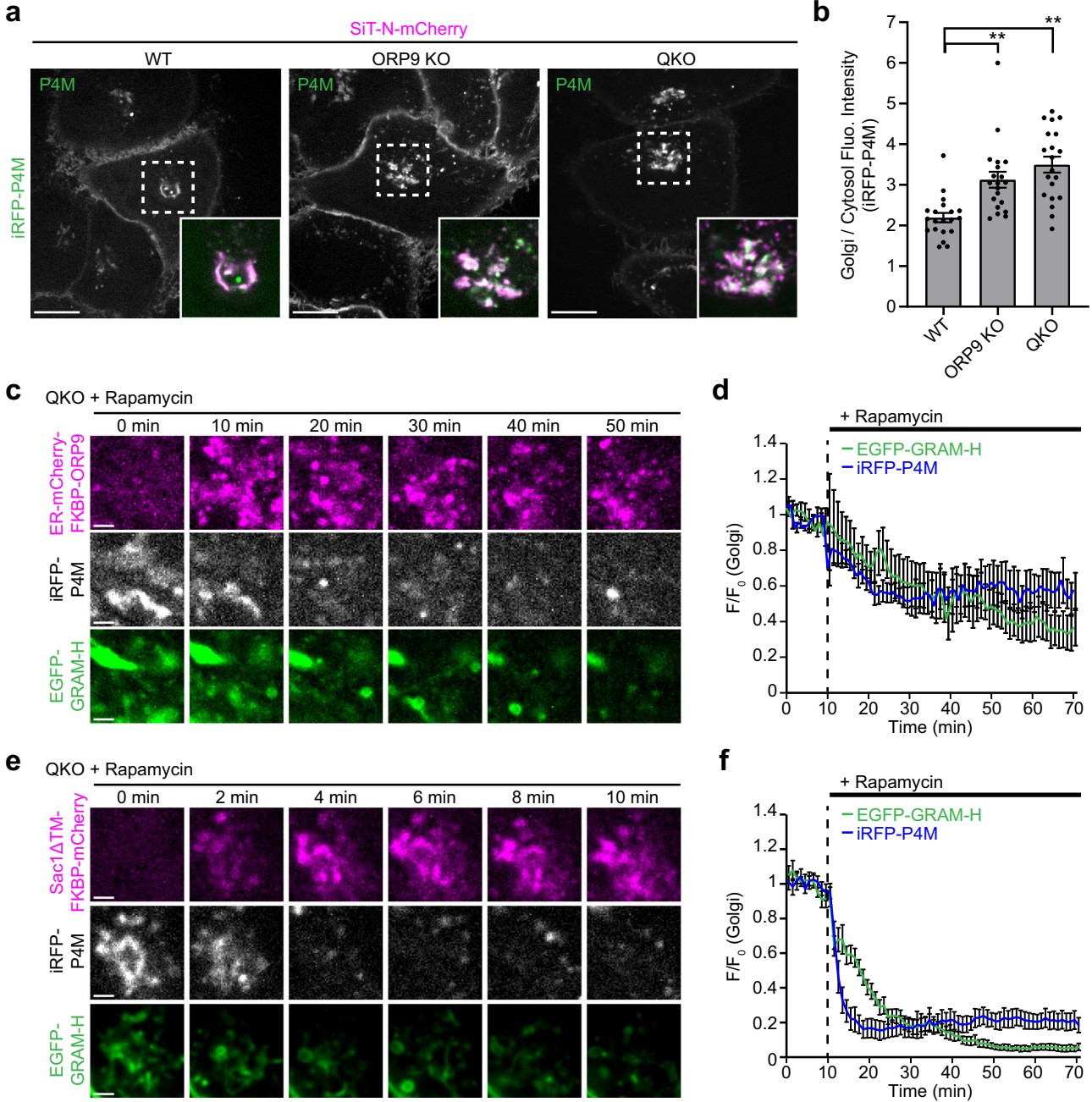

**Fig. 5 | ORP9 extraction of PI4P from the TGN helps maintain levels of accessible cholesterol in the Golgi. a** Confocal images of live WT, ORP9 KO, QKO HeLa cells expressing iRFP-P4M (PI4P biosensor), together with a TGN marker, SiT-N-mCherry. Insets show at higher magnification the regions around the TGN as indicated by white dashed boxes (green: iRFP-P4M; magenta: SiT-N-mCherry). Scale bars, 10 μm. **b** Quantification of the ratio of Golgi signals to the cytosolic signals of iRFP-P4M, as shown in (**a**) [mean ± SEM, $n = 20$ cells for each condition; data are pooled from two independent experiments; Dunnett's multiple comparisons test, **$P = 0.0007$ (WT vs ORP9 KO), **$P < 0.0001$ (WT vs QKO)]. **c** Confocal images of the regions around the Golgi of a live QKO HeLa cell expressing ER-mCherry-FKBP-ORP9 and tagBFP-TGN38-FRB together with EGFP-GRAM-H (accessible cholesterol biosensor) and iRFP-P4M that were treated with rapamycin (200 nM) for the indicated minutes. Scale bars, 2 μm. **d** Time course of normalized signals of EGFP-GRAM-H and iRFP-P4M in response to rapamycin, as assessed by confocal microscopy as shown in (**c**) (mean ± SEM, $n = 14$ cells; data are pooled from three independent experiments.). **e** Confocal images of the regions around the Golgi of a live QKO HeLa cell expressing mCherry-tagged Sac1ΔTM (PI4P phosphatase domain of Sac1) fused with FKBP module [Sac1ΔTM-FKBP-mCherry] and tagBFP-TGN38-FRB together with EGFP-GRAM-H and iRFP-P4M treated with rapamycin (200 nM) for the indicated minutes. Scale bars, 2 μm. **f** Time course of normalized signals of EGFP-GRAM-H and iRFP-P4M in response to rapamycin, as assessed by confocal as shown in (**e**) (mean ± SEM, $n = 15$ cells; data are pooled from two independent experiments). Source data are provided as a Source Data file.

impact of ORP9 depletion on OSBP localization, WT, ORP9 KO, and QKO HeLa cells were fixed and stained with antibodies against OSBP and TGN46. In agreement with increased PI4P at the TGN in cells lacking ORP9 (Fig. 5a, b), OSBP accumulated at the TGN, labelled by TGN46, in both ORP9 KO and QKO cells compared to WT controls (Fig. 6a, b). Overall levels of OSBP protein, as assessed by western

blotting, were comparable to WT in both ORP9 KO and QKO cells (Supplementary Fig. 6a), indicating that the high levels of OSBP at the TGN reflected recruitment of more OSBP to the TGN.

Because OSBP depends on PI4P at the TGN to transport cholesterol from the ER to the Golgi[60,62], we hypothesized that OSBP, which accumulated at the TGN in the absence of ORP9, was also

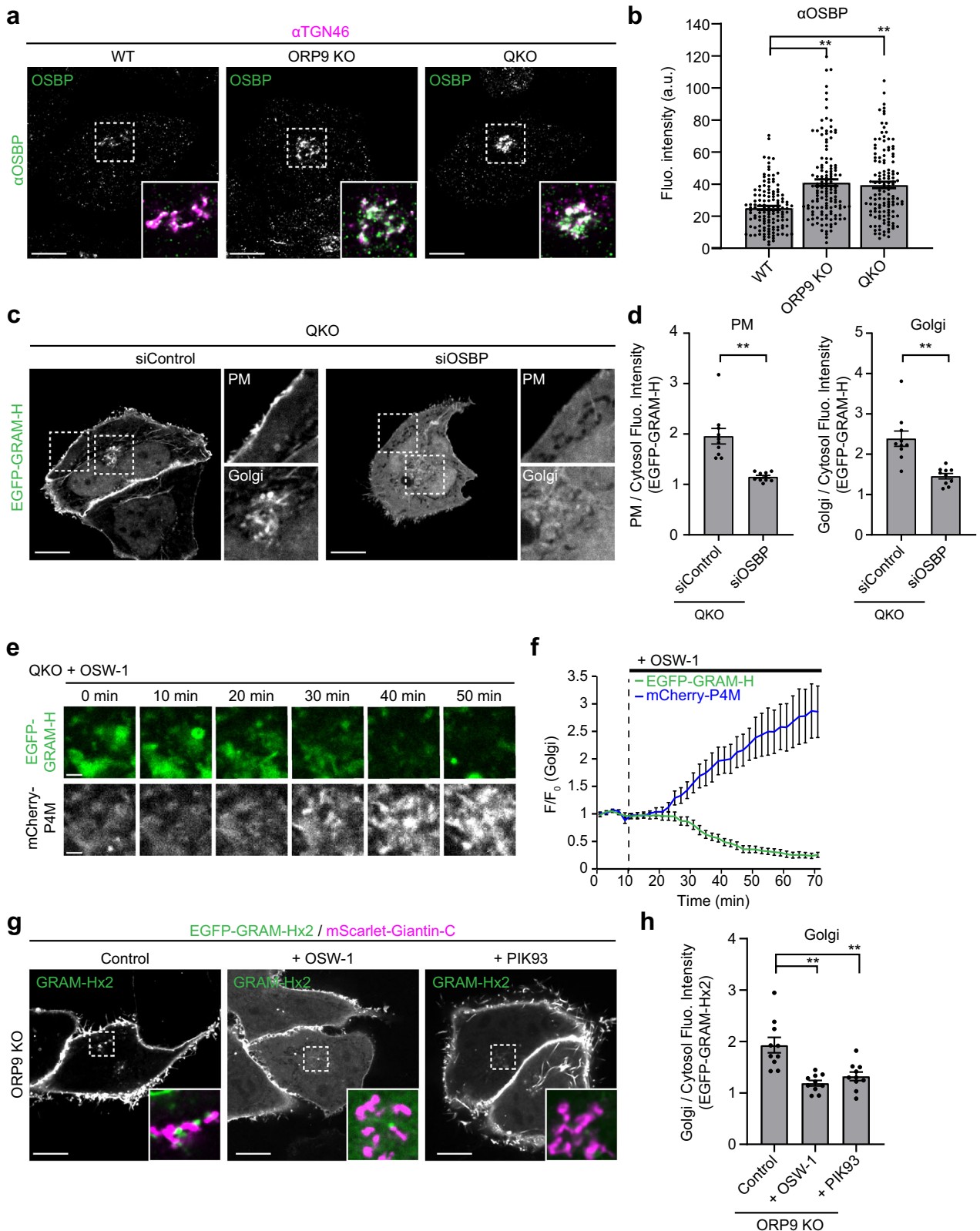

hyperactivated. Thus, we examined whether depletion/inhibition of OSBP is sufficient to reduce the accumulation of accessible cholesterol at the Golgi and PM in QKO cells. QKO cells expressing EGFP-GRAM-H were treated with siRNA against OSBP. Depletion of OSBP (~95%) was confirmed by western blotting (Supplementary Fig. 6b, c). Strikingly, knockdown of OSBP via RNAi completely abolished the accumulation of accessible cholesterol, as assessed by EGFP-GRAM-H or EGFP-D4H,

in both the Golgi and PM (Fig. 6c, d and, Supplementary Fig. 6d, e). Notably, TGN PI4P, which was assessed using mCherry-P4M, showed further accumulation in QKO cells upon knockdown of OSBP. This indicates that OSBP at the TGN of QKO cells actively consumes PI4P at ER-TGN contact in these cells (Supplementary Fig. 6f, g). We also monitored the effect of rapid chemical inhibition of OSBP in QKO cells, using OSW-1. QKO cells co-expressing either EGFP-GRAM-H or

**Fig. 6 | Accumulation of TGN PI4P in QKO cells is associated with hyper-activation of OSBP-mediated cholesterol transport to the Golgi. a** Confocal images of fixed HeLa cells that were immunolabeled with indicated antibodies. Insets show the regions around the Golgi as indicated by white dashed boxes (green: OSBP; magenta: TGN46). Scale bars, 10 μm. **b** Quantification of the signals of anti-OSBP fluorescence at the regions around the Golgi, as shown in (**a**) [mean ± SEM, $n = 131$ cells (WT), $n = 133$ cells (ORP9 KO), $n = 127$ cells (QKO); data are pooled from two experiments; Dunnett's multiple comparisons test, **$P < 0.0001$].
**c** Confocal images of live QKO HeLa cells expressing EGFP-GRAM-H that were treated with indicated siRNA for 72 hrs. Insets show the regions around the PM and Golgi as indicated by white dashed boxes. Scale bars, 10 μm. **d** Quantification of the ratio of PM signals (left) and Golgi signals (right) to the cytosolic signals of EGFP-GRAM-H, as shown in (**c**) [mean ± SEM, $n = 10$ cells for each condition; data are pooled from one experiment; two-tailed unpaired Student's *t*-test **$P < 0.0001$

(PM), **$P = 0.0002$ (Golgi)]. **e** Confocal images of the regions around the TGN of a live QKO HeLa cell expressing EGFP-GRAM-H and mCherry-P4M that were treated with OSW-1 (20 nM) as indicated. Scale bars, 2 μm. **f** Time course of normalized signals of EGFP-GRAM-H and mCherry-P4M in response to OSW-1 as shown in (**e**) (mean ± SEM, $n = 10$ cells for each condition; data are pooled from two independent experiments). **g** Confocal images of live ORP9 KO HeLa cells expressing EGFP-GRAM-Hx2 and mScarlet-Giantin-C that were treated with or without OSW-1 (20 nM for 1 h) or PIK93 (250 nM for 1 h). Insets show the regions around the Golgi as indicated by white dashed boxes (green: EGFP-GRAM-Hx2; magenta: mScarlet-Giantin-C). Scale bars, 10 μm. **h** Quantification of the ratio of Golgi signals to the cytosolic signals of EGFP-GRAM-Hx2 as shown in (**g**) [mean ± SEM, $n = 10$ cells for each condition; data are pooled from one experiment; Dunnett's multiple comparisons test, **$P < 0.0001$ (+ OSW-1), **$P = 0.0006$ (+ PIK93)]. Source data are provided as a Source Data file.

EGFP-D4H and mCherry-P4M were treated with OSW-1 and levels of accessible cholesterol and PI4P at the TGN were simultaneously monitored via time lapse imaging. OSW-1 treatment resulted in a marked increase in PI4P at the TGN within 40 min, which was accompanied by a significant decrease in accessible cholesterol at the Golgi (Fig. 6e, f and Supplementary Fig 6h, i). We further examined the effect of acute recruitment of OSBP to ER-TGN contacts using rapamycin-induced recruitment of the ER-anchored ORD of OSBP to the TGN. QKO cells stably expressing EGFP-GRAM-H and ER-mCherry-FKBP-OSBP were transiently transfected with tagBFP-TGN46-FRB (TGN-targeted FRB module) and iRFP-P4M. Recruitment of ER-mCherry-FKBP-OSBP to the TGN resulted in less PI4P and more accessible cholesterol at the Golgi (Supplementary Fig. 6j, k). Finally, we examined whether OSBP and PI4P are responsible for the increased levels of accessible cholesterol at the Golgi in cells lacking ORP9 alone [as observed using EGFP-GRAM-Hx2 (Fig. 3c, d)]. Inhibition of either OSBP (via OSW-1) or PI4KIIIβ (via PIK93) for 1 hour significantly reduced the levels of accessible cholesterol at the Golgi in ORP9 KO cells (Fig. 6g, h), suggesting that OSBP and PI4P contribute to the accumulation of accessible cholesterol at the Golgi in these cells.

Taken together, these results support our hypothesis that the accumulation of accessible cholesterol in ORP9 KO and QKO cells is caused by OSBP hyperactivation at the TGN. The deletion of ORP9 caused the accumulation of PI4P at the TGN, leading to the accumulation and hyperactivation of OSBP at the TGN, which enhanced cholesterol transport to the Golgi.

## GRAMD1b acts at ER-Golgi contact sites to remove excess cholesterol from the Golgi

Accumulation of accessible cholesterol at the Golgi was significantly enhanced in QKO cells, which lack ORP9 and all three GRAMD1s, compared to cells lacking ORP9 alone. These results suggest a potential role for the ER-anchored GRAMD1s in suppressing the accumulation of accessible cholesterol at the Golgi.

The GRAMD1 GRAM domain acts as a co-incidence detector for accessible cholesterol and anionic lipids[45]. Hence, when levels of cholesterol are elevated at the Golgi (which contains anionic lipids such as PS and PI4P) we reasoned that GRAMD1s may move to ER-Goliogi contact sites (via their GRAM domain) and transport excess accessible cholesterol from the Golgi to the ER (via their StART-like domain). To test this possibility, we asked whether re-expression of GRAMD1b in QKO cells would suppress the accumulation of accessible cholesterol at the Golgi and PM. QKO cells expressing EGFP-GRAM-H were transfected with either mRuby control, mRuby-GRAMD1b, mRuby-tagged mutant GRAMD1b in which the StART-like domain cannot transport cholesterol [mRuby-GRAMD1b (5P)], or mRuby-GRAMD1b (5P) additionally carrying an intellectual disability-associated mutation within the GRAM domain, which impairs its ability to sense accessible cholesterol [mRuby-GRAMD1b (R189W & 5P)][45,53,88]. Effects on the distribution of accessible cholesterol were

then compared using SDC microscopy. Expression of mRuby-GRAMD1b, but not mRuby-GRAMD1b (5P) or mRuby-GRAMD1b (R189W & 5P), in QKO cells reduced accessible cholesterol levels, as assessed by EGFP-GRAM-H or EGFP-D4H, to WT levels in both the Golgi and PM (Fig. 7a, b and Supplementary Fig. 7a, b). Interestingly, mRuby-GRAMD1b (5P) accumulated at both the Golgi and the PM, whereas mRuby-GRAMD1b and mRuby-GRAMD1b (R189W & 5P) were distributed as distinct puncta throughout the ER (Fig. 7a, c and Supplementary Fig. 7c). Accumulation of mRuby-GRAMD1b (5P) at the PM in QKO cells was further analyzed using TIRF microscopy. QKO cells showed much enhanced PM recruitment of mRuby-GRAMD1b (5P) compared to WT HeLa cells (Supplementary Fig. 7d, e), providing additional evidence that mRuby-GRAMD1b (5P) accumulates at the PM in QKO cells. These results suggest that GRAMD1b associates with the Golgi in addition to the PM by sensing accessible cholesterol via its GRAM domain and dissociates from these cellular compartments upon extraction of accessible cholesterol via its StART-like domain (Fig. 7a).

We next examined whether the accumulation of GRAMD1b (5P) at the Golgi in QKO cells depends on the accumulation of accessible cholesterol at the Golgi, which is caused by enhanced OSBP-mediated cholesterol transport in these cells (Fig. 6). Control HeLa and QKO cells co-expressing mRuby-GRAMD1b (5P) and EGFP-GRAM-H were treated with the OSBP inhibitor, OSW-1, and the distributions of mRuby-GRAMD1b (5P) and accessible cholesterol, as assessed by EGFP-GRAM-H, were compared before and after OSW-1 treatment (Fig. 7c, d). Before OSW-1 treatment, mRuby-GRAMD1b (5P) localized throughout the ER in control cells, whereas it accumulated at the Golgi (and weakly at the PM) in QKO cells. After OSW-1 treatment, accumulation of accessible cholesterol at both the Golgi and PM was suppressed. This was accompanied by the dissociation of mRuby-GRAMD1b (5P) from the Golgi and PM and its relocalization as distinct puncta throughout the ER (Fig. 7c, d). In control HeLa cells, mRuby-GRAMD1b (5P) localized throughout the ER before and after OSW-1 treatment (Fig. 7c, d). These data show that GRAMD1b senses elevations in accessible cholesterol at the Golgi and localizes to ER-Golgi contacts.

Finally, we used rapamycin to acutely recruit GRAMD1b to the TGN and asked whether GRAMD1b acts at ER-Golgi contacts to modulate levels of accessible cholesterol in the Golgi via their StART-like domain. We previously used a version of GRAMD1b in which the N-terminus GRAM domain was replaced by the FKBP module (miRFP-FKBP-GRAMD1b) to show that acute recruitment of this protein to the PM results in rapid reduction of accessible PM cholesterol in GRAMD1 TKO cells[53]. Here, miRFP-FKBP-GRAMD1b was recruited to the TGN in QKO cells. QKO cells co-expressing miRFP-FKBP-GRAMD1b and tagBFP-TGN46-FRB (TGN-targeted FRB module) together with mCherry-P4M (PI4P biosensor) and either EGFP-GRAM-H or EGFP-D4H (accessible cholesterol biosensors) were treated with rapamycin during time lapse imaging under SDC microscopy. Strikingly, rapamycin-induced recruitment of miRFP-FKBP-GRAMD1b to the TGN led to a rapid reduction in accessible cholesterol at the Golgi without affecting

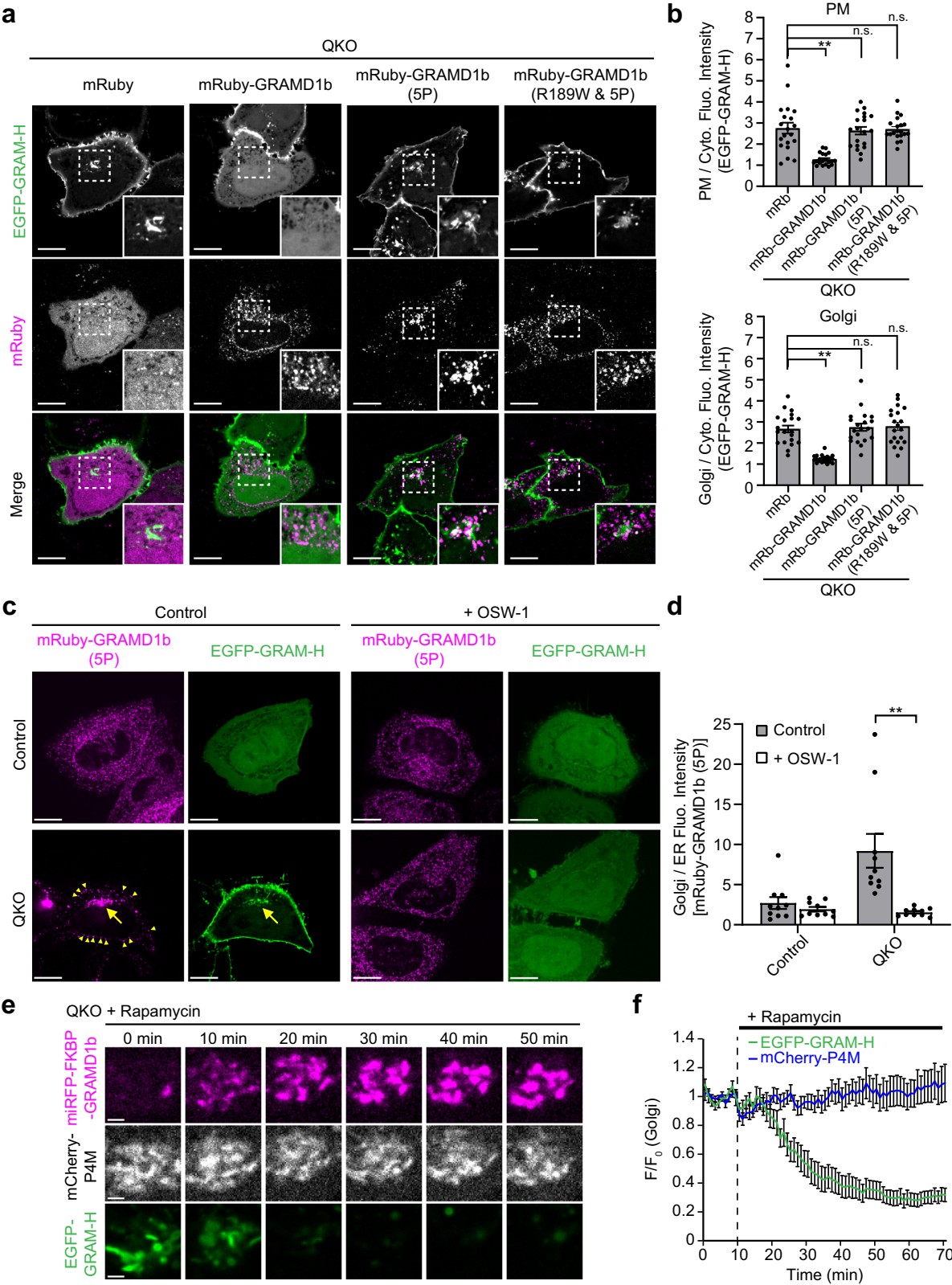

PI4P (Fig. 7e, f, Supplementary Fig. 8a, b, and Supplementary Movie 4). This supports a direct role for GRAMD1b in transporting accessible cholesterol from the Golgi to the ER. In contrast, the recruitment of miRFP-FKBP-GRAMD1b (5P) (whose StART-like domain cannot transport cholesterol) to the TGN did not change the levels of accessible cholesterol at the Golgi (Supplementary Fig. 8c, d), suggesting that GRAMD1b extracts accessible cholesterol directly from the Golgi via its

StART-like domain. To exclude the possibility that GRAMD1b depletes Golgi cholesterol indirectly by functioning at ER-PM contact sites, we expressed in QKO cells a version of mRuby-GRAMD1b, whose GRAM domain was replaced by the PH domain of PLCδ1 (i.e., PM-interacting module) (mRuby-PH-GRAMD1b) to induce its constitutive localization to ER-PM contact sites (Supplementary Fig. 8e). QKO cells expressing mRuby-PH-GRAMD1b showed significant reduction in the levels of

**Fig. 7 | GRAMD1b acts at ER-TGN contact sites to remove excess cholesterol from the Golgi. a** Confocal images of live QKO HeLa cells expressing either mRuby control, mRuby-tagged GRAMD1b (mRuby-GRAMD1b), mRuby-GRAMD1b carrying cholesterol binding-deficient StART-like domain [mRuby-GRAMD1b (5P)], or mRuby-GRAMD1b (5P) carrying cholesterol sensing-deficient GRAM domain [mRuby-GRAMD1b (R189W & 5P)] together with EGFP-GRAM-H (accessible cholesterol biosensor). Insets show at higher magnification the regions around the Golgi as indicated by white dashed boxes. Scale bars, 10 μm. **b** Quantifications of the ratio of PM signals (top) and Golgi signals (bottom) to the cytosolic signals of EGFP-GRAM-H, as shown in (**a**) (mean ± SEM, $n = 10$ cells for each condition; data are pooled from one experiment; Dunnett's multiple comparisons test, **$P < 0.0001$). **c** Confocal images of live WT and QKO HeLa cells expressing EGFP-GRAM-H and mRuby-GRAMD1b (5P) that were treated with or without OSW-1 (20 nM for 1 h; OSBP inhibitor) as indicated. Yellow allows indicate the site of mRuby-GRAMD1b

(5P) accumulation around the Golgi. Yellow arrowheads indicate the sites of mRuby-GRAMD1b (5P) accumulation at the PM. Note the dissociation of mRuby-GRAMD1b (5P) from the Golgi upon OSW-1 treatment. Scale bars, 10 μm. **d** Quantification of the ratio of Golgi signals to the endoplasmic reticulum (ER) signals of mRuby-GRAMD1b (5P), as shown in (**c**) (mean ± SEM, $n = 10$ cells for each condition; data are pooled from one experiment; two-tailed unpaired Student's *t*-test **$P = 0.0020$). **e** Confocal images of the regions around the TGN of a QKO HeLa cell expressing miRFP-FKBP-GRAMD1b and tagBFP-TGN38-FRB together with EGFP-GRAM-H and mCherry-P4M (PI4P biosensor) that were treated with rapamycin (200 nM) for the indicated minutes. Scale bars, 2 μm. **f** Time course of normalized signals of EGFP-GRAM-H and iRFP-P4M in response to rapamycin, as assessed by confocal microscopy as shown in (**e**) (mean ± SEM, $n = 18$ cells; data are pooled from four independent experiments). Source data are provided as a Source Data file.

accessible cholesterol in the PM, as assessed by EGFP-GRAM-H, compared to control QKO cells. By contrast, they showed only minor reduction in the levels of accessible cholesterol in the Golgi (however, this reduction is likely due to the increased levels of cytosolic GRAM-H signals in this condition) (Supplementary Fig. 8e, f). These results provide additional evidence to support a direct role of GRAMD1b in extracting cholesterol from the Golgi.

Based on these results, we conclude that GRAMD1s function at ER-Golgi contacts (in addition to their well-established function at ER-PM contacts) to prevent the buildup of cholesterol at the Golgi. GRAMD1s therefore counteract OSBP-mediated PI4P-driven cholesterol transport to the Golgi, which is suppressed by the extraction of PI4P from the TGN by ORP9. Hence, depletion of GRAMD1s together with ORP9 causes enhanced accumulation of accessible cholesterol at the Golgi compared to depletion of ORP9 alone, resulting in major accumulation of accessible cholesterol in the PM. The intricate crosstalk between these molecules at ER-Golgi contacts contributes to the maintenance of accessible cholesterol levels at the Golgi and post-Golgi membranes, including the PM.

## QKO cells exhibit dysregulated SREBP-2 signalling and increased cholesterol production

Our data suggest that the accumulation of accessible cholesterol in the Golgi and PM of QKO cells is caused by two related mechanisms: (1) an increase in cholesterol transport from the ER to the Golgi due to hyperactivation of OSBP-mediated cholesterol transport caused by depletion of ORP9, and (2) a decrease in cholesterol transport from the Golgi as well as the PM to the ER due to the absence of GRAMD1-mediated cholesterol transport. Thus, the accumulation of accessible cholesterol in these cells might be accompanied by chronic depletion of cholesterol in the ER.

The expression of genes responsible for cholesterol biosynthesis and uptake is regulated by SREBP-2, a master transcription factor that monitors levels of cholesterol in the ER. When levels of cholesterol in the ER dip below a certain threshold, SREBP-2 is delivered from the ER to the Golgi, where it is cleaved and activated[12–14]. The cleaved/activated form of SREBP-2 then enters the nucleus and increases the expression of genes that enhance cholesterol production and uptake. To examine whether QKO cells exhibit chronic depletion of cholesterol in the ER, we performed RNA sequencing analysis of WT, ORP9 KO, GRAMD1 TKO, and QKO cells and systematically analyzed the expression of genes involved in cholesterol biosynthesis and uptake.

RNA was extracted from cells incubated with 10% FBS, a condition in which SREBP-2 cleavage/activation is suppressed in WT cells[89]. mRNA libraries were constructed and subjected to massively parallel sequencing. Sequence reads were then aligned to the reference genome, and expression of each gene was compared between WT and KO cells. In QKO cells, 894 genes were upregulated and 716 were downregulated (more than a 2-fold increase/decrease) compared to WT cells (Fig. 8a). Remarkably, a number of genes involved in cholesterol

biosynthesis were upregulated in these cells (Fig. 8a). These upregulated genes include HMG-CoA reductase (HMGCR) and squalene epoxidase (SQLE), which encode rate limiting enzymes for cholesterol biosynthesis in the mevalonate pathway[90] (Fig. 8b). This is consistent with abnormal activation of SREBP-2. ORP9 KO cells upregulated the same set of genes, albeit to a lesser degree than seen in QKO cells. Finally, in GRAMD1 TKO cells these genes were expressed at WT levels (Fig. 8c and Supplementary Fig. 9). Collectively, these results indicate that the upregulation of genes involved in cholesterol biosynthesis is triggered by depletion of ORP9, which enhances cholesterol transport from the ER to the Golgi by OSBP hyperactivation. This upregulation of genes involved in cholesterol biosynthesis was further enhanced by additional depletion of GRAMD1s in QKO cells due to decreased delivery of cholesterol to the ER.

To further confirm that SREBP-2 was chronically activated in QKO cells, cells were incubated in 10% FBS and lysates were analyzed by western blot using antibodies against SREBP-2. The ratio of cleaved SREBP-2 (~60 kDa) to total SREBP-2 [cleaved SREBP-2 and uncleaved SREBP-2 (~135 kDa)] was compared. In WT and GRAMD1 TKO cells, ~3.5% of SREBP-2 was cleaved (Fig. 8d, e). Slightly more SREBP-2 was cleaved in ORP9 KO cells compared to WT and GRAMD1 TKO cells. Remarkably, more than 6% of SREBP-2 was cleaved in QKO cells, consistent with aberrant activation of SREBP-2 in these cells (Fig. 8d, e). Finally, we assessed the amount of total cellular cholesterol in these cells. Cells were incubated in 10% FBS and lipids were extracted. Total cellular cholesterol was measured using the reaction of cholesterol oxidase (Fig. 8f). WT, ORP9 KO, and TKO cells contained ~10 μg cholesterol/mg protein, whereas QKO cells had ~14 μg cholesterol/mg protein, consistent with increased production of cholesterol by these cells (Fig. 8f). Based on these results, we conclude that the SREBP-2 signalling pathway is chronically activated in cells lacking ORP9. QKO cells, lacking both ORP9 and GRAMD1s, show more severe activation of the SREBP-2 signalling pathway compared to ORP9 KO cells, resulting in the increased production of cholesterol.

These analyses reveal the critical importance of non-vesicular cholesterol transport at ER-TGN contact sites, and that ORP9, OSBP, and GRAMD1s regulate cholesterol transport at these sites to maintain cholesterol distribution and homeostasis (Fig. 9).

## Discussion

In this study, we show that interactions between ORP9, OSBP, and GRAMD1s at ER-Golgi contact sites play a major role in controlling the cellular distribution of cholesterol. ORP9 extracts PI4P from the TGN and contributes to the suppression of OSBP-mediated PI4P-driven cholesterol transport, whereas GRAMD1s extract cholesterol from the Golgi to prevent its build-up (Fig. 9). Via this mechanism, cells control the transport of cholesterol to the Golgi and post-Golgi membranes, including the PM, and maintain cholesterol homeostasis. Our major findings are the following: (1) Taking advantage of a novel accessible cholesterol biosensor based on the GRAM domain of GRAMD1b

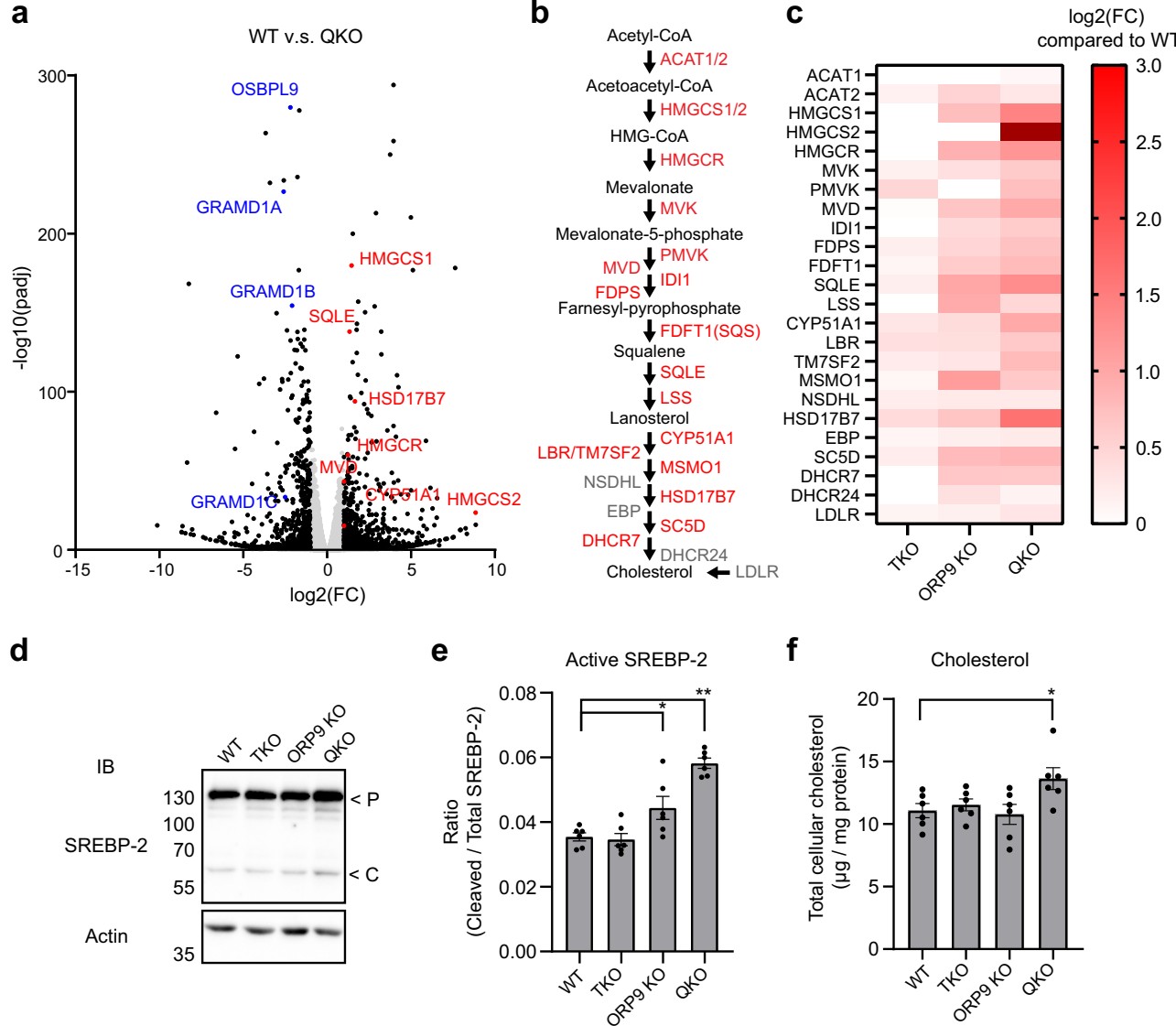

**Fig. 8 | QKO cells exhibit dysregulated SREBP-2 signalling and increased cholesterol production. a** Volcano plot showing transcriptome-wide changes in gene expression in QKO HeLa cells compared to WT HeLa cells. RNA was extracted from cells cultured in medium supplemented with 10% FBS. Data are presented with fold change in log2 [log2(FC)] and adjusted $P$ values in -log10 [-log10(padj)]. Black dots are differentially expressed genes [fold change > 2.0 adjusted $P$ values < 0.02]. Red dots are representative genes involved in the mevalonate pathway. Blue dots are genes deleted in QKO cells by CRISPR/Cas9. Data are pooled from three independent experiments. **b** The mevalonate pathway biosynthesizes cholesterol. Genes shown in red are upregulated in QKO HeLa cells compared to WT HeLa cells [fold change > 2.0 adjusted $P$ values < 0.02]. **c** Heatmap displaying relative expression of genes involved in the mevalonate pathway as shown in (**b**) in GRAMD1 TKO (TKO), ORP9 KO, and QKO HeLa cells compared to WT HeLa cells. **d** Lysates of WT, TKO,

ORP9 KO, and QKO HeLa cells that were cultured in medium supplemented with 10% FBS were processed for SDS-PAGE and IB with anti-SREBP-2 and anti-actin antibodies. Precursor(P) and cleaved (C) forms of SREBP-2 are indicated. **e** Quantification of the ratio of cleaved SREBP-2 to total SREBP-2 [mean ± SEM, $n = 6$ lysates (independent experiments) for each condition; Dunnett's multiple comparisons test, *$P = 0.0285$, **$P < 0.0001$]. **f** Quantification of total cellular cholesterol in WT, TKO, ORP9 KO, and QKO HeLa cells that were cultured in medium supplemented with 10% FBS. Lipids were extracted from the cells, and the amount of total cellular cholesterol was assessed by cholesterol oxidase reaction. Lysate of the cells were collected and processed for BCA protein assay for normalization (mean ± SEM, $n = 6$ independent experiment for each condition; Dunnett's multiple comparisons test, *$P = 0.0443$). Source data are provided as a Source Data file.

(GRAM-H), we performed a screen in GRAMD1 TKO cells and found that ORP9 plays a critical role in regulating the distribution of cellular cholesterol. Inhibition of OSBP phenocopies overexpression of ORP9 in GRAMD1 TKO cells, demonstrating an antagonistic relationship between OSBP and ORP9. (2) Both ORP9 and OSBP localize to the TGN albeit via distinct mechanisms. OSBP localizes to the TGN via its interaction with TGN PI4P, which is primarily generated by PI4KIIIβ. By contrast, ORP9 localizes to the TGN via cooperation of its N-terminal PI4P-sensing PH domain and its tandem α-helices, which interact with ORP10 and ORP11. Specific deletion of the tandem α-helices, but not the PH domain, causes dissociation of ORP9 from the TGN, indicating

the critical role of the tandem α-helices in localizing ORP9 to the TGN. (3) Our cell-based lipid extraction assays demonstrated that the ORP9 ORD preferentially extracts PI4P from cellular membranes. Via its ability to extract PI4P from the TGN, ORP9 contributes to the inhibition of OSBP-mediated PI4P-driven cholesterol transport to the Golgi. Therefore, ORP9 plays a critical role in controlling the abundance of cholesterol in the Golgi. (4) Depletion of ORP9 causes moderate accumulation of accessible cholesterol in the Golgi. Such accumulation is further enhanced when ORP9 and GRAMD1s are simultaneously depleted. Hence, at steady state, the levels of cholesterol at the Golgi are maintained by the removal of excess Golgi cholesterol by

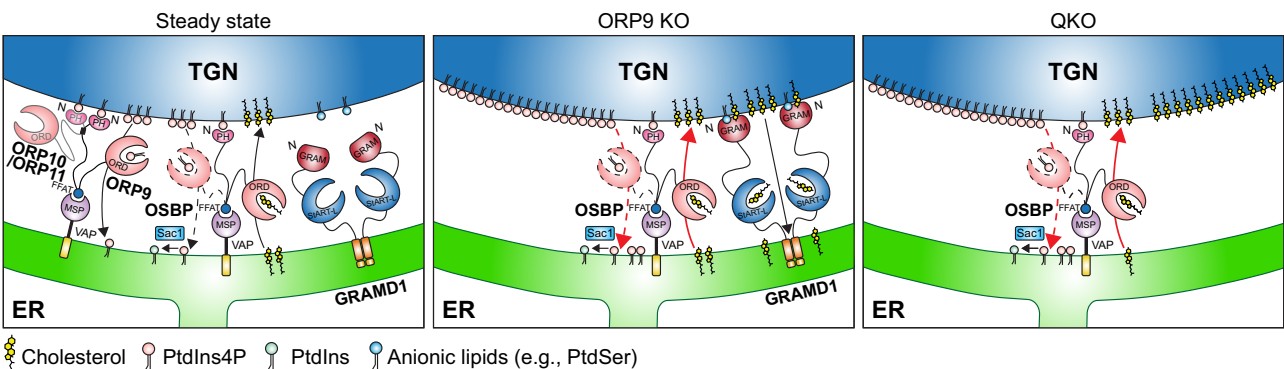

**Fig. 9 | Models of the regulation of non-vesicular cholesterol transport at ER-TGN contacts.** Models of the regulation of non-vesicular cholesterol transport at ER-TGN contacts. ORP9 and OSBP are recruited to the ER via interaction of their FFAT motif with the MSP domain of VAPs. GRAMD1s are anchored to the ER via their transmembrane domain. ORP9 is recruited to the TGN via interaction with ORP10 and ORP11, which is mediated by its tandem α-helices. Left: at steady state, ORP9 extracts PI4P from the TGN membrane via its ORD and inhibits OSBP-mediated PI4P-driven cholesterol transport from the ER to the TGN to maintain cholesterol levels at the Golgi. The GRAM domains of GRAMD1s only weakly interact with the TGN membrane at this state because of the limited abundance/accessibility of cholesterol in the Golgi. Middle: when levels of accessible cholesterol in TGN membranes rise above a certain threshold (e.g., OSBP hyperactivation due to ORP9

KO), GRAMD1s move to ER-TGN contacts by their ability to sense accessible cholesterol via their GRAM domain. They then transport excess cholesterol from the TGN to the ER via their StART-like domain, thereby preventing the buildup of cholesterol in the Golgi. Right: in the simultaneous absence of ORP9 and GRAMD1s (QKO), major accumulation of cholesterol occurs in both the Golgi and post-Golgi membranes, including the PM. This is caused by: (1) hyperactivation of OSBP (due to the lack of ORP9), and (2) impairment of cholesterol extraction and transport from the TGN (and also from the PM) to the ER (due to the lack of GRAMD1s). This is accompanied by chronic depletion of cholesterol from the ER membrane, resulting in aberrant activation of the SREBP-2 signalling pathway and overproduction of cholesterol.

GRAMD1s. (5) Cells lacking both ORP9 and GRAMD1s not only exhibit accumulation of accessible cholesterol at cellular membranes, but they also face constant depletion of cholesterol in the ER, which erroneously activates SREBP-2 signalling. These cells are also hypersensitive to amphotericin B, which kills cells by sensing accessible PM cholesterol, revealing a general disruption of cholesterol distribution and homeostasis in these cells.

ORP9 interacts with the ER and localizes to the TGN[73]. ORP9-ER interactions are mediated by binding of the FFAT motif of ORP9 with ER-anchored VAP proteins[73]. The precise mechanism by which ORP9 is recruited to the TGN, however, remained unclear. Our structural model of ORP9, which we generated using Alphafold, revealed the presence of tandem α-helices adjacent to its N-terminal PH domain. Remarkably, fusion of these tandem α-helices to an EGFP-tagged version of the ORP9 PH domain, which binds PI4P, restricted localization of the EGFP signal to the TGN, despite the presence of PI4P in both the TGN and PM. We found that the tandem α-helices interact with ORP9 itself, as well as ORP10 and ORP11. Simultaneous depletion of ORP10 and ORP11 resulted in dissociation of ORP9 from the TGN, showing that hetero-meric interactions between ORP9 and ORP10/11 contribute to the localization of ORP9. By contrast, the PH domain of ORP9 was not essential for its localization to the TGN, although it acts together with the tandem α-helices to promote the localization of ORP9 to the TGN. ORP9 was recently shown to play a role in recruiting ORP10 to ER-endosome contacts to facilitate ORP10-mediated PI4P/PS exchange between the ER and endosomes to regulate endosomal fission[74]. Moreover, the ORP9/ORP10/ORP11 hetero-meric complex is transiently recruited to contacts between the ER and damaged lysosomes to promote lysosomal membrane repair by transporting PS to damaged lysosomes[75]. Thus, ORP9, together with ORP10 and/or ORP11, plays critical roles in lipid exchange at various membrane contacts, including ER-endosome contacts[74], ER-lysosome contacts[75], and ER-TGN contacts (as shown in the current study), thereby contributing to cell physiology.

Inhibition of OSBP reduces the levels of accessible PM cholesterol to an extent similar to overexpression of ORP9 in GRAMD1 TKO cells. Hence, these two ER-associated LTPs act antagonistically to regulate the abundance of accessible PM cholesterol. While both OSBP and ORP9 transport cholesterol and PI4P between artificial membranes via

their ORDs[60,73,86], the antagonistic relationship between OSBP and ORP9 (revealed by our study) prompted us to re-examine their ligand preference in cell-based assays. When the ORD of OSBP was artificially recruited to the PM, levels of accessible cholesterol in the PM increased, whereas levels of PI4P in the PM decreased. This supports the well-characterized ability of OSBP to mediate PI4P/cholesterol counter transport between the ER and other membranes. By contrast, when the ORD of ORP9 was artificially recruited to the PM, only levels of PI4P in the PM decreased (with no detectable changes in accessible PM cholesterol). Therefore, while these two LTPs are able to extract and transport PI4P, only OSBP mediates PI4P/cholesterol exchange in live cells. Using a series of cell-based assays, we conclude that ORP9 removes PI4P from TGN membranes by acting at ER-TGN contacts, thereby contributing to the inhibition of OSBP-mediated PI4P-driven cholesterol transport to the TGN. OSBP affects the abundance of PM cholesterol by transporting cholesterol to the TGN[62,65]. In agreement with this function of OSBP, a recent study showed that a super complex of OSBP, ceramide transfer protein (CERT), and VAP functions at ER-TGN contact sites and influences the assembly of cholesterol nano-domains in the PM[91]. Our study is consistent with these findings, further demonstrating that the activity of OSBP at ER-TGN contact sites is tightly regulated by ORP9-dependent extraction of PI4P from the TGN. Exactly how Golgi cholesterol is transported to the PM remains unclear. While vesicular transport may mediate transport of some cholesterol from the Golgi to the PM, previous studies indicate the importance of non-vesicular transport for cholesterol transport to the PM[16,17,92]. Further investigation is needed to elucidate the mechanism of regulated transport of cholesterol between the Golgi and the PM.

One of the most well-characterized functions of the GRAMD1s is their ability to transport accessible cholesterol from the PM to the ER at ER-PM contacts[43,45,50,53,55,93,94]. Our results suggest that the GRAMD1s also sense elevated levels of accessible cholesterol in the Golgi and respond by transporting cholesterol from the Golgi to the ER at ER-Golgi contacts, thereby preventing cholesterol overaccumulation at the Golgi (Fig. 9 and Supplementary Fig. 10). In the presence of the GRAMD1s, depletion of ORP9 only induces moderate accumulation of cholesterol at the Golgi. Dramatic accumulation of accessible cholesterol occurs at the Golgi and PM when ORP9 and the GRAMD1s are simultaneously depleted (i.e., QKO cells). Our data suggest that this

accumulation results from the simultaneous occurrence of: (1) hyperactivation of OSBP-mediated cholesterol transport to the TGN (due to the depletion of ORP9), and (2) inactivation of cholesterol transport to the ER (due to the depletion of GRAMD1s) (Fig. 9). In QKO cells, acute recruitment of GRAMD1b to the TGN is sufficient to restore WT levels of accessible cholesterol at the Golgi, supporting a direct role for GRAMD1 in preventing the overaccumulation of accessible cholesterol at the Golgi. Further, a mutant version of GRAMD1b that cannot transport cholesterol was aberrantly recruited to the Golgi in QKO cells, indicating that cholesterol extraction is coupled with the dissociation of GRAMD1 from the Golgi. Based on the ability of the GRAM domain of GRAMD1s to detect the simultaneous presence of accessible cholesterol and anionic lipids[45], we propose that GRAMD1s may localize to a variety of membrane contact sites between the ER and other anionic lipid-containing organelles/membranes, and transport excess cholesterol from these organelles/membranes to the ER when levels of accessible cholesterol exceed a certain threshold (Supplementary Fig. 10). This function of GRAMD1s allows the ER, where cholesterol biosynthesis and uptake are regulated, to constantly monitor levels of cellular cholesterol and maintain cholesterol homeostasis. For example, GRAMD1s function at membrane contact sites between the ER and other organelles that include lysosomes and mitochondria[51,95–97]. In yeast, GRAMD1 homologues (Lam/Ltc proteins) localize to membrane contact sites between the ER and other organelles such as mitochondria, vacuoles (the lysosome equivalent in yeast), and the PM[98,99]. Whether GRAMD1/Lam/Ltc maintains the levels of sterol in these other organelles by transporting excess sterol to the ER remains to be investigated.

Finally, we show that the interplay between ORP9, OSBP, and the GRAMD1s at ER-Golgi contact sites play a critical role in maintaining cellular cholesterol homeostasis. When cholesterol transport is severely dysregulated at ER-Golgi contacts as in the case of QKO cells, accessible cholesterol is abnormally distributed to the Golgi and post-Golgi membranes, including the PM. Importantly, this abnormal distribution of cholesterol is coupled with dysregulation of the SREBP-2 signalling pathway and aberrant cholesterol production. Interestingly, cells lacking only ORP9 or only GRAMD1s do not show major changes in total levels of cellular cholesterol despite some accumulation of accessible cholesterol in the Golgi or the PM, respectively. Therefore, it seems that cells adapt to a certain degree of increase in the accessible pool of cholesterol in cellular membranes and maintain total levels of cellular cholesterol. The aberrant activation of SREBP-2 and increased levels of total cellular cholesterol in QKO cells are likely due to combined effects of OSBP hyperactivation (due to the depletion of ORP9) and inefficient cholesterol transport to the ER (due to the depletion of GRAMD1s). As depletion of ORP9 alone leads to moderate upregulation of SREBP-2 signalling pathways, the activity of OSBP seems to be tightly linked to that of SREBP-2. When hyperactivation of OSBP is coupled with inefficient cholesterol transport to the ER (as in the case of QKO cells), SREBP-2 is hyperactivated, resulting in dysregulation of cholesterol production. Thus, our study suggests that intricate crosstalk between the ER and Golgi, which is regulated by ORP9, OSBP, and the GRAMD1s at ER-Golgi contacts, plays a critical role in determining cellular cholesterol distribution and maintaining normal cell physiology. Future studies are needed to better understand how these LTPs functionally cooperate with other cholesterol-harboring LTPs, including StARDs[18,100] and other members of ORPs, for accessible cholesterol transport and distribution. Notably, GRAMD1b dysfunction is associated with various neurological disorders, including schizophrenia, intellectual disability, and multiple sclerosis[88,101–104], suggesting critical roles of GRAMD1b (and potentially other GRAMD1s) in maintaining neuronal development and function. While future studies are needed to dissect the physiological roles of these proteins in the brain, the current study provides novel insights into the function of GRAMD1s and their interplay with OSBP and ORP9 at ER-Golgi contacts, paving a way toward understanding the link between dysfunction of non-vesicular lipid transport and neurological disorders.

## Methods

### Materials
Primary and secondary antibodies, chemicals and other reagents, as well as DNA plasmids, the sequences of oligos and primers used are listed in Supplementary Table 1.

### Plasmid construction
*mRuby-ORP1* – cDNA of ORP1 (NP_542164) was amplified and assembled by PCR using gBlock fragments (IDT) containing ORP1 (XhoI_ORP1L_Frag1 and ORP1L_Frag2_KpnI) as templates and the following primer set (5'_MCS-XhoI_S and 3'_MCS_KpnI_AS). PCR product was ligated at XhoI and KpnI sites in the pmRuby-C1.

*mRuby-ORP2, mRuby-ORP3, mRuby-ORP7, mRuby-ORP10, and mRuby-ORP11* – Mammalian gene collection (MGC) verified full length clones of ORP2 (BC000296), ORP3 (BC017731), ORP7 (BC065482), ORP10 (BC003168), and ORP11 (BC065213) were obtained from the protein production platform (PPP) at Nanyang technological university (NTU), Singapore. cDNAs of ORP2, ORP3, ORP7, ORP10, and ORP11 were amplified using the following primer sets (ORP2: 5'_XhoI_ORP2_NS and 3'_KpnI_stop_ORP2_CAS; ORP3: 5'_XhoI_ORP3_NS and 3'_KpnI_Stop_ORP3_CAS; ORP7: 5'_EcoRI_ORP7_NS and 3'_SmaI_stop_ORP7_CAS; ORP10: 5'_HindIII_ORP10_NS and 3'_KpnI_stop_ORP10_CAS; ORP11: 5'_XhoI_ORP11_NS and 3'_KpnI_stop_ORP11_CAS) and ligated at XhoI and KpnI sites for ORP2, ORP3 and ORP11, EcoRI and SmaI sites for ORP7, and HindIII and KpnI sites for ORP10 in the pmRuby-C1.

*mRuby-ORP5 and mRuby-ORP6* – gBlocks (IDT) containing cDNAs of ORP5 (NP_065947) and ORP6 (NP_115912) were synthesized (XhoI-ORP5-HindIII and XhoI-ORP6-KpnI) and ligated at XhoI and HindIII sites for ORP5, and XhoI and KpnI sites for ORP6 in the pmRuby-C1.

*mRuby-ORP8* – cDNA of ORP8 (NP_065892.1) was digested from EGFP-OSBPL8 (a gift from De Camilli Lab) and ligated into pmRuby-C1 at SalI and ApaI sites.

*mCherry-OSBP* – cDNA of OSBP (NP_002547) of mRuby-OSBP[53] was digested and ligated at HindIII and BamHI sites in the mCherry-C1.

*mCherry-ORP9* – MGC verified clone containing cDNA corresponding to partial ORP9 (NP_078862.4) (186-736) was obtained from PPP at NTU, Singapore. cDNA of ORP9 (186-736) was amplified using the following primer set (5'_XhoI_ORP9_NS and 3'_HindIII_stop_ORP9_CAS) and ligated at XhoI and HindIII sites in the pmCherry-C1. gBlock (IDT) containing cDNA corresponding to the rest of ORP9 (NP_078862.4) (1-185) was synthesised and assembled into the plasmid using NEBuilder HiFi Assembly (NEB) to generate mCherry-ORP9.

*EGFP-ORP9 and tagBFP-ORP9* – cDNA of ORP9 was digested from mCherry-ORP9 and ligated into pEGFP-C1 and ptagBFP-C1 at XhoI and ApaI sites.

*EGFP-ORP9 (1-113), EGFP-ORP9 (1-195), EGFP-ORP9 (1-282), EGFP-ORP9 (196-282), EGFP-ORP9 (100-328), EGFP-ORP9 (283-736), EGFP-ORP9 (329-736), EGFP-ORP9 (100-736), and EGFP-ORP9 (114-736)* – cDNAs corresponding to the indicated amino acid residues of ORP9 (1-113; 1-195; 1-282; 196-282; 100-328; 283-736; 329-736; 100-736; 114-736) were amplified by PCR using EGFP-ORP9 as a template and the following primer sets (1-113: XhoI_ORP9_1_forward and KpnI_ORP9_113_reverse; 1-195: XhoI_ORP9_1_forward and KpnI_ORP9_195_reverse; 1-282: XhoI_ORP9_1_forward and KpnI_ORP9_282_reverse; 196-282: XhoI_ORP9_196_forward and KpnI_ORP9_282_reverse; 100-328: XhoI_ORP9L_100aa_S and KpnI_ORP9L_328_AS; 283-736: XhoI_ORP9_283_forward and XmaI_ORP9_736_reverse; 329-736: XhoI_ORP9_329_forward and XmaI_ORP9_736_reverse; 100-736: XhoI_ORP9L_100aa_S and XmaI_ORP9_736_reverse; 114-736: XhoI_ORP9_114_forward and XmaI_ORP9_736_reverse). The PCR products were ligated at XhoI and KpnI sites for 1-113, 1-195, 1-282, 196-282,

100-328, and 329-736, and XhoI and XmaI sites for 283-736, 100-736, and 114-736, in pEGFP-C1.

*EGFP-ORP9 (Δ114-195) and EGFP-ORP9 (R22E)* − KLD site-directed mutagenesis (NEB) was used to either delete the nucleotides, corresponding to the amino acid residues (114–195), of ORP9 or mutate nucleotides, corresponding to an amino acid residue (R22) of ORP9, to nucleotides encoding glutamic acid in EGFP-ORP9 using the following primer sets (Δ114-195: ORP9_196_S and ORP9_113_AS; R22E: ORP9_R22E_S and ORP9_R22E_AS).

*pLJM1-ER-mCherry-FKBP-ORP9* − cDNA corresponding to the ORD of ORP9 (329-736) was amplified by PCR using the primer set (link-KpnI-ORP9_329aa_S_hifi and BamHI-ORP9L_CAS_Hifi) and assembled at KpnI and BamHI sites in mCherry-FKBP-GRAMD1b (WT)[53] by NEBuilder HiFi Assembly (NEB) [to generate mCherry-FKBP-ORP9]. Subsequently, cDNA corresponding to the N-terminal ER anchoring region of E-Syt2 (1-117) of Myc-E-Syt2S (a gift from De Camilli lab) was amplified by PCR using the primer set (NheI-kozak-E-Syt2_S_Hifi and E-Syt2_mCherry_A-S_Hifi) and assembled at NheI sit of the plasmid by NEBuilder HiFi Assembly (NEB), to generate ER-mCherry-FKBP-ORP9. cDNA corresponding to ER-mCherry-FKBP-ORP9 was amplified by PCR using the primer set (NheI-kozak-E-Syt2_S_Hifi and EcoRI-stop-ORP9L_CAS) and assembled at NheI and EcoRI sites of pLJM1-EGFP by NEBuilder HiFi Assembly (NEB), to generate pLJM1-ER-mCherry-FKBP-ORP9.

*pLJM1-ER-mCherry-FKBP-ORP9 (HH/AA) and mCherry-ORP9 (HH/AA)* − KLD site-directed mutagenesis (NEB) was used to mutate nucleotides, corresponding to two amino acid residues (H501 and H502) of ORP9 to nucleotides encoding alanines in pLJM1-mCherry-FKBP-ORP9 or mCherry-ORP9 using the primer set, ORP9L_HAHA_S and ORP9L_HAHA_AS.

*mCherry-ORP9 (AAA)* − KLD site-directed mutagenesis (NEB) was used to mutate nucleotides, corresponding to three amino acid residues (L388, V391 and L393) of ORP9 to nucleotides encoding alanines in mCherry-ORP9 using the primer set, ORP9_AAA_mut_S and ORP9_AAA_mut_AS.

*pLJM1-ER-mCherry-FKBP-OSBP* − cDNAs corresponding to the FKBP of pLJM1-ER-mCherry-FKBP-ORP9 and the ORD of OSBP (348-807) of mRuby-OSBP[53] were amplified by PCR using the primer sets (FKBP: pmCherry-C1_Seq_For and KpnI_linker_FKBP_AS_HiFi; ORD of OSBP: KpnI_OSBP_348_S_HiFi and EcoRI_stop_OSBP_CAS_HiFi). The two PCR products were assembled at BsrGI and EcoRI sites of pLJM1-ER-mCherry-FKBP-ORP9 by In-Fusion HD cloning kit (Takara) to generate pLJM1-ER-mCherry-FKBP-OSBP.

*tagBFP-TGN38-FRB* − cDNA corresponding to tagBFP of pDonor-tagBFP-NLS (Addgene; #80766) was amplified by PCR using the primer set (AgeI-kozak-tagBFP-NS and XhoI-tagBFP-CAS) and ligated at AgeI and XhoI sites of TGN38 EGFP (Addgene; #128148). Subsequently, cDNA corresponding to the FRB of PM-FRB-tagBFP was amplified by PCR using the primer set, TGN38-FRB-S_HiFi and FRB_AS_HiFi, and assembled at BamHI site of the plasmid by NEBuilder HiFi Assembly (NEB) to generate tagBFP-TGN38-FRB.

*mCherry-VAPA and mCherry-VAPB* − cDNAs corresponding to VAPA of CMV::VAPA_3xFLAG (a gift from De Camilli lab) and VAPB of CMV::VAPB_3xFLAG (a gift from De Camilli lab) were digested and ligated into pmCherry-C1 at XhoI and EcoRI sites for VAPA and EcoRI and KpnI sites for VAPB.

*mRuby-PH-GRAMD1b* − cDNA corresponding to the PH domain of PLCδ1 (Addgene #66841) was amplified by PCR using the primer set (BglII-PHplcd-S and BglII-PHplcd-AS) and ligated at BglII site in mRuby-GRAMD1b (ΔGRAM)[53].

*EGFP-GRAM-Hx2* − cDNA corresponding to the mCherry-tagged GRAM domain of GRAMD1b with G187L mutation (mCherry-GRAM-H) was amplified by PCR using the primer set (NheI-mCherry-NS and AgeI_delStop_GRAM1b_AS) and ligated at NheI and BspEI sites in pAG153 FKBP-CFAST11 (Addgene #130813), to generate mCherry-GRAM-H-CFAST11. Subsequently, cDNA corresponding to the GRAM

domain of GRAMD1b (G187L) with GS linker was digested by BsrGI and BamHI and ligated at BsrGI and BglII sites in EGFP-GRAM-H, to generate EGFP-GRAM-Hx2.

*EGFP-D4H* − cDNA of corresponding to the Domain 4 of Perfringolysin O (D4) was amplified by PCR using gBlock (IDT) containing codon optimized D4 (D4 for mammalian expression) as a template and the following primer set (5' EcoRI D4_Mam and 3' KpnI D4_Mam_Stop). PCR product was ligated at EcoRI and KpnI site in the pEGFP-C1, to generate EGFP-D4. Subsequently, KLD site-directed mutagenesis (NEB) was used to mutate nucleotides, corresponding to an amino acid residue (D434) of D4 to nucleotides encoding serine in EGFP-D4 using the primer set, D4 D434S F and D4 D434S R, to generate EGFP-D4H.

## Cell culture and transfection

HeLa cells (Gift from Pietro De Camilli) and HEK293T cells (Gift from Nguan Soon Tan) were cultured in Dulbecco's modified Eagle's medium (DMEM) containing 10% fetal bovine serum (FBS) and 1% penicillin/streptomycin at 37 °C and 5% $CO_2$. Transfection of plasmids was carried out with Lipofectamine 2000 (Thermo Fisher Scientific) 1 day before the imaging. For RNAi-mediated knockdown experiments, cells were transfected with either control siRNAs (siNC-1 from IDT), a pool of siRNAs against ORP10 and ORP11 (hs.Ri.OSBPL10.13.1 and hs.Ri.OSBPL11.13.1 from IDT), or a pool of siRNAs against OSBP (hs.Ri.OSBP.13.1, hs.Ri.OSBP.13.2, and hs.Ri.OSBP.13.3 from IDT), using Lipofectamine RNAiMAX (Thermo Fisher Scientific) for 4 days (-96 h) in the case of ORP10/ORP11 double knockdown (DKD) or 3 days (-72 h) in the case of OSBP knockdown before the imaging. Wild-type as well as genome-edited HeLa cell lines were routinely verified as free of mycoplasma contamination at least every two months, using Myco-Guard Mycoplasma PCR Detection Kit (Genecopoeia). No cell lines used in this study were found in the database of commonly mis-identified cell lines that is maintained by ICLAC and NCBI Biosample.

## Generation of ORP9 KO and QKO HeLa cell lines

ORP9 knockout (KO) HeLa cells were generated by first identifying two gRNAs directed to the exon 16 of ORP9, which encodes the ORD, using CRISPOR (http://crispor.tefor.net/)[105]. The ORP9 genomic sequences targeted by the predicted CRISPR gRNAs are: GAAGGATCCCAAG-GATCGAA (ORP9-sgRNA#1) and TCGCCCAAAATGGGATTGTA (ORP9-sgRNA#2). The two CRISPR targeting sites were synthesized by annealing ORP9-sgRNA#1_S and ORP9-sgRNA#1_AS for ORP9-sgRNA#1, and ORP9-sgRNA#2_S and ORP9-sgRNA#2_AS for ORP9-sgRNA#2, respectively, and individually sub-cloned into PX459 (Addgene: #62988) to generate PX459-ORP9_sgRNA_#1 and PX459-ORP9_sgRNA_#2. Wild-type or GRAMD1 triple knock-out (TKO) HeLa cells[53] were transiently transfected with the two ORP9 CRISPR/Cas9 plasmids. 24 h after transfection, cells were supplemented with growth media containing puromycin (1.5 μg/mL) and incubated for 72 h. Cells resistant to puromycin were then incubated with puromycin-free medium for 24 h before harvesting for single cell sorting, and individually isolated clones were assessed by genotyping PCR using the primer set, ORP9_Exon16_GT_S2 and ORP9_Exon16_GT_AS2, to obtain ORP9 KO cell lines or quadruple knock-out (QKO) (lacking both GRAMD1s and ORP9) cell lines.

## Generation of endogenously tagged HeLa cell lines

Endogenous tagging of HeLa cells was carried out using the ORANGE system[76]. In brief, gRNAs directed close to the desired sites within the endogenous reading frame of the indicated target protein were identified using CRISPOR (http://crispor.tefor.net/)[105]. The genomic sequences targeted by the predicted CRISPR gRNAs are: GGTGCTGCCAAGCATTAGGT (ORP9_ORANGE_4_CT) and ATTGCAG-CACTTGGCGGCGG (OSBP_ORANGE_2_NT). The CRISPR targeting sites were synthesized by annealing ORP9_ORANGE_4_CT_S and ORP9_OR-ANGE_4_CT_AS for ORP9, and OSBP_ORANGE_gRNA_2_NT_S and

OSBP_ORANGE_gRNA_2_NT_AS for OSBP, respectively, and individually sub-cloned into pORANGE (Addgene, #131471). cDNA corresponding to the indicated fluorescent protein (mNeonGreen for ORP9, mScarlet-I for OSBP) with a linker region was then cloned into the HindIII and XhoI sites of the pORANGE to generate pORANGE_ORP9_4_CT and pORANGE_mScarlet-I-OSBP_site_2. 5 days after transfection of pOR-ANGE_ORP9_4_CT and pORANGE_mScarlet-I-OSBP_site_2, cells were imaged under spinning disc confocal (SDC) microscopy. Fluorescent positive cells were then isolated using BD FACSAria™ Fusion (BD Biosciences) based on the co-expression of mNeonGreen and mScarlet-I. Expression of tagged proteins (endoORP9-mNG and mSc-endoOSBP) was subsequently confirmed by microscopy and western blotting.

### Generation of ORP10/ORP11 DKO HeLa cells expressing endoORP9-mNG

ORP10/ORP11 double knockout (DKO) HeLa cells were generated using the Alt-R CRISPR-Cas9 System (IDT) and electroporation by the nucleofector system (Lonza) according to the manufacturer's proto-col. In brief, CRISPR gRNAs targeting ORP10 genomic sequences, ACAAGTTGAGGCACTCCCCA (ORP10-sgRNA#1) and GGGACT-CAATGGCGTGCACA (ORP10-sgRNA#2) were designed based on a previous study[75]. Two gRNAs targeting the exon 9 of ORP11 genomic sequences, which encodes the ORD, TTATCGGGGGAGTCTTTGAC (ORP11-sgRNA#1) and TACGGCCTTCATGAAATGAG (ORP11-sgRNA#2), were designed using CRISPOR (http://crispor.tefor.net/)[105]. CRISPR-Cas9 ribonucleoprotein (RNP) complexes consisting of the sgRNAs (ORP10:ORP10-sgRNA#1 and ORP10-sgRNA#2; ORP11: ORP11-sgRNA#1 and ORP11-sgRNA#2), and Cas9 nucleases (IDT) were added to $6.5 \times 10^5$ HeLa cells expressing endoORP9-mNG that were suspended in P3 primary cell nucleofector solution and supplement (Lonza). Electro-poration was carried out using the program "DS-138". Electroporated HeLa cells suspension was then plated and efficiency of DKO was confirmed by immunoblotting after 2 passages.

### Generation of cell lines that stably expressed EGFP-GRAM-H and either ER-mCherry-FKBP-ORP9, ER-mCherry-FKBP-ORP9 (HH/AA) or ER-mCherry-FKBP-OSBP

Lentiviral helper plasmids (pMD2.G, pRSV-REV, and pMDL/pRRE) (3.7 µg each) were transfected together with either one of the following plasmids [pLJM1-EGFP-GRAM-H, pLJM1-ER-mCherry-FKBP-ORP9, pLJM1-ER-mCherry-FKBP-ORP9 (HH/AA), or pLJM1-ER-mCherry-FKBP-OSBP] (7.4 µg) into $4.4 \times 10^6$ HEK293T cells according to manu-facturer's protocol. Supernatant was collected 48 h after transfection and filtered using a 0.45 µm filter unit to recover lentiviruses. QKO HeLa cells were seeded at $1.5 \times 10^5$ cells and transduced with lenti-viruses [containing Open Reading Frame (ORF) of EGFP-GRAM-H] to generate QKO cells stably expressing EGFP-GRAM-H. These stable cells were then seeded at $1.5 \times 10^5$ cells and transduced with lentiviruses [containing ORF of either ER-mCherry-FKBP-ORP9, ER-mCherry-FKBP-ORP9 (HH/AA), or ER-mCherry-FKBP-OSBP]. 2 days after the trans-duction, FACSAria™ Fusion (BD Biosciences) was used to isolate cells that are positive for both EGFP and mCherry fluorescence. The sorted cells were subsequently maintained in 0.5 µg/ml puromycin, and pro-tein expression was confirmed by microscopy and western blotting.

### Live cell imaging and immunofluorescence with fluorescence microscopy

For imaging experiments, cells were plated onto 35 mm glass bottom dishes at low density (MatTek Corporation).

*Fixed cells*—Cells were fixed with 4% paraformaldehyde (PFA), washed in phosphate-buffered saline (PBS), permeabilized with PBS containing 0.1% Saponin and 1% bovine serum albumin (BSA), immu-nostained with designated antibodies in the same buffer and main-tained in PBS. Primary antibodies were incubated at room temperature

for 1 h followed by incubation with Alexa Fluor-conjugated secondary antibodies at room temperature for 1 h. Fixed cell samples were imaged by SDC microscopy. Images from a mid-focal plane are shown.

*Live cells*— Cells were transfected 1 day before the imaging. Cells were washed once and incubated with $Ca^{2+}$ containing imaging buffer (140 mM NaCl, 5 mM KCl, 1 mM $MgCl_2$, 10 mM HEPES, 10 mM glucose, and 2 mM $CaCl_2$, pH 7.4) before imaging with either an SDC micro-scope or a TIRF microscope. All types of microscopy were carried out at 37 °C.

Spinning disc confocal (SDC) microscopy was performed on a setup built around a Nikon Ti2 inverted microscope equipped with a Yokogawa CSU-W1 confocal spinning head, a Plan-Apo objective (100×1.45-NA), a back-illuminated sCMOS camera (Prime 95B; Photo-metrics). Excitation light was provided by 488-nm/150 mW (Coherent) (for GFP/mNeonGreen/Alexa 488), 561-nm/100 mW (Coherent) (for mCherry/mRuby/mScarlet/Alexa 594) and 642-nm/110 mW (Vortran) (for iRFP/Alexa 647) (power measured at optical fiber end) DPSS laser combiner (iLAS system; Gataca systems), and all image acquisition and processing was controlled by MetaMorph (Molecular Device) software. Images were acquired with exposure times in the 400-500 msec range. For time-lapse imaging, images were sampled at 1/60-1/30 Hz with exposure times in the 500 ms range (Figs. 1f, g, 5c–f, 6e, f, 7e, f and Supplementary Figs. 6h–k, 8a–d).

Total internal reflection fluorescence (TIRF) was performed on a setup built around a Nikon Ti2 inverted microscope equipped with a HP Apo-TIRF objective (60×1.49-NA), and a back-illuminated sCMOS camera (Prime 95B; Photometrics). Excitation light was provided by 488-nm/70 mW (for GFP), 561-nm/70 mW (for mCherry) and 647-nm/125 mW (for iRFP) (power measured at optical fiber end) DPSS laser combiner (Nikon LU-NV laser unit), coupled to the motorized TIRF illuminator through an optical fiber cable. Critical angle was main-tained at different wavelengths throughout the experiment from the motorized TIRF illuminator. Acquisition was controlled by Nikon NIS-Element software. Images were sampled at 1/60 Hz with exposure times in the 500 msec range (Fig. 4b, c and Supplementary Fig. 4b).

### Drug stimulation for time-lapse imaging

For all time-lapse imaging experiments with drug stimulation, drugs were added to the cells 10 min after the initiation of the imaging. Drugs were used with the following concentration: 200 nM rapamycin; 20 nM OSW-1; 250 nM PIK93.

### Plasma membrane staining using the recombinant mCherry-D4H proteins

Cells were washed once with $Ca^{2+}$ containing imaging buffer and sub-sequently incubated with the same imaging buffer that contained purified mCherry-D4H proteins (10 µg/ml) for 15 min at room tem-perature. Cells were then washed twice with the same buffer without mCherry-D4H proteins and immediately imaged under SDC micro-scopy at room temperature.

### Image analysis

All images were analyzed off-line using ImageJ (http://fiji.sc/wiki/index.php/Fiji). Quantification of fluorescence signals was performed using Excel (Microsoft) and Prism 9 (GraphPad Software). All data are pre-sented as mean ± SEM. In dot plots, each dot represents value from a single cell with the bar as the mean.

For analysis of the recruitment of either EGFP-GRAM-H or EGFP-D4H to the PM via SDC microscopy (Figs. 1b, 3f, 4e, 6d, 7b and Sup-plementary Figs. 1b, d, 3f, 4d, 5f, 6e, 7b, 8f), line scan analysis was performed. A line of 5 µm in length was manually drawn around the PM, and EGFP fluorescence intensity along the manually drawn line was measured. The peak intensity around the PM region was normalized with the intensity of cytoplasmic region and then plotted for quantification.

For analysis of the binding of the recombinant mCherry-D4H to the PM via SDC microscopy (Supplementary Fig. 3d), the mean fluorescence intensity of mCherry-D4H at the PM was measured after background subtraction.

For analysis of the recruitment of endoORP9-mNG to the TGN via SDC microscopy (Fig. 2i and Supplementary Fig. 2o), cells were fixed and immunolabeled with antibodies against TGN46 as a TGN marker, and an arbitrary threshold was applied to each image of anti-TGN46 signal to segment the TGN region. The mean fluorescence intensity of endoORP9-mNG at the segmented TGN region were measured after background subtraction and plotted.

For analysis of the recruitment of either EGFP-GRAM-H, EGFP-GRAM-Hx2, or EGFP-D4H to the Golgi via SDC microscopy (Figs. 3d, f, 6h and Supplementary Figs. 3f, 4d, 5f, 6e, 7b), mScarlet-Giantin-C (Addgene, #85048) or iRFP-FRB-Giantin (Addgene #139313) was co-expressed as a Golgi marker, and an arbitrary threshold was applied to each image of Giantin signal to segment the Golgi (pixel size cut-off: 10-infinity). Process>Binary>Erode was used to further remove the background signals from the segmented images. Process>Binary>Dilate were then used three times to expand the segmented Golgi region. The mean fluorescence intensity of EGFP-GRAM-H at the segmented Golgi region and at the cytosolic region within the same cell were measured after background subtraction. Normalized values (the ratio of the mean signals in the Golgi to the mean signals in the cytosol) were then plotted.

For analysis of the recruitment of EGFP-GRAM-H or iRFP/mCherry-P4M to the Golgi via SDC microscopy (Figs. 4e, 6d, 7b and Supplementary Figs. 5f, 6g, 8f), an arbitrary threshold was applied to each image of EGFP-GRAM-H or iRFP/mCherry-P4M to segment the perinuclear Golgi region positive for accessible cholesterol (as assessed by EGFP-GRAM-H) and PI4P (as assessed by co-expressed iRFP/mCherry-P4M) in HeLa cells (pixel size cut-off: 10-infinity). ROI was drawn around the perinuclear region and the mean fluorescence intensity of EGFP-GRAM-H or iRFP/mCherry-P4M at the segmented region and at the cytosolic region within the same cell were measured after background subtraction. Normalized values (the ratio of the mean signals in the Golgi to the mean signals in the cytosol) were then plotted.

For analysis of the recruitment of iRFP-P4M to the TGN via SDC microscopy (Fig. 5b), SiT-N-mCherry (Addgene, #55133) was co-expressed as a TGN marker, and an arbitrary threshold was applied to each image of SiT-N-mCherry to segment the TGN region (pixel size cut-off: 10-infinity). Process>Binary>Erode>Dilate was used to further remove the background signals from the segmented images. The mean fluorescence intensity of iRFP-P4M at the segmented TGN region and at the cytosolic region within the same cell were measured after background subtraction. Normalized values (the ratio of the mean signals in the TGN to the mean signals in the cytosol) were plotted.

For analysis of immunostaining with antibodies against OSBP (Fig. 6b) in HeLa cells, the total fluorescence intensity of the signal of OSBP staining in each cell was measured after background subtraction and plotted.

For analysis of the recruitment of mRuby-GRAMD1b (5P) to the Golgi (Fig. 7d), ROIs were drawn around the perinuclear Golgi region (Golgi) and the peripheral region (ER). The mean fluorescence intensity of mRuby-GRAMD1b (5P) at each ROI was measured after background subtraction. Normalized values (the ratio of the mean signals in the Golgi to the mean signals in the ER) were plotted.

For analysis of the recruitment of mRuby-GRAMD1b (5P) to the PM (Supplementary Fig. 5e), ROI was drawn around a single cell using epifluorescence (Epi) image. The mean fluorescence intensity of mRuby-GRAMD1b (5P) in TIRF and Epi image at each ROI was measured after background subtraction. Normalized values [the ratio of the mean signals of TIRF image (PM) to the mean signals of Epi image] were plotted.

For the quantification of Manders' Overlap Coefficient (Supplementary Figs. 2a, c, d, e, i, 3b, 5b), JACoP plugin was used to obtain the Manders' Overlap Coefficient[106,107]. In brief, arbitrary threshold was applied to images of HeLa cells expressing indicated proteins to segment fluorescence signals of interest from the background. The overlapping fractions of the fluorescence signals derived from the plugin were then plotted.

For time-lapse imaging via SDC microscopy (Figs. 1g, 5d, f, 6f, 7f and Supplementary Figs. 6i, k, 8b, d), changes in fluorescence over time were analyzed by manually selecting regions of interest covering the largest possible area around the perinuclear Golgi region. An arbitrary threshold was applied to a single frame (10 min time point, right before the drug treatment) of the time-lapse images to segment the Golgi region positive for accessible cholesterol (as assessed by EGFP-GRAM-H) and PI4P (as assessed by co-expressed iRFP/mCherry-P4M) in QKO cells (pixel size cut-off: 10-infinity). The area and mean fluorescence intensity of EGFP-GRAM-H and iRFP/mCherry-P4M signals at the perinuclear Golgi region were measured after background subtraction for each time-lapse image, and the total fluorescence intensity of the fluorescence signals (product of area and mean fluorescence intensity) were normalized by the fluorescence signals before the drug treatment (10 min time point).

For time-lapse imaging via TIRF microscopy (Fig. 4b, c and Supplementary Fig. 4b), changes in PM fluorescence over time were analyzed by manually selecting regions of interest covering the largest possible area of the cell foot-print. Mean fluorescence intensity values of the selected regions were obtained and normalized to the average fluorescence intensity before stimulation after background subtraction.

## Immunoblotting

HeLa cells were lysed in buffer containing 2% sodium dodecyl sulfate (SDS), 150 mM NaCl, 10 mM Tris (pH 8.0), and incubated at 60 °C for 20 min followed by additional incubation at 70 °C for 10 min. The lysates were treated with benzonase nuclease (Sigma-Aldrich/Merck or SantaCruz) for 10–15 min at room temperature. The bicinchoninic acid assay (BCA assay) kit (Thermo Fisher Scientific) was used to measure protein concentration. Cell lysates were processed for SDS-PAGE and immunoblotting with standard procedure. All immunoblottings were developed by chemiluminescence using the SuperSignal West Dura reagents (Thermo Fisher Scientific). For the quantification of the knockout and knockdown efficiency (Supplementary Figs. 2k, m, and 6c), the intensity of chemiluminescence signals of the protein of interest was measured and plotted as indicated.

## Immunoprecipitation

HeLa cells expressing the indicated constructs (Supplementary Fig. 2h) were washed in cold PBS and lysed on ice in lysis buffer [50 mM Tris, 150 mM NaCl, 1% NP-40, 0.5 mM EDTA, 10% glycerol (pH 7.4) and protease inhibitor cocktail (Complete, mini, EDTA-free; Roche)]. Cell lysates were then centrifuged at $21,000 \times g$ for 20 min at 4 °C, and supernatants were kept and incubated with GFP-trap agarose beads (Chromotek) for 30 min at 4 °C under rotation. Subsequently, beads were washed in lysis buffer containing 1% NP-40 once and 0.2% NP-40 twice. Immunoprecipitated proteins bound to the beads were then incubated in PAGE sample loading buffer (containing 2% SDS) and further incubated at 60 °C for 20 min followed by additional incubation at 70 °C for 10 min. Immunoprecipitates were processed for SDS-PAGE and immunoblottings were carried out as described above.

## Amphotericin B resistance assay

$1 \times 10^4$ HeLa cells were plated onto 96-well plates one day prior to drug treatment. On the day of drug treatment, each well was washed with PBS twice and further treated with 100 μl DMEM containing 20% FBS, 1% penicillin/streptomycin, and Amphotericin B (25, 50, 100, 150, 200,

300, or 400 µg/ml) for 20 min at 37 °C. Cells were washed with PBS twice and then recovered in DMEM containing 20% FBS, 1% penicillin/streptomycin for 24 h at 37 °C. CellTiter-Glo® Luminescent Cell Viability Assay Kit was then used to quantify cell viability according to manufacturer's protocol.

## Cholesterol oxidase assay

$6 \times 10^4$ HeLa cells were plated onto 12-well plates two days prior to lipid extraction. Cells were cultured in DMEM containing 10% FBS and 1% penicillin/streptomycin at 37 °C and 5% $CO_2$ and treated with Trypsin-EDTA (0.25%; Gibco) for 3 min to be detached from plates. The suspension of the cells was washed with PBS and separated into two tubes. Cells in one tube were used for the quantification of cholesterol and cells in the other tube were used for the quantification of the amount of total proteins. For the quantification of cholesterol, lipids were extracted from cells according to the Bligh-Dyer method[108]. The Amplex Red Cholesterol Assay Kit (Thermo Fisher Scientific) was then used to measure cholesterol amount by the reaction of cholesterol oxidase. For the quantification of the amount of total proteins, cells were lysed in lysis buffer [2% SDS, 150 mM NaCl, 10 mM Tris (pH 8.0)] and incubated at 60 °C for 20 min followed by additional incubation at 70 °C for 10 min. The acquisition of fluorescence intensity was controlled by Gen5 3.09 (BioTek) software. The BCA assay kit (Thermo Fisher Scientific) was then used to measure protein concentration. Normalized values (the ratio of ng cholesterol to µg protein) were plotted.

## Expression and purification of mCherry-D4H

mCherry-D4H proteins were overexpressed in E. coli BL21-DE3 Rosetta cells as previously described[45,53]. A 750 mL culture was grown until $OD_{600}$ ~0.5-0.7 with appropriate antibiotics. 0.1 mM IPTG (Thermo Fisher Scientific) was then added, and the culture was further grown at 18°C for 18 hrs to allow protein expression. Cells were harvested by centrifugation at $4700 \times g$, at 4°C for 15 min, and re-suspended in 30 ml of lysis buffer (100 mM HEPES, 500 mM NaCl, 10 mM imidazole, 10% glycerol, 0.5 mM TCEP, pH 7.5), supplemented with protease inhibitors (Complete, EDTA-free; Roche) together with the cocktail of 100 µg/ml lysozyme (Sigma-Aldrich/Merck) and 50 µg/ml DNAse I (Sigma-Aldrich/Merck). Bacteria were lysed with sonication on ice in a Vibra Cell (Sonics and Materials, Inc) (70% power, 3 s pulse on, 3 s pulse off for 3 min for three to five rounds). The lysate was clarified by centrifugation at $47,000 \times g$, at 4°C for 20 min. The supernatants were incubated at 4°C for 30 min with Ni-NTA resin (Thermo Fisher Scientific), which had been equilibrated with 2.5 ml of wash buffer 1 (20 mM HEPES, 500 mM NaCl, 10 mM Imidazole, 10% glycerol, 0.5 mM TCEP, pH 7.5). The protein-resin mixtures were then loaded onto a column to be allowed to drain by gravity. The column was washed with 10 ml of wash buffer 1 twice and 10 ml of wash buffer 2 (20 mM HEPES, 500 mM NaCl, 25 mM imidazole, 10% glycerol, 0.5 mM TCEP, pH 7.5) twice, and then eluted with 1.25 ml of elution buffer 1 (20 mM HEPES, 500 mM NaCl, 500 mM imidazole, 10% glycerol, 0.5 mM TCEP, pH 7.5). The proteins were then concentrated using Vivaspin 20 MWCO 30 kDa (GE Healthcare) and further purified by gel filtration (Superdex 200 increase 10/300 GL, GE Healthcare) with elution buffer 2 (20 mM HEPES, 300 mM NaCl, 10% glycerol, 0.5 mM TCEP, pH 7.5), using the AKTA Pure system (GE Healthcare). Relevant peaks were pooled, and the protein sample was concentrated.

## Molecular modeling

The modelled structure of the human ORP9 (1-736) (UniProt: Q96SU4) was obtained from AlphaFold Protein Structure Database (https://alphafold.ebi.ac.uk)[78].

## In-gel protein identification of the proteins that interact with the tandem α-helices of ORP9

HEK293T cells expressing the indicated constructs (Supplementary Fig. 2f) were washed in cold PBS and lysed on ice in lysis buffer [50 mM Tris, 150 mM NaCl, 1% NP-40, 0.5 mM EDTA, 10% glycerol (pH 7.4) and protease inhibitor cocktail (Complete, mini, EDTA-free; Roche)]. Cell lysates were then centrifuged at $21,000 \times g$ for 20 min at 4 °C. Supernatants were kept and incubated with GFP-trap agarose beads (Chromotek) for 30 min at 4 °C under rotation. Subsequently, beads were washed in lysis buffer containing 1% NP-40 once and 0.2% NP-40 twice. Afterwards, immunoprecipitated proteins bound to the beads were incubated in PAGE sample loading buffer (containing 2% SDS) and further incubated at 60 °C for 20 min followed by additional incubation at 70 °C for 10 min. Immunoprecipitates were loaded and separated in 10% SDS−PAGE gel, and the gel was stained using colloidal blue staining kit (Invitrogen). Gels were then excised, and mass spectrometry was carried out by Protein and Proteomics Centre, National University of Singapore, to identify enriched proteins according to a standard procedure.

## mRNA-seq

$5 \times 10^5$ HeLa cells were plated onto 60 mm dishes two days prior to RNA extraction and cultured in DMEM containing 10% FBS and 1% penicillin/streptomycin at 37 °C and 5% $CO_2$. RNeasy mini kit (Qiagen) was used for the purification of total RNA from the cells. mRNA-seq and data analysis were performed by Genewiz / Azenta Life Sciences. For each genotype (WT, ORP9 KO, GRAMD1 TKO, and QKO), data were obtained from three independent experiments. In brief, 1 µg total RNA was used for library preparation and subsequent mRNA-seq experiments. mRNA isolation was performed using Oligo(dT) beads, and the mRNA fragmentation was performed using divalent cations and high temperature. First strand cDNA and the second-strand cDNA were synthesized using random primers. The purified double-stranded cDNA was then treated to repair both ends and add a dA-tailing in one reaction, followed by a T-A ligation to add adaptors to both ends. Size selection of Adaptor-ligated DNA was then performed using DNA Clean Beads. Each sample was then amplified by PCR using P5 and P7 primers and the PCR products were validated. Then libraries with different indexs were multiplexed and loaded on an Illumina Novaseq6000 S4 flow cell for sequencing using a $2 \times 150$ paired-end (PE) configuration according to manufacturer's instructions. Obtained data were cleaned and aligned to reference genome via software Hisat2 (v2.0.1). HTSeq (v0.6.1) was used to estimate gene and isoform expression levels from the pair-end clean data. Finally, DESeq2 Bioconductor package was used to perform differential expression analysis. The estimates of dispersion and logarithmic fold changes incorporate data-driven prior distributions, Padj of genes were set <0.05 to detect differential expressed ones.

## Statistics and reproducibility

No statistical method was used to predetermine sample size, and the experiments were not randomized for imaging. Sample size and information about replicates are described in the figure legends. All experiments were independently conducted at least 2 times to confirm reproducibility. The number of biological replicates is shown as the number of independent experiments within figure legends for each figure. Comparisons of data were carried out by the two-tailed unpaired Student's t-test, or the one-way ANOVA followed by Tukey or Dunnett corrections for multiple comparisons as appropriate with Prism 9 (GraphPad software). Unless $P < 0.0001$, exact $P$ values are shown within figure legends for each figure. $P > 0.05$ was considered not significant. All data are presented as mean ± SEM unless otherwise noted.

**Reporting summary**

Further information on research design is available in the Nature Portfolio Reporting Summary linked to this article.

## Data availability

The authors declare that the data supporting the findings of this study are available within the paper and its supplementary information file and available from the corresponding author upon request. Reagents and strains generated for this study are available directly from the authors upon request. The structure for the ORP9 can be accessed from the AlphaFold Protein Structure Database [https://alphafold.ebi.ac.uk/entry/Q96SU4]. All RNA sequencing data are available at the Gene Expression Omnibus accession number GSE240960. Source data are provided with this article. Source data are provided with this paper.

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

## Acknowledgements

We thank Bilge Ercan, Dennis Dharmawan, Raihanah Harion, Minyoung Na, Vini Natalia, Darshini Jeyasimman, and Jingbo Sun for discussion. This work was supported by the Singapore Ministry of Education Academic Research Fund Tier 2 (MOE-T2EP30120-0002 & MOE2017-T2-2-001), the Singapore Ministry of Education Academic Research Fund Tier 1 (RG20/21), and Grant-in-Aid for Scientific Research (B) (22H02620) from the Japanese Society for Promotion of Science (JSPS), and a Nanyang Assistant Professorship (NAP) to Y.S., LKCMedicine LEARN grant to T.N., and Singapore Ministry of Education Academic Research Fund Tier 2 (T2EP30221-001) to L.L. T.N. was supported by a fellowship from the JSPS.

## Author contributions

All authors participated in the design of experiments, data analysis and interpretation. T.N. performed most of the experiments. H.Y. characterized the mechanisms responsible for the localization of ORP9 to the Golgi. D.H.Z.K. performed amphotericin sensitivity assays. Y.S. participated in designing the imaging and cell-based biochemical assays that were performed by T.N., H.Y., and D.H.Z.K. L.L. and D.M. participated in designing the imaging-based assays for characterization of the Golgi that were performed by T.N. T.N. and Y.S. wrote the manuscript with input from all the authors.

## Competing interests

The authors declare no competing interests.
