## [Peer Review File · Nature Communications]

Regulation of cellular cholesterol distribution via non-vesicular lipid transport at ER-Golgi contact sitesREVIEWER COMMENTS

Reviewer #1 (Remarks to the Author):

This work investigated how ORP proteins may control the trafficking of endogenous cholesterol. The authors made use of the TKO cells that accumulate plasma membrane cholesterol because they lack all three GramD1 proteins, and found that ORP9 overexpression can suppress the accumulation of PM cholesterol in these cells. They further demonstrated that QKO cells lacking all GramD1s and ORP9 accumulated more PM and Golgi cholesterol, which was due to increased PI4P on the Golgi and a hyperactive OSBP. Moreover, GramD1 proteins can transfer cholesterol back to the ER from the Golgi in normal cells. Overall, this study proposed an interesting collaborative network that helps maintain Golgi cholesterol and PI4P homeostasis. This is an important contribution to the field of nonvesicular lipid trafficking. The connection between ORPs and GramD1 is highly interesting. However, there are some issues to be addressed and clarified before acceptance.

1. The central conclusion here is that ORP9 controls Golgi PI4P, which then controls OSBP function. If this is the case, then other means that can increase PI4P in the TKO cells should also activate OSBP and causes cholesterol accumulation on the Golgi and PM. This should be tested. For instance, what happens when PI4K3b is overexpressed in the TKO cells? What happens when PI4K3b is silenced or inhibited in the QKO cells?
2. The role of ORP10 and 11 in ORP9 localization and function should be investigated further. Does ORP9 localize to Golgi in ORP10/11 deficient cells? Are there any synergistic effects on PM cholesterol when ORP9/10/11 are co expressed? (Fig. S1A-1B). These assays would provide more insights into the interaction between ORP9 and 10/11.
3. Fig. S1A/B, why ORP5 and ORP6 are missing? The effect of ORP2 is very strong (stronger than OSBP). Is there any possibility that ORP9 might also modulate ORP2 function?
4. The authors avoided mentioning PS, which is the major ligand for ORP9. Why is that? Would PS play a role in the ORP9/OSBP/GramD1 network?
5. The direct role of GramD1 at the Golgi may need more evidence. For instance, the effect on Golgi cholesterol upon GramD1b re-expression might be indirect: it depletes PM cholesterol which allows more Golgi cholesterol to reach the PM (Fig. 7A). In Fig. 7C, the 5P mutant accumulated on the Golgi but not the PM. Why is that?
6. In many figures including 2C-G, S2A, 3G, S5A, there is a single image of an apparent morphology/phenotype (which presumably applies to the whole cell population) without any quantification, number of cells or biological replicates in which this was observed, or correlation analyses. It would be preferable for the authors to provide correlations or quantitation, like in Figures 4E and 6B, to demonstrate these findings more objectively.
7. The mechanism behind the increase in PM cholesterol with ORP9 KO in the TKO cells is unclear, given endogenous OSBP is not thought to transport cholesterol to the PM. The authors mention the role of

CERT in the discussion as the factor underlying the TGN-PM transport of cholesterol in the QKO line. Clarification of this would be beneficial to the impact of the study and could be investigated with the depletion of CERT in the QKO line.

8. In Figure 3B there is still a lot of signal in the immunostaining of ORP9 in the ORP9 KO cell line, is this thought to be non-specific staining? The authors should comment on this.

9. It would be useful to have the concentration and duration of OSW-1 treatment in the text and/or figure legends.

Reviewer #2 (Remarks to the Author):

In Naito et al., authors show that two membrane contact site proteins, OSBP and ORP9, localizing between the ER and the Golgi regulate cholesterol dynamics in GRAMD1 knockout cells. Authors show that ORP9 and OSBP over-expression can rescue and elevate phenotype caused by the loss of GRAMD1. Furthermore, loss of ORP9 furthers the phenotype of cholesterol accumulation at the plasma membrane and causes cholesterol accumulation in the Golgi apparatus, latter of which is caused by OSBP. I find authors observations solid and there is some novelty to their findings. However, I find the manuscript insufficient for publications due to various issues. One of the issues is that most of the observations are done in GRAMD1 triple and GRAMD1-ORP9 quadruple knockout cells. This raises the question whether this is biologically any significant has any biological meaning, i.e. what observed is most likely a compensatory mechanism to what might never occur in nature. Is there is a condition, e.g. starvation or virus infection, where ORP9 in addition to all three GRAMD1 proteins are somehow inhibited that might lead to increased Golgi cholesterol levels. A list of issues are listed below:

Main Issues:

1) Main issue is that the majority of the conclusions are based on the observations made in GRAMD1 triple knockout (TKO) cells, which led to the characterizing a role for OSBP and ORP9 in cholesterol distribution in these TKO cells. While this appears to be a compensatory mechanism in TKO cells, whether this takes places in normal conditions, i.e. has a biological significance, is a big question mark to be addressed.

2) Page 2, Highlight nr. 1: "ORP9 ... [is]... a key regulator of intracellular cholesterol distribution." & Page 10, Line 1: "...ORP9 is important for proper distribution of cholesterol in the Golgi and post Golgi membranes, including the plasma membrane".

This is inaccurate. Authors show that this is only the case when all three GRAMD1 proteins are dysfunctional, which itself already causes cholesterol deregulation, knockout of ORP9 only contributes to it. ORP9 knockout cells show no changes in intracellular cholesterol distribution (figure 3).

3) Page 2, Highlight nr. 2: “Cells lacking ORP9 and GRAMD1s show massive cholesterol accumulation”.

Massive is very subjective. TKO cells already have elevated plasma membrane cholesterol levels and loss of ORP9 only contributes to this. More importantly, figure 8F shows only 10-20% increase in total cholesterol levels, so it is hardly “massive”.

4) Page 2, Highlight nr. 3: “ORP9 ... inhibits OSBP-driven cholesterol transport to the TGN”.

The findings are too little to conclude that ORP9 blocks OSBP-driven cholesterol transport, neither in TKO nor wild-type cells. The results only lead to the conclusion that OSBP is responsible for the cholesterol accumulation in Golgi of QKO cells.

5) Some aspects of the paper seem redundant to what is described in the field or within itself:

- Figure 2 showing ORP9 localization and its known interaction with ORP10 and ORP11 (Zhou 2010; Finkel 2022).

- Both figure 4 and 5 are showing that ORP9 can extract PI4P.

6) I find it risky that the study mainly uses the GRAMD1-based cholesterol sensor for detecting changes in the cholesterol distribution caused by GRAMD1-depletion, especially in the figures 6 and 7. D4H, another cholesterol binding protein, is also used but only to a limited extent. Many of the cholesterol experiments must be repeated using D4H to support conclusions.

7) Cholesterol is most abundant in the extracellular leaflet of the plasma membrane and in the Golgi is accepted to accumulate at the luminal side of the membrane. Both D4H experiments (figure S3) and GRAMH experiments suggest that both cytosolic and extracellular leaflets of the plasma membrane have increased cholesterol levels. What is the authors' explanation to accumulation to the cytosolic leaflet?

8) Figure 8 is not convincing enough to explain the difference in cholesterol levels. It appears that the manuscript is rather incomplete.

- There is no cholesterol increase in TKO cells and QKO have about 10-20%.

- SQLE, HMGCR, HMGCS1, DHCR24 are upregulated in ORP9 KO cells without affecting cholesterol levels or distribution.

9) Authors nicely show that the trans Golgi have various subdomains. I do not know how this is relevant to the rest of the story. Regardless, I wonder what the location of cholesterol accumulation in QKO is. Is it the ORP9 or OSBP subdomain?

Minor Issues:

1) If OSBP responsible for cholesterol accumulation in Golgi of TKO cells, would not its over-expression in figure S1A cause cholesterol accumulation in Golgi? Over-expression of ORP2 does it to an extent, at least some place in the cytoplasm.

2) "GRAMD1s extract cholesterol from the TGN to prevent its build up in the Golgi." They show by live cell imaging, which is very nice.! It can extract from Golgi in QKO, but does it ever occur in normal conditions?

3) Figure 7A is not uniform ER distribution!

4) Why use purified EGFP-D4H for permeabilized and mCherry-D4H for impermeabilized cells?

Reviewer #3 (Remarks to the Author):

This is an encyclopedic paper, with a large number of technically sophisticated experiments that make use of a huge number of bespoke reagents and knock-out cell lines. The sheer volume of minutiae makes the paper very difficult to read. Indeed, without consulting the graphical abstract and the summary at the start of the Discussion, it is next to impossible to track the logical flow of the paper to try to understand the experiments and their conclusions.

The experiments are set up and viewed through the singular lens of the PI4P/sterol exchange model, focused on contact sites, without reference to the broader literature on intracellular cholesterol dynamics and distribution. For example, the literature is clear that cholesterol equilibrates between the ER and PM such that its chemical activity is similar in both compartments (Kaplan and Simoni, JCB 1985; Baumann et al. Biochemistry 2005, Menon Curr Opin Cell Biol 2018). Thus, an excess above the sequestering capacity of the PM is dissipated to the ER to restore the chemical activity balance. This can be achieved by any LTP, regardless of whether it is at a contact site. For example, the diffusible, ubiquitously abundant LTP StARD4 could carry out this role (Iaea, Maxfield et al. Mol Biol Cell 2017) but is not referred to anywhere in the manuscript. Indeed, it is surprising to see the specific effects that are reported in the current manuscript in the face of what is surely a high LTP activity provided by StARD4.

The statement in the Introduction that the PM contains most of the cholesterol content of the cell, estimated at 90% of cell cholesterol, is wrong. This statement is often presented in papers and review articles, most recently by Kennelly and Tontonoz (2023), but is wrong, nevertheless. A simple calculation demonstrates this. In a hepatocyte, the relative areas of PM and ER are 2 and 50% of total cell membrane (<http://book.bionumbers.org/how-big-is-the-endoplasmic-reticulum-of-cells/>), respectively and the cholesterol concentrations at the PM and ER are 40 and 5 mole percent, respectively. Assuming for the moment that there are no other relevant membranes to consider, then the relative amount of cholesterol mass in the PM and ER is $40 \times 2 = 80$ and $5 \times 50 = 250$, respectively. PM cholesterol is therefore $100 \times 80 / (80 + 250)$, i.e., <25% of cell cholesterol. If there is cholesterol in other membranes - such as mitochondria and the endocytic recycling compartment, as will inevitably be the case, then the proportion of cholesterol in the PM is even less. This misstatement of experimental information is a poor starting point for this paper.

A note about the use of the GRAM sensor - the authors should be aware of the cautionary work of Courtney, Zha et al. eLife 2018 where it is made clear that many factors, including high protein concentrations such as found in the cytoplasm, can affect the readout.

REBUTTAL TO THE COMMENTS RAISED BY THE REVIEWERS

We thank the editor and the reviewers for the constructive comments and suggestions. We have carried out extensive new experimentation to address all the concerns and made necessary changes to the texts.

We have now

- 1) used an alternative cholesterol biosensor based on the domain 4 of the Perfringolysin O protein (D4H) and shown that all the phenotypes that we reported using the GRAM domain-based biosensor (GRAM-H) in our original manuscript were reproduced (for Reviewers #2 and #3).
- 2) investigated the role of other ORPs and provided further evidence that ORP9 primarily modulates the activity of OSBP to control intracellular cholesterol distribution (for Reviewer #1).
- 3) investigated physiological relevance of our study and provided new evidence that ORP9 functions to control cholesterol distribution and homeostasis even in the background, where GRAMD1s are present (for Reviewer #2).
- 4) provided additional evidence supporting the direct role of GRAMD1b in mediating cholesterol transport at ER-Golgi contacts (for Reviewer #1).
- 5) performed rigorous quantification of various images and strengthened the live cell imaging studies with addition of new experiments/images as requested.

A specific list of changes is indicated below, followed by a point-by-point rebuttal from page 3.

Supplementary Figure 7 is now split into **new Supplementary Fig. 7** and **new Supplementary Fig. 8** with additional data.

Supplementary Figure 8 is now in **Supplementary Fig. 9**.

We now have eight main figures (**Fig. 1-8**) and nine supplementary figures (**Supplementary Fig. 1-9**).

The following data are new:

Data using an alternative cholesterol sensor

Supplementary Fig. 3e, f [showing the analysis of indicated cell lines with an alternative cholesterol biosensor, EGFP-tagged D4H (EGFP-D4H)]

Supplementary Fig. 4c, d [showing the rescue of the phenotype of QKO cells by re-expression of ORP9, but not by re-expression of ORP9 (HH/AA), using EGFP-D4H]

Supplementary Fig. 6d, e (showing the decrease in the association of EGFP-D4H to the PM and Golgi by OSBP RNAi in QKO cells)

Supplementary Fig. 6h, i (showing the decrease in the association of EGFP-D4H to the Golgi by a chemical inhibitor of OSBP, OSW-1, in QKO cells)

Supplementary Fig. 7a, b [showing the rescue of the phenotype of QKO cells by re-expression of GRAMD1b, but not by re-expression of GRAMD1b (5P), using EGFP-D4H]

Supplementary Fig. 8a, b (showing the decrease in the association of EGFP-D4H to the Golgi by rapamycin-dependent acute recruitment of GRAMD1b to ER-TGN contacts in QKO cells)

Further investigation of ORPs

Fig. 4d, e [showing additional images and graphs obtained by expressing phosphatidylserine binding-deficient mutant of ORP9, ORP9 (AAA), in QKO cells]

Supplementary Fig. 1a, b (showing additional images and graphs obtained by expressing mRuby-ORP5 and mRuby-ORP6 in GRAMD1 TKO cells)

Supplementary Fig. 1c, d (showing new images and graphs obtained by co-expressing OSBP and ORP9 or co-expressing ORP2 and ORP9 in GRAMD1 TKO cells)

Supplementary Fig. 2j-l (showing that the double knock-down of ORP10 and ORP11 has little effect on the localization of ORP9 at steady state)

Appendix Fig. 2 for reviewer #1 (showing that co-expression of ORP9 together with either ORP10 or ORP11 does not enhance the effect of the expression of ORP9 alone in GRAMD1 TKO cells)

Appendix Fig. 3 for reviewer #1 (showing that the knock-down of ORP2 does not alter the accumulation of accessible cholesterol at the PM and Golgi in QKO cells)

Appendix Fig. 4 for reviewer #2 (showing Manders' coefficient for the colocalization analysis of GRAM-H and ORP9 (HH/AA) or OSBP in QKO cells)

Physiological relevance of the work

Fig. 3g, h [showing that ORP9 KO cells exhibit accumulation of accessible cholesterol at the Golgi using EGFP-tagged tandem GRAM-H (EGFP-GRAM-Hx2)]

Fig. 6g, h [showing the impact of OSBP inhibition (via OSW-1 treatment) or PI4KIII β inhibition (via PIK93 treatment) on the levels of accessible cholesterol, as assessed by EGFP-tagged tandem GRAM-H (EGFP-GRAM-Hx2), at the Golgi in ORP9 KO cells]

Supplementary Fig. 5e, f [showing the impact of PI4KIII β inhibition (via PIK93 treatment) on cholesterol distribution in QKO cells]

Appendix Fig. 1 for reviewer #1 (showing that the overexpression of PI4KIII β alone does not result in the same magnitude of PI4P accumulation as observed in cells lacking ORP9)

Additional quantification as requested by the reviewers

Supplementary Fig. 2a, c, d, e, i (showing Manders' coefficient for the colocalization analysis of various versions of ORP9 with indicated proteins)

Supplementary Fig. 3b (showing Manders' coefficient for the colocalization analysis of mCherry-GRAM-H with indicated Golgi proteins)

Supplementary Fig. 5b [showing Manders' coefficient for the colocalization analysis of iRFP-P4M with indicated Golgi proteins]

Supplementary Fig. 7d, e [showing the representative images and quantification of the binding of GRAMD1b (5P) to the PM in wild-type control and QKO cells as assessed by TIRF microscopy]

Other new data to strengthen our conclusions

Supplementary Fig. 6b, c [showing the efficiency of OSBP depletion by RNAi in QKO cells]

Supplementary Fig. 8e, f [showing the selective decrease in the association of EGFP-GRAM-H to the PM by the expression of PM-targeted GRAMD1b (GRAMD1b fused with the PH domain of PLC δ) in QKO cells]

-Sentences from the reviewers' comments are *in italics*
 -Our responses are in blue

Reviewer #1 (Remarks to the Author):

This work investigated how ORP proteins may control the trafficking of endogenous cholesterol. The authors made use of the TKO cells that accumulate plasma membrane cholesterol because they lack all three GramD1 proteins, and found that ORP9 overexpression can suppress the accumulation of PM cholesterol in these cells. They further demonstrated that QKO cells lacking all GramD1s and ORP9 accumulated more PM and Golgi cholesterol, which was due to increased PI4P on the Golgi and a hyperactive OSBP. Moreover, GramD1 proteins can transfer cholesterol back to the ER from the Golgi in normal cells. Overall, this study proposed an interesting collaborative network that helps maintain Golgi cholesterol and PI4P homeostasis. This is an important contribution to the field of nonvesicular lipid trafficking. The connection between ORPs and GramD1 is highly interesting. However, there are some issues to be addressed and clarified before acceptance.

We thank the reviewer for all the positive and constructive comments to further improve our manuscript.

1. The central conclusion here is that ORP9 controls Golgi PI4P, which then controls OSBP function. If this is the case, then other means that can increase PI4P in the TKO cells should also activate OSBP and causes cholesterol accumulation on the Golgi and PM. This should be tested. For instance, what happens when PI4K3b is overexpressed in the TKO cells? What happens when PI4K3b is silenced or inhibited in the QKO cells?

Reply: We thank the reviewer for these suggestions. We inhibited PI4KIII β using PIK93 (a chemical inhibitor of PI4KIII β) and showed that the levels of cholesterol (as assessed by GRAM-H) at the Golgi and PM of QKO cells were significantly reduced after 1 hour treatment (**new Supplementary Figure 5e, f**). This new result shows that the pool of PI4P generated by PI4KIII β is necessary for driving cholesterol transport to the Golgi in QKO cells. By contrast, overexpression of mCherry-tagged PI4KIII β in TKO cells did not lead to major accumulation of cholesterol at the Golgi and PM. We found that the levels of PI4P in these cells, as assessed by iRFP-P4M, were lower than those in ORP9 KO cells (**please see below the results of these experiments**). These results suggest that the levels of PI4P in the Golgi do not reach a level sufficient to induce detectable cholesterol accumulation in the presence of ORP9. This is likely because ORP9 actively removes excess Golgi PI4P generated by PI4KIII β . Collectively, our results show that the abundance of Golgi PI4P plays a major role in determining the levels of cholesterol in the Golgi.

2. The role of ORP10 and 11 in ORP9 localization and function should be investigated further. Does ORP9 localize to Golgi in ORP10/11 deficient cells?

Are there any synergistic effects on PM cholesterol when ORP9/10/11 are co expressed? (Fig. S1A-1B). These assays would provide more insights into the interaction between ORP9 and 10/11.

Reply: We found that the double knockdown of ORP10 and ORP11 by RNAi did not affect the localization of ORP9 to the TGN (**new Supplementary Figure 2j-l**), suggesting that the complex formation of ORP9, ORP10, and ORP11 is not essential for the recruitment of ORP9 to the TGN in HeLa cells. Inspired by the reviewer's comments, we carried out new experiments and compared the effect of the expression of ORP9 alone to the effect of the co-expression of ORP9 together with either ORP10 or ORP11. We found that co-expression of ORP9 together with either ORP10 or ORP11 in GRAMD1 TKO cells did not result in further decrease in the binding of EGFP-GRAM-H to the PM compared to GRAMD1 TKO cells expressing ORP9 alone (**please see below the results of these experiments**). This may be due to the already strong effect of ORP9 expression to modulate intracellular cholesterol distribution. While this data does not provide evidence that supports synergistic effects of ORP9 and ORP10/11, we do not exclude the possibility that these proteins might function together in a physiological context.

We changed the following sentence in the discussion to be consistent with our new findings (**page 17-18**).

- Original sentence: "Deletion of the tandem α -helices resulted in dissociation of ORP9 from the TGN, suggesting that homo- and hetero-meric interactions between ORP9 and ORP10/11 are important for its TGN localization."
- New sentence: "While simultaneous depletion of ORP10 and ORP11 by RNAi did not result in dissociation of ORP9 from the TGN, deletion of the tandem α -helices resulted in dissociation of ORP9 from the TGN, suggesting that homo- and hetero-meric interactions between ORP9 and ORP10/11 might play some roles in the localization or function of ORP9."

3. Fig. S1A/B, why ORP5 and ORP6 are missing?

Reply: We expressed mRuby-tagged ORP5 and mRuby-tagged ORP6 in TKO cells (as we did for other ORPs) for the relevant experiment. The expression of ORP5 or ORP6 did not affect PM binding of EGFP-GRAM-H. These new results have been added to **Supplementary Figure 1a, b**.

The effect of ORP2 is very strong (stronger than OSBP). Is there any possibility that ORP9 might also modulate ORP2 function?

Reply: We thank the reviewer for this comment. To assess the possibility that ORP9 might modulate the function of ORP2, we co-expressed ORP2 together with ORP9 in GRAMD1 TKO cells and compared the levels of accessible PM cholesterol in these cells and GRAMD1 TKO cells expressing only ORP2. There was no significant difference between the cells expressing ORP2 alone and the cells co-expressing ORP2 together with ORP9 (**new Supplementary Figure 1c, d**). In an additional experiment, we depleted ORP2 by RNAi and assessed its impact on the distribution of accessible cholesterol in QKO cells. Knockdown of ORP2 in QKO cells did not affect the levels of accessible cholesterol at the PM and Golgi (**please see below the results of these experiments**). Collectively, these new results indicate that ORP9 does not play a major role in modulating ORP2 function.

Appendix Fig. 3. A. Lysates of HeLa cells lacking ORP9 and GRAMD1s (QKO) that were treated with indicated siRNA for 72 hrs and processed by SDS-PAGE and immunoblotting (IB) with indicated antibodies. **B.** Quantification of the signals of ORP2 that are normalized with those of actin, as shown in (A). **C.** Confocal images of live QKO HeLa cells expressing EGFP-GRAM-H and mScarlet-Giantin-C (Golgi marker) that were treated with indicated siRNA for 72 hrs before the imaging. Insets show at higher magnification the regions around the PM and the Golgi as indicated by white dashed boxes. Scale bars, 10 μ m. **D.** Quantification of the ratio of PM signals (left) and Golgi signals (right) to the cytosolic signals of EGFP-GRAM-H, as shown in (C) (mean \pm SEM, n = 10 cells for each condition; data are pooled from one experiment).

4. The authors avoided mentioning PS, which is the major ligand for ORP9. Why is that? Would PS play a role in the ORP9/OSBP/GRAMD1 network?

Reply: We thank the reviewer for these comments. To address whether the recently reported property of ORP9 to transport PS (PMID: 36071159) might play a role in cholesterol distribution (mediated by the interplay between ORP9, OSBP, and GRAMD1s), we expressed PS binding deficient mutant of ORP9 [ORP9 (AAA)] (PMID: 36071159) in QKO cells and examined whether it would still rescue the cholesterol distribution defects in these cells or not. Expression of mCherry-tagged ORP9 (AAA) restored normal cholesterol distribution in QKO cells (**new Figure 4d, e**), suggesting that PS transport property of ORP9 is not required for the regulation of the ORP9/OSBP/GRAMD1 network.

5. The direct role of GramD1 at the Golgi may need more evidence. For instance, the effect on Golgi cholesterol upon GramD1b re-expression might be indirect: it depletes PM cholesterol which allows more Golgi cholesterol to reach the PM (Fig. 7A).

Reply: We thank the reviewer for this constructive suggestion. Data in our original manuscript show that the acute recruitment of GRAMD1b to the TGN (via rapamycin-mediated heterodimerization of a chimeric GRAMD1b and a TGN protein) rapidly reduce the levels of cholesterol in the Golgi in QKO cells (as assessed by GRAM-H) (**Figure 7e, f**). To exclude the possibility that GRAMD1b depletes Golgi cholesterol indirectly by functioning at ER-plasma membrane (PM) contact sites, we expressed in QKO cells a version of mRuby-GRAMD1b, whose GRAM domain was replaced by the PH domain of PLC δ (i.e., PM-interacting module) (mRuby-PH-GRAMD1b) to induce its constitutive localization to ER-PM contact sites. QKO cells expressing mRuby-PH-GRAMD1b showed significant reduction in the levels of accessible cholesterol in the PM, as assessed by EGFP-GRAM-H, compared to control QKO cells expressing mRuby. By contrast, they showed only minor reduction in the levels of accessible cholesterol in the Golgi compared to control QKO cells expressing mRuby (however, this reduction is likely due to increased levels of cytosolic GRAM-H signals in this condition) (**new Supplementary Figure 8e, f**). These results suggest that GRAMD1 needs to function at ER-Golgi contacts to extract Golgi cholesterol efficiently and provide additional evidence to support the direct role of GRAMD1b in extracting cholesterol from the Golgi.

In Fig. 7C, the 5P mutant accumulated on the Golgi but not the PM. Why is that?

Reply: We thank the reviewer for this comment. GRAMD1b 5P mutant indeed accumulates on both the PM and the Golgi in QKO cells. We realized that PM accumulation of GRAMD1b 5P mutant is less obvious in images taken under confocal microscopy. To obtain more quantitative data for the PM accumulation of GRAMD1b (5P) in QKO cells (compared to wild-type control cells), QKO cells expressing mRuby-GRAMD1b (5P) were observed under TIRF microscopy. Under TIRF microscopy, it was evident that QKO cells showed much enhanced PM recruitment of mRuby-GRAMD1b (5P) compared to WT HeLa cells (**new**

Supplementary Figure 7d, e), providing additional evidence that mRuby-GRAMD1b (5P) accumulates at the PM in QKO cells. We also indicated accumulation of GRAMD1b (5P) in the PM of QKO cells by yellow arrowheads in **Figure 7c**.

6. *In many figures including 2C-G, S2A, 3G, S5A, there is a single image of an apparent morphology/phenotype (which presumably applies to the whole cell population) without any quantification, number of cells or biological replicates in which this was observed, or correlation analyses. It would be preferable for the authors to provide correlations or quantitation, like in Figures 4E and 6B, to demonstrate these findings more objectively.*

Reply: We thank the reviewer for this constructive suggestion. We measured Manders' overlap coefficient using JACoP plugin in FIJI (PMID: 17210054) and added the graphs for Figures 2C-G, S2A, 3G, and S5A. They are in **new Supplementary Figure 2a, c, d, e, and I, new Supplementary Figure 3b, and new Supplementary Figure 5b**.

7. *The mechanism behind the increase in PM cholesterol with ORP9 KO in the TKO cells is unclear, given endogenous OSBP is not thought to transport cholesterol to the PM. The authors mention the role of CERT in the discussion as the factor underlying the TGN-PM transport of cholesterol in the QKO line. Clarification of this would be beneficial to the impact of the study and could be investigated with the depletion of CERT in the QKO line.*

Reply: We thank the reviewer for these insightful comments. Our current study shows that non-vesicular lipid transport at ER-Golgi contacts regulates the abundance of cholesterol in the Golgi and that such regulation is important for maintaining the levels of cholesterol in the PM. We would like to clarify the function of CERT that we mentioned in the discussion of our original manuscript. A previous study by Anwar et al. (PMID: 36174556) showed that a super complex of CERT, OSBP, and VAP functions at ER-Golgi contacts and "indirectly" influence the levels of cholesterol in the PM. They did not show that CERT is directly involved in transporting cholesterol from the TGN to the PM. We apologize for the confusion; we now state this point more clearly in the discussion. Previous literatures suggest the importance of non-vesicular transport for the transport of cholesterol to the PM (e.g., PMID: 15823040; PMID: 10890900; PMID: 29783105); however, the identity of lipid transfer proteins involved in Golgi to PM transport remains unclear. Hence, further investigation is needed to elucidate the exact mechanism that regulates cholesterol transport between the Golgi and the PM; investigating this process is beyond the scope of the current study.

We changed the following sentence in the discussion to avoid confusion (**page 18**).

- Original sentence: "In agreement with this function, a recent study showed that formation of a complex between OSBP and ceramide transfer protein (CERT), which transports ceramide from the ER to the TGN, at ER-TGN contact sites regulates transport of cholesterol to the PM for the assembly of cholesterol nanodomains (Anwar et al., 2022)."
- New sentence: "In agreement with this function of OSBP, a recent study showed that a super complex of OSBP, ceramide transfer protein (CERT), and VAP functions at ER-TGN contact sites and influences the assembly of cholesterol nanodomains in the PM (Anwar et al., 2022)"

We also inserted the following new sentence in the discussion to discuss this point more clearly (**page 18**).

- "Exactly how Golgi cholesterol is transported to the PM remains unclear. While vesicular transport may mediate transport of some cholesterol from the Golgi to the PM, previous studies indicate the importance of non-vesicular transport for cholesterol transport to the PM (Baumann et al., 2005; Heino et al., 2000; Menon, 2018). Further investigation is needed to elucidate the mechanism of regulated transport of cholesterol between the Golgi and the PM."

8. *In Figure 3B there is still a lot of signal in the immunostaining of ORP9 in the ORP9 KO cell line, is this thought to be non-specific staining? The authors should comment on this.*

Reply: Yes, it is non-specific staining that results from the anti-ORP9 antibodies. We mentioned this in a relevant part of the Figure legend.

9. *It would be useful to have the concentration and duration of OSW-1 treatment in the text and/or figure legends.*

Reply: Thank you for the suggestion. We added the information of the drug treatment in the relevant part of the main text as well as in the relevant part of the Figure legends.

Reviewer #2 (Remarks to the Author):

In Naito et al., authors show that two membrane contact site proteins, OSBP and ORP9, localizing between the ER and the Golgi regulate cholesterol dynamics in GRAMD1 knockout cells. Authors show that ORP9 and OSBP over-expression can rescue and elevate phenotype caused by the loss of GRAMD1. Furthermore, loss of ORP9 furthers the phenotype of cholesterol accumulation at the plasma membrane and causes cholesterol accumulation in the Golgi apparatus, latter of which is caused by OSBP. I find authors observations solid and there is some novelty to their findings. However, I find the manuscript insufficient for publications due to various issues. One of the issues is that most of the observations are done in GRAMD1 triple and GRAMD1-ORP9 quadruple knockout cells. This raises the question whether this is biologically any significant has any biological meaning, i.e. what observed is most likely a compensatory mechanism to what might never occur in nature. Is there is a condition, e.g. starvation or virus infection, where ORP9 in addition to all three GRAMD1 proteins are somehow inhibited that might lead to increased Golgi cholesterol levels. A list of issues are listed below:

We thank the reviewer for all the positive and constructive comments to improve our manuscript.

Main Issues:

1) Main issue is that the majority of the conclusions are based on the observations made in GRAMD1 triple knockout (TKO) cells, which led to the characterizing a role for OSBP and ORP9 in cholesterol distribution in these TKO cells. While this appears to be a compensatory mechanism in TKO cells, whether this takes places in normal conditions, i.e. has a biological significance, is a big question mark to be addressed.

Reply: We thank the reviewer for these insightful comments; we acknowledge that the majority of the conclusions in our original manuscript were based on our observations in GRAMD1 TKO cells. To provide evidence that ORP9 functions to regulate cholesterol distribution in normal conditions (where GRAMD1s are present), we re-examined the cholesterol distribution in ORP9 single KO cells using tandem GRAM-H (EGFP-GRAM-Hx2), a more sensitive version of a cholesterol biosensor compared to the original EGFP-GRAM-H. Remarkably, EGFP-GRAM-Hx2 was significantly more associated with the Golgi in ORP9 single KO cells (compared to wild-type control HeLa cells) even without additional depletion of GRAMD1s, suggesting that cholesterol accumulates at the Golgi in ORP9 single KO cells (**new Figure 3g, h**). Further, we showed that this cholesterol accumulation was suppressed by inhibition OSBP (via OSW-1) or PI4KIII β (via PIK93) (**new Figure 6g, h**). These results support the role of ORP9 in controlling PI4P levels in the Golgi and contributing to the regulation of OSBP-mediated cholesterol transport to the Golgi. This is also supported by data in our original manuscript shown in **Figure 8a-e**, where ORP9 single KO cells showed moderate, but significant, increase in SREBP-2 activity compared to control HeLa cells. Collectively, our results demonstrate that ORP9 functions to control cholesterol distribution and homeostasis even in the background, where GRAMD1s are present (i.e., normal conditions), providing a biological significance to our study. We have not been able to identify a condition, where ORP9 function is inhibited. Identifying such a condition (e.g., disease-relevant contexts etc.) will require further investigation.

We changed the following sentence in the abstract to be consistent with our new findings (**Page 3**).

- Original sentence: “Accordingly, cells lacking both ORP9 and GRAMD1s, but not ORP9 alone, exhibit dramatic accumulation of cholesterol at the Golgi and post-Golgi membranes, including the plasma membrane. Importantly, this is accompanied by chronic activation of the SREBP-2 signalling pathway and increased production of cholesterol.”
- New sentence: “Cells lacking ORP9 exhibit accumulation of cholesterol at the Golgi, which is further enhanced by additional depletion of GRAMD1s with major accumulation in the plasma membrane. This is accompanied by chronic activation of the SREBP-2 signalling pathway.”

We also changed the following sentence in the discussion (**Page 17**).

- Original sentence: “While depletion of ORP9 alone is sufficient for PI4P and OSBP to accumulate at the TGN, dramatic accumulation of accessible cholesterol occurs in the Golgi only when ORP9 and GRAMD1s are simultaneously depleted.”

- New sentence: “Depletion of ORP9 results in accumulation of PI4P and OSBP at the TGN and causes moderate accumulation of accessible cholesterol in the Golgi. Such accumulation is further enhanced when ORP9 and GRAMD1s are simultaneously depleted.”

Finally, we updated relevant sentences throughout the manuscript to be consistent with our new findings.

2) Page 2, Highlight nr. 1: “ORP9 ... [is]... a key regulator of intracellular cholesterol distribution.” & Page 10, Line 1: “...ORP9 is important for proper distribution of cholesterol in the Golgi and post Golgi membranes, including the plasma membrane”.

This is inaccurate. Authors show that this is only the case when all three GRAMD1 proteins are dysfunctional, which itself already causes cholesterol deregulation, knockout of ORP9 only contributes to it. ORP9 knockout cells show no changes in intracellular cholesterol distribution (figure 3).

Reply: As we mentioned above for this reviewer’s comment #1, we now show in the revised manuscript that ORP9 single KO cells show changes in intracellular cholesterol distribution (i.e., moderate accumulation of cholesterol in the Golgi). We thank the reviewer for these important comments.

3) Page 2, Highlight nr. 2: “Cells lacking ORP9 and GRAMD1s show massive cholesterol accumulation”.

Massive is very subjective. TKO cells already have elevated plasma membrane cholesterol levels and loss of ORP9 only contributes to this. More importantly, figure 8F shows only 10-20% increase in total cholesterol levels, so it is hardly “massive”.

Reply: We toned down the statement by changing the word “massive cholesterol accumulation” to “accumulation of cholesterol” for the highlights and for relevant sentences throughout the manuscript.

4) Page 2, Highlight nr. 3: “ORP9 ... inhibits OSBP-driven cholesterol transport to the TGN”.

The findings are too little to conclude that ORP9 blocks OSBP-driven cholesterol transport, neither in TKO nor wild-type cells. The results only lead to the conclusion that OSBP is responsible for the cholesterol accumulation in Golgi of QKO cells.

Reply: We thank the reviewer for these comments. We showed in our original manuscript that PI4P accumulated at the TGN in the absence of ORP9 (**Figure 5a, b**). We also showed that this was accompanied by OSBP accumulation at the TGN (**Figure 6a, b**). In the revised manuscript, we additionally show that cholesterol accumulates at the Golgi in ORP9 single KO cells (**new Figure 3g,h**) and that such cholesterol accumulation in ORP9 single KO cells (as well as in QKO cells) was suppressed by inhibition of either PI4KIIIβ or OSBP (**new Figure 6g,h, new supplementary figure 5e, f, Figure 6e-f**). Collectively, these results suggest that ORP9 inhibits the activity of OSBP by regulating the levels of PI4P in the TGN.

We acknowledge that the role of ORP9 in regulating OSBP-mediated cholesterol transport is not direct; to avoid confusion, we changed the relevant sentence in the highlights as show below.

- Original sentence: “ORP9 extracts TGN PI4P and inhibits OSBP-driven cholesterol transport to the TGN.”
- New sentence: “OSBP-driven cholesterol transport to the TGN is responsible for cholesterol accumulation in these cells.” “ORP9 extracts PI4P from the TGN and contributes to the regulation of OSBP function.”

We also changed relevant sentences throughout the manuscript to be consistent with these points.

5) Some aspects of the paper seem redundant to what is described in the field or within itself:

- Figure 2 showing ORP9 localization and its known interaction with ORP10 and ORP11 (Zhou 2010; Finkel 2022).

Reply: While the interaction between ORP9 and ORP10/11 was previously reported, this manuscript is the first to report the role of the tandem alpha-helices within ORP9 in the localization of ORP9 to the TGN (which we identified in the current study). In brief, we show the following in **Figure 2**: 1) The PH domain of ORP9

interacts with both the PM and the TGN; 2) When the tandem alpha-helices are added to the PH domain, the PH domain is exclusively targeted to the TGN, mimicking the endogenous localization of ORP9; 3) Further, the tandem alpha-helices alone are sufficient to localize to the TGN. These data provide novel insights into the mechanism of the selective localization of ORP9 to the TGN at steady state.

- Both figure 4 and 5 are showing that ORP9 can extract PI4P.

Reply: In **Figure 4**, we show that ORP9 can extract PI4P and that such property is important for the regulation of intracellular cholesterol distribution. In **Figure 5**, we further show that acute recruitment of ORP9 to the Golgi (to allow it to extract PI4P specifically from the Golgi membranes) is sufficient to reduce of the levels of cholesterol in the Golgi. Therefore, these two Figures are rather complementary.

6) I find it risky that the study mainly uses the GRAMD1-based cholesterol sensor for detecting changes in the cholesterol distribution caused by GRAMD1-depletion, especially in the figures 6 and 7. D4H, another cholesterol binding protein, is also used but only to a limited extent. Many of the cholesterol experiments must be repeated using D4H to support conclusions.

Reply: We thank the reviewer for this comment. We repeated all the relevant experiments by expressing an alternative accessible cholesterol biosensor [EGFP-tagged D4H (EGFP-D4H)] and provided additional support to the abnormal cholesterol distribution in our KO cells. First, the level of D4H binding to the PM was significantly enhanced in GRAMD1 TKO cells as well as in QKO cells compared to wild-type control (**new Supplementary Figure 3e, f**). In QKO cells, EGFP-D4H was additionally accumulated around the Golgi (**new Supplementary Figure 3e, f**). These phenotypes were rescued by re-expression of wild-type ORP9 or wild-type GRAMD1b, but not by mutant versions of these proteins (**new Supplementary Figure 4c, d** for ORP9; **Supplementary Figure 7a, b** for GRAMD1b). Furthermore, the accumulation of EGFP-D4H in the Golgi and PM of QKO cells was significantly suppressed by additional depletion of OSBP by RNAi (**new Supplementary Figure 6d, e**, related to **Figure 6c, d**), and the accumulation of EGFP-D4H in the Golgi was rapidly suppressed by chemical inhibition of OSBP by OSW-1 (**new Supplementary Figure 6h, i**, related to **Figure 6e, f**). Finally, the accumulation of EGFP-D4H in the Golgi was rapidly suppressed by acute recruitment of GRAMD1b to the TGN via the rapamycin-mediated heterodimerization method (**new Supplementary Figure 8a, b**, related to **Figure 7e, f**).

We also found that Golgi association of EGFP-D4H was slightly elevated in ORP9 single KO cells compared to wild-type control cells (**new Supplementary Figure 3e, f**); however, this was not statistically significant. Unlike tandem GRAM-H (EGFP-GRAM-Hx2) (**new Figure 3g, h**), EGFP-D4H tends to form aggregates in cytosol. We think that this property of D4H makes it difficult to detect mild changes in the levels of Golgi cholesterol as observed using EGFP-GRAM-Hx2 in ORP9 single KO cells.

Taken together, these new results (obtained by using EGFP-D4H) support our conclusion that ORP9, GRAMD1s, and OSBP contribute to the maintenance of the distribution of cholesterol in the cell.

7) Cholesterol is most abundant in the extracellular leaflet of the plasma membrane and in the Golgi is accepted to accumulate at the luminal side of the membrane. Both D4H experiments (figure S3) and GRAMH experiments suggest that both cytosolic and extracellular leaflets of the plasma membrane have increased cholesterol levels. What is the authors' explanation to accumulation to the cytosolic leaflet?

Reply: D4H and GRAM-H specifically detect the pool of accessible (or chemically active) cholesterol in cellular membranes (e.g., PMID: 36791915). Accessible cholesterol is thought to spontaneously flip-flop across bilayer membranes (e.g., PMID: 11566796; PMID: 29896788). The remaining pool of cholesterol, known as inaccessible (or chemically inactive) pool of cholesterol, is forming complexes with other membrane lipids, including sphingomyelin and phospholipids. We understand that the reviewer might be referring to the inaccessible pool of cholesterol in this comment. Indeed, inaccessible pool of cholesterol is likely enriched in the extracellular leaflet of the plasma membrane bilayer and the luminal leaflet of the Golgi membranes due to the abundance of sphingomyelin in these leaflets (e.g., PMID: 32367017; PMID18216768). When there is an increase in the levels of cholesterol beyond the sequestration capacity of sphingomyelin and phospholipids in cellular membranes, there is an expansion of the accessible pool of cholesterol. This results in increased binding of cytosolically expressed D4H or GRAM-H to cytosolic leaflets of cellular membranes (because newly expanded pool of accessible cholesterol can spontaneously flip-flop across bilayer membranes).

8) *Figure 8 is not convincing enough to explain the difference in cholesterol levels. It appears that the manuscript is rather incomplete.*

- There is no cholesterol increase in TKO cells and QKO have about 10-20%.

Reply: We thank the reviewer for these comments. We previously reported that GRAMD1 TKO cells do not show changes in the levels of total cholesterol despite accumulation of accessible cholesterol in the PM (which is caused by inefficient cholesterol transport to the ER in these cells) (Naito et al., *eLife*, 2019, PMID: 31724953, Figure 4-figure supplement 3). Therefore, it seems that cells can adapt to a certain degree of increase in the accessible pool of PM cholesterol and maintain total levels of cellular cholesterol. By contrast, QKO cells, lacking both GRAMD1s and ORP9, show aberrant activation of SREBP-2 and increased production of cholesterol (shown in our original **Figure 8a-c**). This is likely caused by combined effects of inefficient cholesterol transport to the ER (due to the depletion of GRAMD1s) and OSBP hyperactivation (due to the depletion of ORP9). Thus, increase in total cellular cholesterol is a unique phenotype of QKO cells, which we do not observe in GRAMD1 TKO cells.

- SQLE, HMGCR, HMGCS1, DHCR24 are upregulated in ORP9 KO cells without affecting cholesterol levels or distribution.

Reply: As we mentioned above for this reviewer's comment #1, we now show in the revised manuscript that ORP9 single KO cells show moderate accumulation of cholesterol in the Golgi (**new Figure 3g, h**). We also show that this cholesterol accumulation was suppressed by inhibition of OSBP, consistent with our model where OSBP is hyperactivated in the absence of ORP9 (**new Figure 6g, h**). As GRAMD1 TKO cells do not show upregulation of SREBP-2 signalling pathways, accumulation of accessible cholesterol in the Golgi resulting from OSBP hyperactivation in ORP9 KO cells, seems to be the major root cause for the moderate activation of SREBP-2 in these cells (as reflected by upregulation of SQLE, HMGCR, HMGCS1, DHCR24 as the reviewer pointed out).

We revised the following statement in the discussion to clarify these points for the reviewer's comment #8 (**Page 19**).

- Original sentences: "When LTP-mediated cholesterol transport is dysregulated at ER-TGN contacts, accessible cholesterol is abnormally distributed to the Golgi and post-Golgi membranes, including the PM. Importantly, this abnormal distribution of cholesterol leads to dysregulation of the SREBP2 signalling pathway and aberrant cholesterol production. Thus, intricate crosstalk between the ER and TGN, which is regulated by these three LTPs at ER-TGN contacts, plays a critical role in determining cellular cholesterol distribution and maintaining normal cell physiology."
- New sentences: "When LTP-mediated cholesterol transport is severely dysregulated at ER-Golgi contacts as in the case of QKO cells, accessible cholesterol is abnormally distributed to the Golgi and post-Golgi membranes, including the PM. Importantly, this abnormal distribution of cholesterol is coupled with dysregulation of the SREBP2 signalling pathway and aberrant cholesterol production. Interestingly, cells lacking only ORP9 or only GRAMD1s do not show major changes in total levels of cellular cholesterol despite some accumulation of accessible cholesterol in the Golgi or the PM, respectively. Therefore, it seems that cells adapt to a certain degree of increase in the accessible pool of cholesterol in cellular membranes and maintain total levels of cellular cholesterol. The aberrant activation of SREBP-2 and increased levels of total cellular cholesterol in QKO cells are likely due to combined effects of OSBP hyperactivation (due to the depletion of ORP9) and inefficient cholesterol transport to the ER (due to the depletion of GRAMD1s). As depletion of ORP9 alone leads to moderate upregulation of SREBP-2 signalling pathways, the activity of OSBP seems to be tightly linked to that of SREBP-2. When hyperactivation of OSBP is coupled with inefficient cholesterol transport to the ER (as in the case of QKO cells), SREBP-2 is hyperactivated, resulting in dysregulation of cholesterol production. Thus, our study suggests that intricate crosstalk between the ER and Golgi, which is regulated by ORP9, OSBP, and the GRAMD1s at ER-Golgi contacts, plays a critical role in determining cellular cholesterol distribution and maintaining normal cell physiology."

9) Authors nicely show that the trans Golgi have various subdomains. I do not know how this is relevant to the rest of the story. Regardless, I wonder what the location of cholesterol accumulation in QKO is. Is it the ORP9 or OSBP subdomain?

Reply: We thank the reviewer for these comments. To assess the location of cholesterol accumulation in QKO cells, we measured Manders' overlap coefficient between various Golgi marker proteins and GRAM-H using JACoP plugin in FIJI (PMID: 17210054) and added the graphs in **new Supplementary Figure 3b**. GRAM-H signals overlapped with multiple markers of the Golgi, suggesting that the accumulation of cholesterol occurs throughout the wide regions of the Golgi, including cis-medial Golgi, trans-Golgi, and TGN, in QKO cells. We also checked the co-localization between GRAM-H and ORP9 (or OSBP) in QKO cells (**please see below the results of these experiments**). Because re-expression of wild type ORP9 restores normal cholesterol distribution in QKO cells, we expressed mCherry-tagged ORP9 (HH/AA), which cannot transport PI4P, together with GRAM-H in QKO cells. mCherry-ORP9 (HH/AA) signals partially overlapped with GRAM-H signals. To assess whether cholesterol accumulates at the subdomain of the Golgi where endogenous OSBP is present, we performed immunolabeling of OSBP using anti-OSBP antibodies in QKO cells expressing GRAM-H and found that some GRAM-H signals partially overlapped with OSBP. Collectively, the cholesterol accumulation in QKO cells appears to occur throughout the Golgi, rather than being located in specific subdomains of the Golgi.

Minor Issues:

1) If OSBP responsible for cholesterol accumulation in Golgi of TKO cells, would not its over-expression in figure S1A cause cholesterol accumulation in Golgi?

Reply: We thank the reviewer for this comment. To our surprise, we did not observe accumulation of cholesterol in the Golgi by overexpression of OSBP. ORP9 depletion leads to accumulation of PI4P in the Golgi (which in turn activates OSBP). On the other hand, OSBP overexpression leads to consumption of PI4P in the Golgi (thus generally reducing the levels of PI4P in the Golgi). Therefore, the impact of ORP9 depletion and OSBP overexpression on the Golgi is not identical. Based on our results, PI4P levels in the Golgi seem to be critical for determining the extent of cholesterol accumulation in the Golgi, as acute reduction of Golgi PI4P in ORP9 single KO cells or QKO cells (by inhibition of PI4KIIIβ via PIK93 or by acute recruitment of Sac1 PI4P phosphatase) results in decrease in the levels of cholesterol in the Golgi (**new Figure 6g,h, new supplementary figure 5e, f, and Figure 5e-f**). Further studies are needed to better understand the role of PI4P in this process.

Over-expression of ORP2 does it to an extent, at least some place in the cytoplasm.

Reply: We would like to clarify this point for the reviewer. We sometimes observe minor accumulation of EGFP-GRAM-H in vesicular structures in cytoplasm. Overexpression of ORP2 does not result in accumulation of cholesterol in the Golgi in TKO cells. To avoid confusion, we replaced the representative image for ORP2 overexpression in **Supplementary Figure 1a**.

2) "GRAMD1s extract cholesterol from the TGN to prevent its build up in the Golgi." They show by live cell imaging, which is very nice.! It can extract from Golgi in QKO, but does it ever occur in normal conditions?

Reply: We thank the reviewer for these comments. As we mentioned above for this reviewer's comment #1, we now show in the revised manuscript that ORP9 single KO cells show moderate accumulation of cholesterol in the Golgi (**new Figure 3g, h**). Importantly, such accumulation of Golgi cholesterol was further enhanced in QKO cells (**new Figure 3g, h, Figure 3c, new Supplementary Figure 3e, f**), suggesting that GRAMD1s extract cholesterol from the Golgi to prevent the build-up of Golgi cholesterol when OSBP is hyperactivated. These results suggest that OSBP and GRAMD1s maintain the levels of cholesterol in the Golgi, by transport (via OSBP) and extraction (via GRAMD1s), in normal conditions.

3) *Figure 7A is not uniform ER distribution!*

Reply: We thank the reviewer for this comment. Indeed, we previously showed that GRAMD1s localize to the ER as distinct puncta (Naito et al., *eLife*, 2019, PMID: 31724953). We rephrased the relevant sentence (**Page 14**) as indicated below.

- Original sentence: "Interestingly, mRuby-GRAMD1b (5P) accumulated at the Golgi, whereas mRuby-GRAMD1b and mRuby-GRAMD1b (R189W & 5P) were distributed uniformly throughout the ER (**Figures 7A and S7A**)."
- New sentence: "Interestingly, mRuby-GRAMD1b (5P) accumulated at both the Golgi and the PM, whereas mRuby-GRAMD1b and mRuby-GRAMD1b (R189W & 5P) were distributed as distinct puncta throughout the ER (**Fig. 7a, c and Supplementary Fig. 7c**)."

4) *Why use purified EGFP-D4H for permeabilized and mCherry-D4H for impermeabilized cells?*

Reply: In the original manuscript, purified EGFP-D4H proteins were used to perform multi-colour imaging with mCherry-GRAM-H. In the revised manuscript, we replaced the relevant figure with new figures with cytosolically expressed EGFP-D4H (**new Supplementary Figure 3e, f**) as we mentioned above for this reviewer's comments #6.

Reviewer #3 (Remarks to the Author):

This is an encyclopedic paper, with a large number of technically sophisticated experiments that make use of a huge number of bespoke reagents and knock-out cell lines. The sheer volume of minutiae makes the paper very difficult to read. Indeed, without consulting the graphical abstract and the summary at the start of the Discussion, it is next to impossible to track the logical flow of the paper to try to understand the experiments and their conclusions.

We thank the reviewer for all the constructive comments to further improve our manuscript.

The experiments are set up and viewed through the singular lens of the PI4P/sterol exchange model, focused on contact sites, without reference to the broader literature on intracellular cholesterol dynamics and distribution. For example, the literature is clear that cholesterol equilibrates between the ER and PM such that its chemical activity is similar in both compartments (Kaplan and Simoni, JCB 1985; Baumann et al. Biochemistry 2005, Menon Curr Opin Cell Biol 2018). Thus, an excess above the sequestering capacity of the PM is dissipated to the ER to restore the chemical activity balance. This can be achieved by any LTP, regardless of whether it is at a contact site. For example, the diffusible, ubiquitously abundant LTP StARD4 could carry out this role (Iaea, Maxfield et al. Mol Biol Cell 2017) but is not referred to anywhere in the manuscript. Indeed, it is surprising to see the specific effects that are reported in the current manuscript in the face of what is surely a high LTP activity provided by StARD4.

Reply: We thank the reviewer for these insightful comments. We agree with the reviewer that numerous literatures (including the ones referred by this reviewer) support the model, whereby changes in the chemical activity (or the accessibility) of cholesterol in the PM leads to shuttling of chemically active cholesterol (or accessible cholesterol) between the PM and the ER to maintain the similar chemical activity of cholesterol in these cellular compartments. While previous studies suggest that lipid transfer proteins (LTPs), such as StARD4 and ORPs, act redundantly to mediate accessible cholesterol transport in this process (as depletion of one particular type of LTPs often does not result in major defects in cholesterol transport/distribution) (reviewed in PMID: 29783105; PMID: 27421092 etc.), several labs, including the Tontonoz lab, the Brown and Goldstein lab, and our lab, recently showed in mammalian cells that GRAMD1s/Asters play a critical role in transporting accessible cholesterol from the PM to the ER. Based on the properties of other LTPs, including ORPs, to transport accessible cholesterol, GRAMD1s and ORPs might act together to maintain intracellular cholesterol distribution. This possibility (also being inspired by previous literatures in the field) led us to investigate the functional interplay of GRAMD1s and ORPs in regulating accessible cholesterol distribution. In our manuscript, we used multiple knock-out cell lines and a wide range of approaches to address this question and showed that GRAMD1s, ORP9, and OSBP act together at ER-Golgi contact sites to regulate the abundance of cholesterol in the Golgi, thereby contributing to the maintenance of cellular cholesterol distribution. Thus, our findings are complementary to previous literatures, supporting the complexity of the orchestrated functions of multiple LTPs in cells and providing new insights into the process of intracellular accessible cholesterol distribution. Our manuscript is not against, by any means, the seminal studies on StARD4. As the main focus of our current study has been on GRAMD1s and ORPs, we focused on literatures on these LTPs. We apologize the reviewer for not citing the important literatures on StARD4 in our original manuscript; we now cite several papers on StARD4, including the one suggested by the reviewer, as well as other papers on cholesterol equilibration highlighted by the reviewer. We agree with this reviewer that how StARD4 and other LTPs might collaborate with GRAMD1s and ORPs to regulate cellular cholesterol distribution requires further investigation.

We reordered several key sentences in the introduction to more broadly discuss the role of various LTPs in non-vesicular cholesterol transport in the cell and revised the following sentences (**Page 4**).

- Original sentence #1: Thus, intracellular cholesterol transport machineries must deliver cholesterol to the ER so that SREBPs can monitor cholesterol levels and adjust its synthesis and uptake.
- New sentence #1: Thus, cells must deliver cholesterol to the ER so that SREBPs can monitor cellular cholesterol levels and adjust its synthesis and uptake. Non-vesicular transport of cholesterol facilitated by lipid transfer proteins (LTPs) plays a critical role in this process (Baumann et al., 2005; Iaea et al., 2017; Iaea and Maxfield, 2015; Kaplan and Simoni, 1985; Menon, 2018).

- Original sentence #2: “However, when levels of PM cholesterol increase beyond the sequestration capacity of other membrane lipids, the excess pool of accessible cholesterol is transported to the ER via LTPs.”
- New sentence #2: “However, when levels of PM cholesterol increase beyond the sequestration capacity of other membrane lipids, the excess pool of accessible cholesterol is transported to the ER via LTPs to restore the balance of chemical activity between these two cellular compartments (Menon, 2018).”

We also revised the following sentence in the introduction (**Page 4**)

- Original sentence: “Evolutionarily conserved and ER-anchored LTPs, namely the GRAMD1s/Asters (GRAMD1a/Aster-A, GRAMD1b/Aster-B, and GRAMD1c/Aster-C), form homo- and hetero-meric complexes and play major roles in this process (Naito and Saheki, 2021).”
- New sentences: “Numerous LTPs have been suggested to participate in this process (Ikonen and Zhou, 2021; Kennelly and Tontonoz, 2022; Maxfield et al., 2016; Menon, 2018). Some of these LTPs function primarily at membrane contact sites, where the ER forms close contact with other organelles and membranes (Arora et al., 2022; Kennelly and Tontonoz, 2022; Koh and Saheki, 2021; Luo et al., 2019a; Naito and Saheki, 2021). Growing evidence suggest that evolutionarily conserved and ER-anchored LTPs, namely the GRAMD1s/Asters (GRAMD1a/Aster-A, GRAMD1b/Aster-B, and GRAMD1c/Aster-C), form homo- and hetero-meric complexes and play major roles in accessible cholesterol transport to the ER at membrane contact sites (Kennelly and Tontonoz, 2022; Naito and Saheki, 2021).”

Furthermore, we added the following sentence in the discussion (**Page 18**)

- “Future studies are needed to better understand how these LTPs functionally cooperate with other cholesterol-harboring LTPs, including StARDs (Alpy and Tomasetto, 2014; Iaea et al., 2017) and other members of ORPs, for accessible cholesterol transport and distribution.”

We also updated the relevant sentences throughout the manuscript to be consistent with these points.

The statement in the Introduction that the PM contains most of the cholesterol content of the cell, estimated at 90% of cell cholesterol, is wrong. This statement is often presented in papers and review articles, most recently by Kennelly and Tontonoz (2023), but is wrong, nevertheless. A simple calculation demonstrates this. In a hepatocyte, the relative areas of PM and ER are 2 and 50% of total cell membrane (<http://book.bionumbers.org/how-big-is-the-endoplasmic-reticulum-of-cells/>), respectively and the cholesterol concentrations at the PM and ER are 40 and 5 mole percent, respectively. Assuming for the moment that there are no other relevant membranes to consider, then the relative amount of cholesterol mass in the PM and ER is $40 \times 2 = 80$ and $5 \times 50 = 250$, respectively. PM cholesterol is therefore $100 \times 80 / (80 + 250)$, i.e., <25% of cell cholesterol. If there is cholesterol in other membranes - such as mitochondria and the endocytic recycling compartment, as will inevitably be the case, then the proportion of cholesterol in the PM is even less. This misstatement of experimental information is a poor starting point for this paper.

Reply: We thank the reviewer for these comments. We understand that there are conflicting data for regarding the amounts of cholesterol present in the PM compared to other cellular membranes, including the ER, in the field. We removed the sentences that specifically state that 90% of cellular cholesterol is present in the PM from the introduction as well as from the rest of the manuscript.

We revised the following sentence in the introduction (**Page 4**).

- Original sentence: “Regardless of the source, up to 90% of cellular cholesterol is then delivered to the PM (de Duve, 1971; Lange et al., 1989; Ray et al., 1969).”
- New sentence: “Regardless of the source, cellular cholesterol is then delivered to the PM, where cholesterol contributes to almost half of the total lipids in this bilayer (de Duve, 1971; Lange et al., 1989; Ray et al., 1969).”

A note about the use of the GRAM sensor - the authors should be aware of the cautionary work of Courtney, Zha et al. eLife 2018 where it is made clear that many factors, including high protein concentrations such as found in the cytoplasm, can affect the readout.

Reply: We thank the reviewer for this comment. We repeated all the relevant experiments by expressing an alternative accessible cholesterol biosensor [EGFP-tagged D4H (EGFP-D4H)] and provided additional support to the abnormal cholesterol distribution in our KO cells. First, the level of D4H binding to the PM was significantly enhanced in GRAMD1 TKO cells as well as in QKO cells compared to wild-type control (**new Supplementary Figure 3e, f**). In QKO cells, EGFP-D4H was additionally accumulated around the Golgi (**new Supplementary Figure 3e, f**). These phenotypes were rescued by re-expression of wild-type ORP9 or wild-type GRAMD1b, but not by mutant versions of these proteins (**new Supplementary Figure 4c, d** for ORP9; **Supplementary Figure 7a, b** for GRAMD1b). Furthermore, the accumulation of EGFP-D4H in the Golgi and PM of QKO cells was significantly suppressed by additional depletion of OSBP by RNAi (**new Supplementary Figure 6d, e**, related to **Figure 6c, d**), and the accumulation of EGFP-D4H in the Golgi was rapidly suppressed by chemical inhibition of OSBP by OSW-1 (**new Supplementary Figure 6h, i**, related to **Figure 6e, f**). Finally, the accumulation of EGFP-D4H in the Golgi was rapidly suppressed by acute recruitment of GRAMD1b to the TGN via the rapamycin-mediated heterodimerization method (**new Supplementary Figure 8a, b**, related to **Figure 7e, f**). Taken together, these new results (obtained by using EGFP-D4H) support our conclusion that ORP9, GRAMD1s, and OSBP contribute to the maintenance of the distribution of cholesterol in the cell.

REVIEWER COMMENTS

Reviewer #1 (Remarks to the Author):

The authors have successfully addressed my concerns.

Reviewer #2 (Remarks to the Author):

In this revised form, the manuscript is still insufficient for publications. The authors' observations are somehow novel, but the manuscript lacks a coherent story line. The manuscript is loaded with various experiments that do not necessarily serve the purpose of the story: How does the endogenous tagging of ORP9 and OSBP serve the manuscript (Fig.1)? What is the added value of the proteomics experiments? It identified ORP10 and ORP11, which authors show not to affect ORP9 localization! Also, why perform immunoprecipitations with all other ORP proteins.

- Authors now show loss of ORP9 by itself is sufficient to change cholesterol distribution. This, sort of, nullifies the point that TKO is needed to identify the observed phenotype.

- The new sensor should be characterized properly. As tandem sensors can cause artefacts (see P4M sensor in Hammond et al. 2014).

- Loss of ORP9 does not increase cellular cholesterol levels despite increasing transcription of genes AND causing SREBP cleavage.

I suggest that the authors clean up and revise the manuscript.

Reviewer #3 (Remarks to the Author):

The authors are to be commended for responding in considerable and specific detail to the previous review. Many questions remain, but these are now well beyond the scope of this already encyclopedic manuscript. With respect to the specific points raised in my review, the authors have responded completely.

REBUTTAL TO THE COMMENTS RAISED BY THE REVIEWERS

We thank the editor and the reviewers for the constructive comments and suggestions. We have carried out extensive new experimentation to address all the concerns and made necessary changes to the texts.

We have now

- 1) characterized the tandem GRAM-H biosensor that we used during the previous revision.
- 2) used the CRISPR/Cas9-mediated gene editing approach (instead of RNAi) and showed that ORP10 and ORP11 are important for the recruitment of ORP9 to the TGN.
- 3) cleaned and edited the manuscript extensively to make it more concise and cohesive.

A specific list of changes is indicated below, followed by a point-by-point rebuttal from page 3.

The order of the presentation of the results is changed for **Figure 3** (previous Figure 3g-h are now Figure 3c-d).

The following data are new:

Characterization of EGFP-GRAM-Hx2

Appendix Fig. 1 for reviewer #2 (showing the characterization of the tandem GRAM-H)

Importance of ORP10 and ORP11 for the recruitment of ORP9 to the TGN

Fig. 2h-l & Supplementary Fig. 2j-k (showing that the double depletion of ORP10 and ORP11 via CRISPR/Cas9-mediated gene knock-out causes dissociation of ORP9 from the TGN)

-Sentences from the reviewers' comments are *in italics*
-Our responses are in blue

Reviewer #1 (Remarks to the Author):

The authors have successfully addressed my concerns.

We would like to thank the reviewer for all the constructive comments to improve our manuscript.

Reviewer #2 (Remarks to the Author):

In this revised form, the manuscript is still insufficient for publications. The authors' observations are somehow novel, but the manuscript lacks a coherent story line. The manuscript is loaded with various experiments that do not necessarily serve the purpose of the story:

We would like to thank the reviewer for the detailed assessment of our manuscript. We appreciate all the constructive comments from this reviewer to further improve our manuscript.

How does the endogenous tagging of ORP9 and OSBP serve the manuscript (Fig.1)?

Reply: We would like to clarify the importance of these experiments. We detected some non-specific signals in the ORP9 KO HeLa cells using the anti-ORP9 antibodies (**Figure 1c, 3b**), which we clarified with the reviewer 1 in the previous round of revision (reviewer 1's comment #8). Similar non-specific signals were also present in HeLa cells immunolabelled with the anti-OSBP antibodies (**Figure 1c**). Furthermore, both anti-ORP9 and anti-OSBP antibodies were produced in rabbits, which prevented us from performing dual immunolabeling experiments to assess the localization of ORP9 and OSBP within the same cell (as we mentioned in the page 7 of our original/revised manuscript). These issues prompted us to use the endogenous tagging approach, as a complementary method, to determine the localization of ORP9 and OSBP in live cells. Using the endogenously tagged cells, we successfully monitored their localization within the same cell in live and found that their localization is differentially regulated; while the localization of OSBP to the TGN is largely dependent on PI4P generated by PI4KIII β , the localization of ORP9 is not (**Figure 1f, g**). This prompted us to further characterize how ORP9 is localized to the TGN in **Figure 2**.

What is the added value of the proteomics experiments? It identified ORP10 and ORP11, which authors show not to affect ORP9 localization!

Reply: We thank the reviewer for raising concerns for these key issues. Being inspired by the reviewer's comments, we re-examined the potential role of ORP10 and ORP11 in the recruitment of ORP9 to the TGN. To this end, we depleted ORP10 and ORP11 in HeLa cells carrying endoORP9-mNG (endogenously tagged ORP9) using the CRISPR/Cas9-mediated gene editing approach (instead of using RNAi as we performed during the previous round of revision) and assessed its impact on the recruitment of endoORP9-mNG to the TGN. We found that depletion of ORP10 and ORP11 caused significant reduction in the interaction of endoORP9-mNG to the TGN immunolabelled by anti-TGN46 antibodies. These new results show the importance of the interaction of ORP9 with ORP10/ORP11 for the recruitment of ORP9 to the TGN. We replaced our old data with RNAi with the new data with CRISPR/Cas9 (**Figure 2h, i, Supplementary Figure 2j, k**) and revised the text as shown below.

We changed the following sentences in the results to be consistent with our new findings (**page 9**).

- Original sentence: "Based on these results, we examined whether depletion of ORP10 and ORP11 affects the localization of ORP9 to the TGN. To this end, HeLa cells carrying endoORP9-mNG (Fig. 1d) were treated with small interfering RNA (siRNA) against ORP10 and ORP11. Double depletion of ORP10 and ORP11 (~90% and ~80%, respectively) was confirmed by western blotting (Supplementary Fig. 2j, k). However, the localization of endoORP9-mNG to the TGN was not affected in this condition (Supplementary Fig. 2l). Thus, the complex formation of ORP9, ORP10, and ORP11 is not essential for the recruitment of ORP9 to the TGN in HeLa cells (please see the discussion)."
- New sentence: "Finally, we examined whether depletion of ORP10 and ORP11 affects the localization of ORP9 to the TGN. To this end, ORP10 and ORP11 were simultaneously depleted via the CRISPR/Cas9-mediated gene editing approach in HeLa cells carrying endoORP9-mNG (Fig. 1d). Double depletion of ORP10 and ORP11 was confirmed by western blotting (Supplementary Fig. 2j, k). The localization of endoORP9-mNG to the TGN was significantly reduced in this condition (Fig. 2h, i). Thus, the complex formation of ORP9, ORP10, and ORP11 is important for the recruitment of ORP9 to the TGN."

We also changed the following sentence in the discussion to be consistent with our new findings (**page 17**).

- Original sentence: "While simultaneous depletion of ORP10 and ORP11 by RNAi did not result in dissociation of ORP9 from the TGN, deletion of the tandem α -helices resulted in dissociation of ORP9

from the TGN, suggesting that homo- and hetero-meric interactions between ORP9 and ORP10/11 might play some roles in the localization or function of ORP9.”

- New sentence: “Simultaneous depletion of ORP10 and ORP11 resulted in dissociation of ORP9 from the TGN, showing that hetero-meric interactions between ORP9 and ORP10/11 contribute to the localization of ORP9.”

Also, why perform immunoprecipitations with all other ORP proteins.

Reply: We thank the reviewer for this comment. We agree with the reviewer that performing immunoprecipitations with all other ORPs does not add much value to our manuscript. In the revised manuscript, we only show the relevant immunoprecipitation data to confirm the interactions between ORP9 and ORP10/ORP11 as detected via the proteomics experiments (**Supplementary Figure 2h**).

We changed the following sentences in the results to be consistent with our new findings (**page 9**).

- Original sentence: “To further examine whether ORP9 interacts with OSBP or any other ORPs, EGFP-ORP9 was co-expressed together with one of 12 mRuby-tagged ORPs in HeLa cells. Potential interactions were assessed via anti-EGFP immunoprecipitation followed by western blotting.”
- New sentence: “To confirm their interactions, EGFP-ORP9 was co-expressed together with mRuby-tagged ORP9, ORP10 or ORP11 in HeLa cells. Interactions were assessed via anti-EGFP immunoprecipitation followed by western blotting.”

- Authors now show loss of ORP9 by itself is sufficient to change cholesterol distribution. This, sort of, nullifies the point that TKO is needed to identify the observed phenotype.

Reply: We thank the reviewer for these comments. As we showed in our manuscript, accumulation of accessible cholesterol in the Golgi of ORP9 KO cells was further enhanced by the additional depletion of GRAMD1s with major accumulation in the PM (i.e., QKO cells lacking both ORP9 and GRAMD1s). Therefore, QKO cells were needed to study the interplay between ORP9 and GRAMD1s in the regulation of cellular cholesterol distribution. To make this point clearer, we changed the order of the presentation of our data in the section of results “Depletion of ORP9 causes accumulation of accessible cholesterol in the Golgi, which is further enhanced by the additional depletion of GRAMD1s with major accumulation in the PM” and extensively revised the relevant texts (**page 9-10**). In our revised manuscript, we now present the results from the experiments using EGFP-GRAM-Hx2 before presenting the results from the experiments using EGFP-GRAM-H (**Figure 3**) to emphasize the importance of ORP9 for the regulation of cellular cholesterol distribution. We think that the new order also clarifies why QKO cells were used/needed in the subsequent experiments for the dissection of ORP9 and its interplay with GRAMD1s in the regulation of cellular cholesterol distribution.

Finally, we edited the following sentences in the results to be more clear about this point (**page 10**).

- Original sentence: “Collectively, these results show that ORP9 is important for proper distribution of cholesterol in the Golgi. Importantly, accumulation of accessible cholesterol in ORP9 KO cells was further enhanced when GRAMD1s were additionally depleted in these cells (i.e., QKO cells), suggesting that ORP9 and GRAMD1s function together. Unless otherwise noted, QKO cells were used in the rest of the study to dissect the interplay between ORP9 and GRAMD1s in regulating cellular cholesterol distribution.”
- New sentence: “Collectively, these results show that ORP9 is important for proper distribution of cholesterol in the Golgi. Importantly, accumulation of accessible cholesterol in ORP9 KO cells was further enhanced by the additional depletion of GRAMD1s with major accumulation in the PM (i.e., QKO cells), suggesting that ORP9 and GRAMD1s functionally cooperate. Unless otherwise noted, QKO cells were used in the rest of the study to dissect the role of ORP9 and its interplay with GRAMD1s in the regulation of cellular cholesterol distribution.”

- The new sensor should be characterized properly. As tandem sensors can cause artefacts (see P4M sensor in Hammond et al. 2014).

Reply: We thank the reviewer for these comments. We performed additional experiments to characterize the property of the tandem GRAM-H biosensor (**please see below the results of these experiments**). In the first set of experiments, we expressed EGFP-tagged tandem GRAM-H (EGFP-GRAM-Hx2) in WT and ORP9 KO HeLa cells and examined if expression of EGFP-GRAM-Hx2 would cause disruption of the Golgi structures or changes in the total areas of the cells. As shown in the **Appendix Figure 1a, b**, expression of EGFP-GRAM-Hx2 did not cause disruption of the Golgi structures, as assessed by immunolabelling against GM130 and TGN46, nor changes in the total areas of the cells. In the second set of experiments, we show that EGFP-GRAM-Hx2 maintains the sensitivity against accessible cholesterol in cellular membranes. Depletion of accessible cholesterol via combined treatment with mevastatin and lipoprotein-deficient serum (LPDS) resulted in dissociation of EGFP-GRAM-Hx2 to cytosol from the plasma membrane (PM) (in the case of WT HeLa cells) and from the PM and the Golgi (in the case of ORP9 KO HeLa cells) (**Appendix Figure 1c**). Depletion of accessible PM cholesterol via methyl- β -cyclodextrin (MCD) resulted in rapid dissociation of EGFP-GRAM-Hx2 from the PM similar to EGFP-GRAM-H [assessed by total internal reflection fluorescence (TIRF) microscopy] (**Appendix Figure 1d**). Taken together, tandem GRAM-H detects accessible cholesterol with higher sensitivity compared to original GRAM-H without causing major artefacts.

Appendix Fig. 1. **a.** Confocal images of fixed WT and ORP9 KO HeLa cells expressing either EGFP or EGFP-GRAM-Hx2, that were immunolabeled with antibodies against TGN46 and GM130. Scale bars, 10 μ m. **b.** Quantification of the cell size of the cells as described in (a) [mean \pm SEM, n = 44 cells (WT & EGFP), n = 35 cells (WT & EGFP-GRAM-Hx2), n = 37 cells (ORP9 KO & EGFP), and n = 38 cells (ORP9 KO & EGFP-GRAM-Hx2); data are pooled from two independent experiments; two-tailed unpaired Student's t-test, n.s. denotes not significant]. **c.** Confocal images of live WT and ORP9 KO HeLa cells expressing EGFP-GRAM-Hx2 that were treated with DMEM supplemented with 10% FBS (Control) or 10% LPDS and 50 μ M Mevastatin for 16 hrs. **d.** left: Time course of normalized EGFP-GRAM-H or EGFP-GRAM-Hx2 signals, as assessed by TIRF microscopy, from HeLa cells expressing either EGFP-GRAM-H, or EGFP-GRAM-Hx2, as indicated. Methyl- β -cyclodextrin (MCD) treatment (5 mM) is indicated. right: Values of $\Delta F/F_{0\text{min}}$ corresponding to the timepoint as indicated by the arrow [mean \pm SEM, n = 52 cells (GRAM-H), n = 47 cells (GRAM-Hx2); data are pooled from two independent experiments for each condition; two-tailed unpaired Student's t-test, **P < 0.0001].

- Loss of ORP9 does not increase cellular cholesterol levels despite increasing transcription of genes AND causing SREBP cleavage.

Reply: We agree with the statement of this reviewer. In fact, we had discussed this issue in the discussion during the previous round of revision (**Page 19**). Key points of this discussion are highlighted in yellow:

“Importantly, this abnormal distribution of cholesterol is coupled with dysregulation of the SREBP2 signalling pathway and aberrant cholesterol production. Interestingly, cells lacking only ORP9 or only GRAMD1s do not show major changes in total levels of cellular cholesterol despite some accumulation of accessible cholesterol in the Golgi or the PM, respectively. Therefore, it seems that cells adapt to a certain degree of

increase in the accessible pool of cholesterol in cellular membranes and maintain total levels of cellular cholesterol. The aberrant activation of SREBP-2 and increased levels of total cellular cholesterol in QKO cells are likely due to combined effects of OSBP hyperactivation (due to the depletion of ORP9) and inefficient cholesterol transport to the ER (due to the depletion of GRAMD1s). As depletion of ORP9 alone leads to moderate upregulation of SREBP-2 signalling pathways, the activity of OSBP seems to be tightly linked to that of SREBP-2. When hyperactivation of OSBP is coupled with inefficient cholesterol transport to the ER (as in the case of QKO cells), SREBP-2 is hyperactivated, resulting in dysregulation of cholesterol production. Thus, our study suggests that intricate crosstalk between the ER and Golgi, which is regulated by ORP9, OSBP, and the GRAMD1s at ER-Golgi contacts, plays a critical role in determining cellular cholesterol distribution and maintaining normal cell physiology.”

I suggest that the authors clean up and revise the manuscript.

Reply: We would like to thank the reviewer once again for all the constructive suggestions to improve our manuscript. We looked through our manuscript carefully based on this reviewer’s comments and cleaned it up to make it more concise and cohesive.

Reviewer #3 (Remarks to the Author):

The authors are to be commended for responding in considerable and specific detail to the previous review. Many questions remain, but these are now well beyond the scope of this already encyclopedic manuscript. With respect to the specific points raised in my review, the authors have responded completely.

We would like to thank the reviewer for all the constructive and critical assessment to improve our manuscript.

REVIEWERS' COMMENTS

Reviewer #2 (Remarks to the Author):

Authors addressed my concerns with minor exceptions:

1) A fundamental control is missing: Double knockout cells should be also immunoblotted for ORP9 to show that the protein levels are not altered when ORP10 and ORP11 is deleted. Without this control their claim does not hold.

2) I suggest authors do not omit the knockdown experiments regarding ORP9/ORP10/ORP11. The discrepancy should be reported including western blots showing knockdown efficiencies, including effect on ORP9 levels (as above).

With these included, the manuscript is okay.

REBUTTAL TO THE COMMENTS RAISED BY THE REVIEWERS

We thank the editor and the reviewer for their constructive comments and suggestions. We have addressed all the concerns and made necessary changes to the texts.

We have now

- 1) shown that the protein levels of ORP9 are not altered by the depletion of ORP10 and ORP11.
- 2) performed additional experiments to determine the effects of RNAi-mediated double knock down of ORP10 and ORP11 on the recruitment of ORP9 to the TGN.

A specific list of changes is indicated below, followed by a point-by-point rebuttal from page 2.

The following data are new:

Supplementary Fig. 2j (showing that the double depletion of ORP10 and ORP11 either via the CRISPR/Cas9-mediated gene editing approach or via the RNAi-mediated knockdown approach does not affect the protein levels of ORP9)

Supplementary Fig. 2i-o (showing that the double knockdown of ORP10 and ORP11 via RNAi for 96 hours causes dissociation of ORP9 from the TGN, albeit less effectively compared to the CRISPR/Cas9-mediated gene editing approach)

-Sentences from the reviewers' comments are *in italics*

-Our responses are in blue

Reviewer #2 (Remarks to the Author):

Authors addressed my concerns with minor exceptions:

We would like to thank the reviewer for all the constructive comments to further improve our manuscript.

1) A fundamental control is missing: Double knockout cells should be also immunoblotted for ORP9 to show that the protein levels are not altered when ORP10 and ORP11 is deleted. Without this control their claim does not hold.

Reply: We thank the reviewer for this comment. We agree with the reviewer for the importance of the control experiment for this assay. We performed additional immunoblotting experiments to determine the protein levels of ORP9 in cells, where ORP10 and ORP11 are simultaneously depleted by the CRISPR/Cas9-mediated gene editing approach. In the revised manuscript, we show that the depletion of ORP10 and ORP11 did not affect the protein levels of the endogenously-tagged ORP9, as assessed by anti-ORP9 antibodies (**Supplementary Figure 2j**). Based on these results, we conclude that the depletion of ORP10 and ORP11 results in the dissociation of ORP9 from the TGN without causing a major impact on its expression. We included these new data in the revised manuscript.

2) I suggest authors do not omit the knockdown experiments regarding ORP9/ORP10/ORP11. The discrepancy should be reported including western blots showing knockdown efficiencies, including effect on ORP9 levels (as above).

Reply: We thank the reviewer for this comment. In the earlier version of our manuscript, we showed that the double knockdown of ORP10 and ORP11 by RNAi for 72 hours did not affect the TGN localization of ORP9 (unlike the double knockout of ORP10 and ORP11 by the CRISPR/Cas9-mediated gene editing). To address the discrepancy between the results obtained via the RNAi-mediated knockdown and the results obtained via the CRISPR/Cas9-mediated knockout, we compared the efficiency of the deletion of ORP10 and ORP11 proteins and found that CRISPR/Cas9-mediated knockout resulted in more efficient depletion of ORP10 and ORP11 compared to RNAi-based knockdown when we measured the depletion efficiency 72 hours after RNAi (i.e., the condition which we used in our earlier experiments). We performed additional RNAi experiments and found that treating cells with siRNAs against ORP10 and ORP11 for 96 hours (instead of 72 hours) enhanced the efficiency of the depletion of ORP10 and ORP11. Using this new condition, we repeated the imaging experiment and found that RNAi-mediated knockdown of ORP10 and ORP11 also resulted in the dissociation of ORP9 from the TGN, albeit less effectively compared to the CRISPR/Cas9-mediated knockout of ORP10 and ORP11 (**new Supplementary Figure 2i-o**). Importantly, protein levels of ORP9 were not affected by the RNAi, consistent with the results obtained using the CRISPR/Cas9-mediated gene editing (**new Supplementary Figure 2i**). We included these new RNAi data in the revised manuscript.

We changed the following sentences in the results to be consistent with our data (**page 8**).

- Original sentence: "To this end, ORP10 and ORP11 were simultaneously depleted via the CRISPR/Cas9-mediated gene editing approach in HeLa cells carrying endoORP9-mNG. Double depletion of ORP10 and ORP11 was confirmed by western blotting (**Supplementary Fig. 2j, k**). The localization of endoORP9-mNG to the TGN was significantly reduced in this condition (**Fig. 2h, i**). Thus, the complex formation of ORP9, ORP10, and ORP11 is important for the recruitment of ORP9 to the TGN."
- New sentence: "To this end, ORP10 and ORP11 were simultaneously depleted either via the CRISPR/Cas9-mediated knockout (KO) approach or via the RNAi-mediated knockdown (KD) approach

in HeLa cells carrying endoORP9-mNG. Double depletion of ORP10 and ORP11 was confirmed by western blotting (**Supplementary Fig. 2j-m**). There was no major impact on the expression of endoORP9-mNG (**Supplementary Fig. 2j, l**). Importantly, the localization of endoORP9-mNG to the TGN was significantly reduced, although such reduction was more prominent in the double KO (ORP10/ORP11 DKO) cells compared to double KD (ORP10/ORP11 KD) cells (**Fig. 2h, i and Supplementary Fig. 2n, o**). Thus, the complex formation of ORP9, ORP10, and ORP11 is important for the recruitment of ORP9 to the TGN.”

With these included, the manuscript is okay.

We would like to thank the reviewer again for all the constructive comments.